# A glyoxal-specific aldehyde signaling axis in *Pseudomonas aeruginosa* that influences quorum sensing and infection

Christopher J. Corcoran [1,20], Bonnie J. Cuthbert [2], David G. Glanville[3], Mailyn Terrado[1], Diana Valverde Mendez [4,5], Benjamin P. Bratton [6,7,8,9], Daniel E. Schemenauer[1], Valerie L. Tokars[10,11,21], Thomas G. Martin[12,22], Lawrence W. Rasmussen[3], Matthew C. Madison [3], Andrew F. Maule[13], Joshua W. Shaevitz [4,9], Boo Shan Tseng [14], Julian P. Whitelegge [15,16], Catherine Putonti [17], Amit Gaggar[3], Jordan R. Beach[12], Jonathan A. Kirk [12], Alfonso Mondragón [18], Abby R. Kroken [1], Jonathan P. Allen [1], Celia W. Goulding[19] & Andrew T. Ulijasz [3] ✉

The universally conserved α-oxoaldehydes glyoxal (GO) and methylglyoxal (MGO) are toxic metabolic byproducts whose accumulation can lead to cell death. In the absence of a known, natural inducer of the GO-specific response in prokaryotes, we exploited RNA-seq to define a GO response in the bacterial pathogen *Pseudomonas aeruginosa*. The highest upregulated operon consisted of the known glyoxalase (*gloA2*) and an antibiotic monooxygenase (ABM) domain of unknown function - renamed here **A**ldehyde **r**esponsive **q**uorum-sensing **I**nhibitor (ArqI). The *arqI-gloA2* operon is highly specific to GO induction and ArqI protein responds by migrating to the flagellar pole. An ArqI atomic structure revealed several unique features to the ABM family, including a 'pinwheel' hexamer harboring a GO-derived post-translational modification on a conserved arginine residue (Arg49). Induction of ArqI abrogates production of the Pseudomonas Quinolone Signal (PQS) quorum sensing molecule and was found to directly interact with PqsA; the first enzyme in the PQS biosynthesis pathway. Finally, we use a sepsis model of infection to reveal a survival requirement for *arqI-gloA2* in blood-rich organs (heart, spleen, liver and lung). Here we define a global GO response in a pathogen, identify and characterize the first GO-specific operon and implicate its role in PQS production and host survival.

The low molecular weight toxic α-oxoaldehydes glyoxal (GO) and methylglyoxal (MGO) are byproducts of basic cellular processes, including glycolysis, lipid peroxidation, and oxidation of DNA and glucose, and are therefore found in all three kingdoms of life[1]. To help counter these toxins, cells have evolved glyoxalase (Glo) enzymes to metabolize GO and MGO into glycolic acid or lactate, respectively, which are nontoxic and can reenter central metabolic processes. However, when allowed to accumulate, GO and MGO can react with arginine, lysine, cysteine and histidine to produce irreversible modifications classically referred to in eukaryotes as Advanced Glycation End products (or AGEs)[1]. While both GO and MGO are byproducts of glycolysis, they are generated by different pathways. MGO is derived

enzymatically from dihydroxyacetone phosphate (DHAP) through MGO synthase, or alternatively can form from DHAP spontaneously, whereas GO is a product of glycolaldehyde oxidation, or oxidation of lipids or DNA[1]. Regardless of their origin, AGE accumulation results in dysfunctional proteins, which are the underlying cause of numerous human diseases[2–5]. Although an MGO-specific responsive class of receptors (Receptor(s) for AGE; or RAGE) has been identified in eukaryotes, and MGO responsive pathways have been somewhat examined in prokaryotes[6,7], GO-specific data is almost non-existent[1].

Bacterial studies have identified response and remediation systems to both GO and MGO treatment. These studies are mainly confined to non-pathogenic *E. coli*, where the transcription factor YqhC responds to GO to activate several aldehyde reductase enzymes, and the transcription factor NemR responds to general reactive electrophilic species (RES; GO, MGO, quinones) and upregulates the N-ethylmaleimide reductase NemA and the glyoxylase GloA[1,8,9]. Interestingly, NemR also responds to hypochlorous acid (HOCl) or "bleach" used by neutrophils to kill pathogens[10]. On the other hand, GO-specific pathways and the natural circumstances by which this aldehyde induces the conserved and dedicated signaling and remediation in bacteria has remained enigmatic. Therefore, given the paucity of information surrounding the GO response in bacteria, and that GO detoxification systems have recently been identified as critical for infection with viral, fungal and bacterial pathogens[11–16], we examined the global response to GO treatment in the Gram-negative pathogen *Pseudomonas aeruginosa*.

In this work we demonstrate that GO induces a novel, GO-specific, two-gene operon consisting of *gloA2*, which encodes a glyoxalase to counter aldehyde assaults, and *arqI* (**A**ldehyde **r**esponsive **q**uorum-sensing **I**nhibitor), which encodes an ABM domain protein that influences the synthesis of the major quorum sensing Pseudomonas Quinolone Signal (PQS) molecule. An ArqI atomic resolution structure revealed an unusual hexamer wheel-like configuration with a novel GO-derived posttranslational modification (PTM) on a conserved arginine reside (Arg49). Upon GO induction, we show that AqI localizes to the flagellar pole of the cell and inhibits PQS biosynthesis. We then show that ArqI interacts both in vivo and in vitro with first enzyme in the PQS biosynthesis pathway (PqsA). Finally, we demonstrate that this operon modulates sepsis colonization of downstream tissues which suggests involvement of the GO response in *P. aeruginosa* phagocyte clearance. From these data we propose a model where the GO-response is triggered by an unknown mechanism during infection, resulting in *arqI-gloA2* operon synthesis and other GO remediation genes to enable host survival.

## Results

### Global response to GO in *P. aeruginosa*

How bacteria respond to GO induction is poorly characterized[1,7]. To address this problem, we first determined GO and MGO minimal inhibitory concentrations (MICs) in LB. We chose rich (LB) medium for our studies so as to not restrict carbon sources and favor certain metabolic pathways. GO and MGO MICs were determined to be 12.5 and 7.0 mM, respectively (Supplementary Fig. 1A). Next, we treated *P. aeruginosa* grown in LB with a sub-MIC concentration of GO (4 mM), and incubated cells for either 15 min or 1 h before harvesting and performing RNA-seq in triplicate.

Several genes were differentially expressed with GO treatment: 104 differentially expressed genes were shared between the 15 min and 1-h timepoints and 43 genes were unique to the 1-h or "late expressed" timepoint. Surprisingly, in many cases the 15 min and 1 h GO treated data exhibited opposite expression of certain genes/operons, indicating a truly dynamic response and metabolic readjustment to GO remediation over time (Fig. 1A; Supplementary Information (SI)). STRING[17] was used to compare related gene families that were differentially expressed greater than 3-fold (Supplementary Fig. 1B–G; **SI**).

Addition of GO resulted in the activation of a sulfur starvation response[18,19] and included upregulation of genes encoding sulfur-compound transporters and enzymes involved in their respective catabolism, e.g., taurine (*tau*), alkanesulfonate (*ssu*) and cysteine (*cys*)[20,21]. These operons import sulfur resources and enzymatically liberate free sulfur to replenish lost elemental sulfur pools, likely lost due to cellular sulfur reacting with the excessive and highly reactive GO aldehyde. In particular, after GO addition three separate genes encoding predicted taurine dioxygenases (PA3935 (*tauD1*), PA2310 (*tauD2*), PA0193 (*tauD3*) (Supplementary Fig. 2) - bacterial homologs of the human mitochondrial α-oxoaldehyde remediation enzyme AlkB 7 dioxygenase (ALKBH7)[22], and an alkanesulfonate monooxygenase (*ssuD*; PA3444), were upregulated by several orders of magnitude. Moreover, upon GO exposure reactive oxygen species (ROS) detoxifying genes were upregulated (*katB*, *trxB2*, *ahpF*, *ahpB*, and *ohr*)[23], which have also been associated with the bacterial sulfur starvation response[19]. A bacterial homolog of human Parkinson's disease gene PARK7 (DJ-1, PA4336)[24,25] was also strongly upregulated after GO exposure[26] (Fig. 1A).

RNA transcripts that diverged from reported sulfur starvation responses included a GO-mediated global downregulation of the anaerobic denitrification pathway (e.g., the nitrite reductase *nirS*;[27,28]), anaerobic metabolism regulators *dnr*[29] and *anvM*[30] and the recently described sRNA *sicX*[31] that regulates the oxygen-dependent switch to chronic *P. aeruginosa* infection (14-fold reduction; **SI**). Despite the strong downregulation of the denitrification pathway, a large GO-induced upregulation was observed for the nitrous oxide ($N_2O$) reduction operon (*nos*) that catalyzes the last step in denitrification. This process is preceded by the nitric oxide (NO) reductase (*nor* operon) step that rids the cell of toxic NO used by neutrophils to eliminate pathogens[32].

Upon GO exposure, a global repression in iron uptake and utilization pathways was also observed, which included the siderophore (pyoverdine) $Fe^{3+}$−PQS transporter (*fptA*), and pyoverdine synthesis gene *pvdH* (PA4221; only at the 1-h timepoint - see **SI**), ferric iron uptake operon *hitAB* (PA4687/PA4688), ferrous iron-dependent dioxygenase *piuC*, heme synthesis and catabolism genes (*hemN* and *hemO*, respectively), and bacterial ferritin *bfrB*. Additionally, we saw a strong downregulation of the phosphate uptake and sensing *pst* system[33,34] (at the 1-h timepoint only), as well as molybdenum uptake *icmP* genes[35]. Collectively, these responses fit well into a hypothesis whereby phosphate and metals such as iron and molybdenum, which can act with ROS as catalysts to facilitate GO formation in the presence of glyceraldehyde[36,37], are prevented from entering the cell after GO exposure.

Another discovery was the induction of several virulence genes in response to GO, including the encoded extracytoplasmic function (ECF) sigma factor *vreI* ($\sigma^{VreI}$), and adjacent receptor-like protein, *VreA*[38] (only during the early 15 min timepoint GO exposure; **SI**). In other reports the *vreIAR* (or PUMA3) operon is induced during phosphate starvation and, in turn, induces the *hxc* operon; one of two Type II Secretion System (T2SS) operons found in *P. aeruginosa*[39,40]. The hemolytic phospholipase C (*plcH*) gene, which is secreted by the T2SS, was also upregulated with GO treatment. PlcH degrades host phosphatidylcholine to allow internalization and is used as a major source of carbon during infection[41]. Overall, the RNA-seq data highlights a vast network of cellular responses to GO exposure, including sulfur starvation, iron uptake and utilization pathways, and induction of virulence factors.

### Interrogation of the GO-specific *arqI-gloA2* operon

The most highly upregulated region in response to GO at both 15-min and 1-h timepoints was an operon encoding an ABM domain of unknown function (PA0709) and the glyoxalase GloA2 (PA0710) (Fig. 1A, B; **SI**), whose synteny is conserved in most *P. aeruginosa*

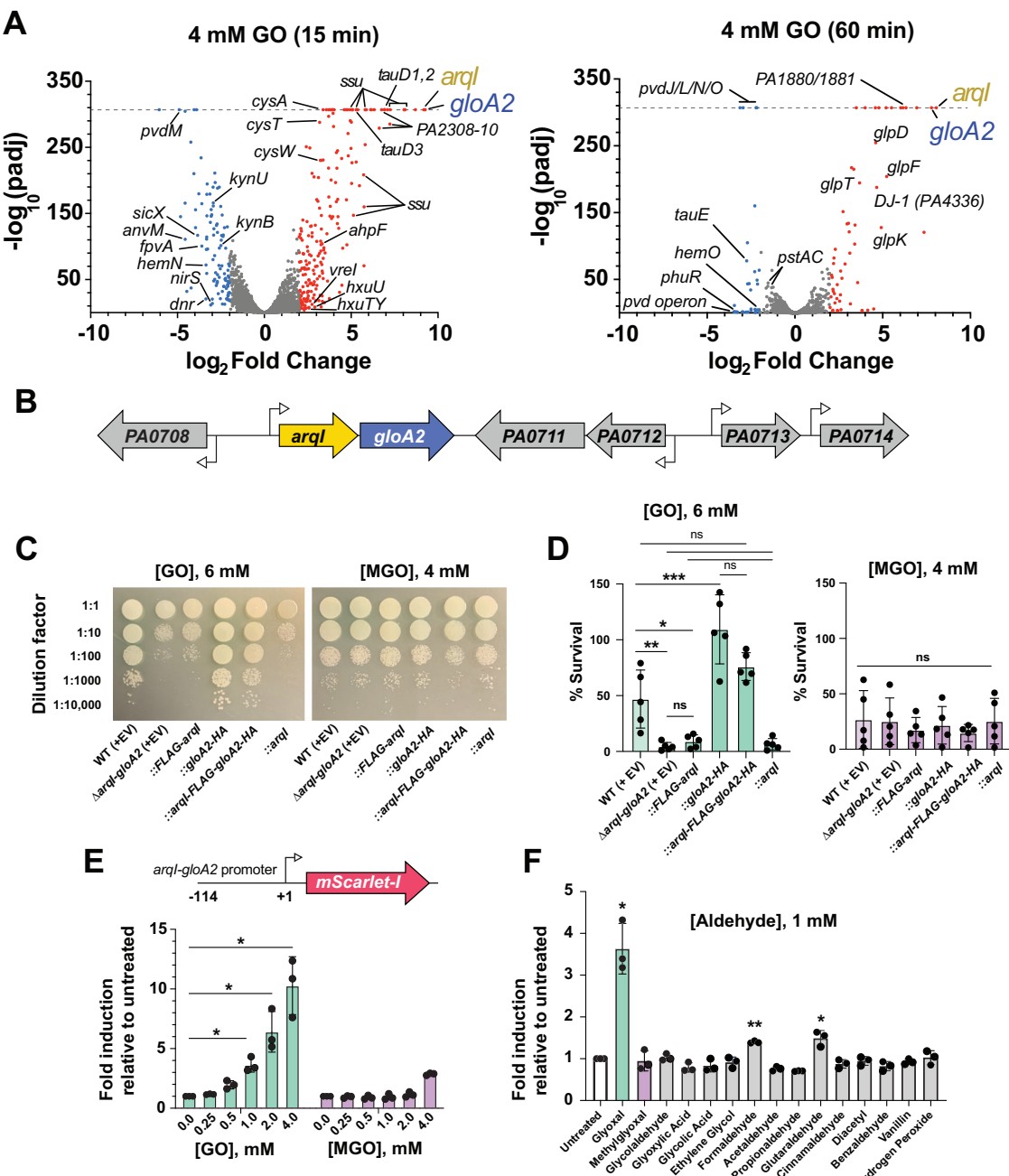

**Fig. 1 | Transcriptional response of *P. aeruginosa* to glyoxal (GO). A** Volcano plot of differentially regulated genes by RNA-sequencing upon exposure of *P. aeruginosa* to 4 mM GO for 15-min (left panel) or 1-h (right panel). Dotted line indicates the limit of plotting for the y-axis. $n = 3$ biological replicates. **B** Schematic of the GO-responsive locus encompassing genes *PA0709 (arqI)* to *PA0714*. **C** Spot killing assays of the indicated *P. aeruginosa* MPAO1 strains on LB agar media containing 6 mM GO and 4 mM MGO. **D** Quantification of *C* encompassing 5 biological repeats. **E** Above: Schematic diagram of mScarlet-I *arqI-gloA2* promoter fusion. Below: the *arqI-gloA2* promoter is induced by GO in a dose-dependent manner. Reporter activity was assessed 3 h post addition. $n = 3$ biological replicates. **F** Specificity of *arqI-gloA2* transcriptional induction by various aldehydes and structurally similar chemicals to GO. Compounds were added to a concentration of 1 mM. $n = 3$ biological replicates. Statistics: A two-sided Wald test was utilized to calculate *P* values for *A*. Bars indicate arithmetic mean. Whiskers denote standard deviation. *$P < 0.05$; **$P < 0.01$; ***$P < 0.001$ as an ordinary one-way ANOVA with Tukey's multiple comparisons test for *D* and two-sided one-sample *t* test (against untreated control) for *E* and *F*. Significant *P* values for *1 d* for GO addition were calculated as (left to right): 0.0096, 0.0252, 0.0001. For *1e*: (GO) 0.0172, 0.0314, 0.0221 and *1f*: 0.0172, 0.0017 and 0.0432. Source data are provided as a Source Data file.

strains (Supplementary Fig. 3A). ABM domain proteins can have enzymatic activity towards a variety of substrates, including heme and polyketides[42–45]. The observation that PA0709 expression coincided with heme biosynthesis and uptake regulation (Fig. 1A; SI), validated our previously observed association of PA0709 in iron/heme regulatory processes[46]. Given its newfound role in the GO response and quorum sensing (see below), PA0709 was subsequently assigned

the name **A**ldehyde-**r**esponsive **q**uorum-sensing **I**nhibitor (ArqI). Wild-type MPAO1 and the *arqI-gloA2* operon deletion strain (Δ*arqI-gloA2*) were tested for their ability to survive both GO and MGO challenges. Upon GO exposure the MIC of the Δ*arqI-gloA2* strain was markedly reduced (from 12.5 mM to 8.5 mM), whereas MGO had no observed effect and even trended upwards in the MIC with the mutant (Supplementary Fig. 1A). We next complemented the Δ*arqI-gloA2*

strain with (i) the whole operon, (ii) *arqI* (both with and without FLAG-tag) or (iii) *gloA2* (with HA-tag) individually using an arabinose inducible plasmid[47]. Strains were then cultured, and serial dilutions were spot-plated onto LB plates containing the appropriate antibiotic selection, arabinose and 6 mM GO (a concentration below the MIC). Results shown in Fig. 1C demonstrate that deletion of the *arqI-gloA2* operon substantially diminished GO resistance, but not MGO resistance. Both the wild-type and FLAG-tagged complemented versions of ArqI were unable to restore measurable GO resistance to statistical significance (however an approximate 2-fold increase was observed compared to the Δ*arqI-gloA2* deletion strain), whereas complementation of *gloA2* increased resistance to GO treatment compared to complementing with the full operon (Fig. 1C, D). We attribute the latter to a higher overall expression of GloA2 when compared to that of its expression within the native operon (Supplementary Fig. 3B). These results show that GloA2 plays a highly specific role in GO remediation. On the other hand, at least under these in vitro conditions, ArqI was shown to have no appreciable role in GO resistance.

After predicting the *arqI-gloA2* transcriptional start site (Supplementary Fig. 3c), we monitored *arqI-gloA2* operon native promoter activity using a new mScarlet-I transcriptional reporter we constructed for these studies (Fig. 1E, above). Addition of GO, but not MGO, resulted in a dose-dependent increase in mScarlet-I signal (Fig. 1E, below), and was shown to be highly specific to GO after a battery of other aldehyde compounds were tested for induction, with only formaldehyde and glutaraldehyde compounds exhibiting slight statistically significant increases compared to the no addition control (Fig. 1F). To the best of our knowledge, this is the first example of a highly GO-specific induced gene/operon.

## Atomic structure of ArqI reveals unique quaternary and dimer-dimer interface features

To gain better insight into the functional role of ArqI in glyoxal signaling we initiated structural studies and solved the ArqI structure to atomic resolution (2.0 Å; PDB ID 8ECX; Supplementary Table 1). The crystal structure of ArqI revealed a D3 symmetric hexamer ("trimer of dimers" hexamer) (Fig. 2A; Supplementary Fig. 3D). Negative stain EM showed a volume with perpendicular 2- and 3-fold axes, consistent with D3 symmetry. Volumes were calculated using C2, C3, and D3 symmetry, and C2 symmetry was selected as it resulted in the smoothest Fourier Cell Correlation (FSC), but still showed the 3-fold axis (Supplementary Fig. 3E, F). The ArqI crystal structure hexamer fit very well into the EM volume (Supplementary Fig. 3F). Finally, size exclusion chromatography (SEC) demonstrated a predicted mass of 70,188 Da, which was in line with a hexameric oligomer state (Supplementary Fig. 3G). The ArqI hexamer was determined to be unique in the ABM domain family after employing structural homology searches with DALI (Supplementary Fig. 4A)[48].

The ArqI trimeric interface consists of four major dimer-dimer contacts. Asp81 (α2) from one dimer interacts with (i) Ser13 (loop between β1 and α1), (ii) Arg16 (α1), (iii) Arg49 (loop between β2 and β3) of an adjacent dimer, and (iv) Gly86 (loop between α2 and β4) and Asn48 (loop between β2 and β3) interact between neighboring dimers (Fig. 2B)[49]. Interestingly, ArqI has a proline (Pro87) located between α2 and β4 in the structure that adopts the *cis* conformation in every subunit, orienting Gly86 so that it can participate in trimeric contacts (Fig. 2B). The *cis*-bonded Gly-Pro, conserved in many ABM domains (*e.g.* see Supplementary Fig. 4B), is clearly an important structural feature which ensures that Leu88 is buried. Mutating Pro87 would therefore be predicted to produce a structural effect and result in the repositioning of Phe85, which together with the Phe85 from the other two monomers forms a hydrophobic (Phe85) "gate" to the central cavity of the trimer (Fig. 2C).

## The ArqI dimer is most similar to LsrG family ABM domains

The basic ArqI homodimer was most similar to characterized LsrG proteins[49] (Supplementary Fig. 4B). LsrG enzymes bind the universally conserved bacterial quorum sensing autoinducer-2 (AI-2) precursor phospho-(S)−4,5-dihydroxy-2,3-pentanedione (P-DPD), whose reaction eventually yields 3,4,4-trihydroxy-2-pentanone-5-phosphate (P-TPO)[49]. ArqI aligns to *E. coli* and *Y. pestis* LsrG with 1.1 and 1.0 Å root mean square deviation (rmsd) across 482 and 483 atoms (PDB ID 2GFF and 3QMQ[49]) and has 34% and 39% sequence identity to ArqI, respectively. ArqI also aligns to *M. tuberculosis* Rv0793 ABM domain with 1.1 Å rmsd and *S. nogalater* SnoaB ABM domain with 1.5 Å rmsd (PDB ID 1Y0H and 3KG0)[48,50,51], and includes several conserved residues within the inter-trimeric interface (Supplementary Fig. 4A,B)[49]. In addition, some LsrG structures include the equivalent G86/P87 motif (Supplementary Fig. 4B), suggesting they might function by a similar mechanism to ArqI. However, the equivalent proline residue in LsrG structures was found in the more common *trans* isomer, resulting in an -180˚ flip of the G86/P87 motif, and freeing the ArqI Leu88 equivalent residue (Arg88) from being buried in the inter-dimer pocket (Supplementary Fig. 4C), suggesting that a *cis-trans* flip of the proline would result in dramatic structural rearrangements.

We previously showed that ArqI binds heme[46]. However, when compared with the canonical heme binding pocket defined by heme degrading ABM domains[52,53], the equivalent ArqI structure would be too small to accommodate such a large molecule without dramatic structural rearrangements. This suggests that ArqI either binds heme in a non-canonical location, and/or accommodates an unidentified, smaller physiological ligand in the typical ABM pocket. We therefore modeled potential ligands DPD, MGO, and acetate (AC; as GO was deemed too small to model) against two LsrG homologs (PDB codes 2GFF and 3QMQ) and ArqI using the program HDOCK[54]. With all three ABM domains we could model all tested ligand binding with high confidence in their predicted binding pockets[49] (Supplementary Fig. 4D, E). These results suggest that ArqI could have evolved from an LsrG ancestor to accommodate a novel biological role in the aldehyde response. Indeed, the DPD molecule contains MGO as part of its structure (in red; Supplementary Fig. 4F).

## Two unusual post-translational modifications (PTMs) within the ArqI crystal

Further examination of the ArqI structural data revealed extra electron density stemming from for the sidechains of two individual protein residues, suggesting two separate PTMs. The first PTM was determined to be a βME modification on Cys68 (α2 helix; Fig. 2D; Supplementary Fig. 5A) by mass spectrometry (MS) verification using the protein crystal (+76 amu shift; Supplementary Fig. 5). The second ArqI PTM was GO-derived and on the R group of Arg49, but only was seen in subunits C, D, and E of the hexameric wheel (Fig. 2E; Supplementary Fig. 5B). Given the 3-dimensional placement of Arg49 in the structure, this modification could assist in further stabilization of the trimeric contacts due to several extra static interactions afforded by the PTM (Fig. 2F). All potential GO-derived arginine adducts (Supplementary Fig. 5C) were modeled and refined in the experimental electron density and indicated that linear form adducts (hemiaminal, carboxymethylarginine or Schiff base) fulfilled the electron density better than closed-ring adducts (dihydroxyimidizolidine or hydroimidazolone) (Fig. 2E; Supplementary Fig. 5B). MS performed on a ArqI crystal revealed a mass shift of +58 amu for peptides bearing Arg49, indicating no water loss. This observation pointed to either a hemiaminal or carboxymethylarginine modification, which have identical masses (Supplementary Fig. 5C, D).

We then verified the Arg49 modification was due to GO by expressing ArqI in *E. coli* cells and adding increasing concentrations of GO (0-2 mM). MS analysis of purified ArqI generally showed a dose-

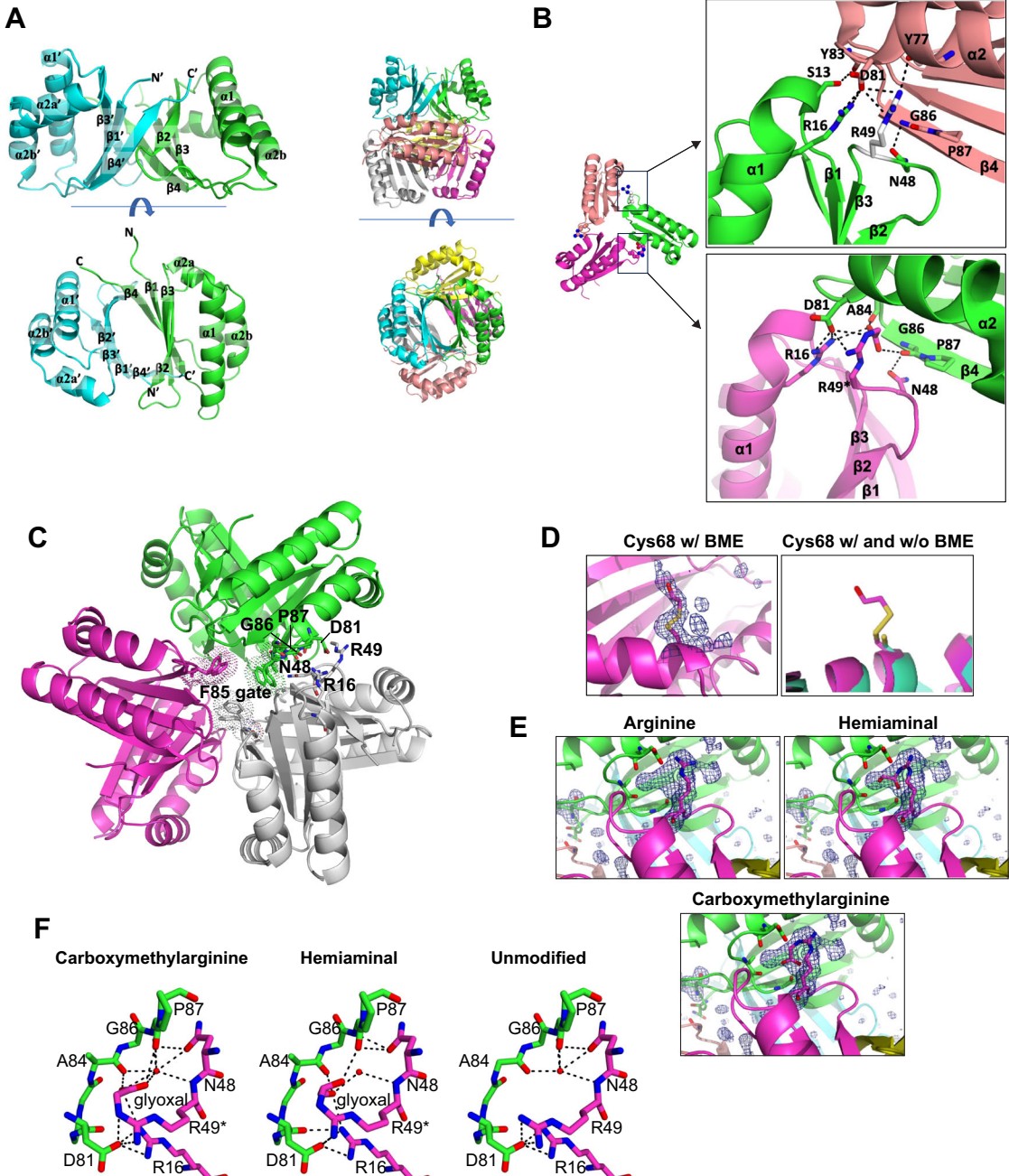

**Fig. 2 | ArqI forms a hexamer of three dimers. A** ArqI dimer shown in cartoon representation with green and cyan respective monomers (left panel), and hexamer (right panel) representation colored by subunit: A (green), B (cyan), C (magenta), D (yellow), E (salmon), and F (grey). The dimer (left panel) is labeled for secondary structure elements and shows chains A (green) and B (cyan). **B** A glyoxal modification was observed on Arg49 for three subunits (chains C, D and E) and a β-mercaptoethanol (βME) modification was observed on Cys68 for all 6 subunits. Shown here is the modeled carboxymethylarginine adduct modification on subunits C and E in the context of the trimer formed by the A, C, and E subunits. Arg49 is shown in sphere representation, where unmodified chain E-Arg49 is colored white (above). Two insets show the interactions at the A:C (chain C: modified Arg49) and A:E interface (chain A: unmodified Arg49) with Arg49 and interacting residues shown as sticks. Hydrogen bonds are shown as black dashed lines. **C** Cartoon representation of the ArqI hexamer. Dimers are colored magenta, grey, and green. Key residues are depicted as stick representations. Phe85 from three monomers are shown as sticks with dots representing electron density, which

together comprise the Phe85 "gate". **D** Left panel: Cys68 is modified by βME throughout the hexamer. Shown here is a Polder map for chain C-Cys68, electron density is shown in blue with sigma level of 4 and Cys68 is shown in a stick representation[137]. Right panel: Overlay of cartoon representations of the βME modified Cys68 structure (magenta) and unmodified structure (cyan). **E** Polder map for chain C-Arg49, electron density is shown in blue with a sigma level of 4 and Arg49 and the arginine adducts are shown in a stick representation, as are residues that make polar contacts with Arg49 or Arq49 adduct. Electron density for the ArqI structures indicate that the glyoxal modification on Arg49 is either carboxymethylarginine or hemiaminal adducts. Both modifications were modeled alongside an unmodified arginine residue at position 49. **F** Hydrogen bonds surrounding chain C-Arg49 for the carboxymethylarginine adduct, the hemiaminal adduct, and for unmodified Arg49 at the A:C interface. Residues with polar contacts at the interface are shown as sticks and hydrogen bonds are shown as black dashed lines. Source data are provided as a Source Data file.

dependent decrease in the unmodified Arg49 accompanied by a concomitant increase in the Arg49 GO-modified adduct (Supplementary Fig. 5E, F). Finally, we expressed FLAG-ArqI in *P. aeruginosa* MPAO1-cells and after an immunoprecipitation (IP) step using anti-FLAG beads (Supplementary Fig. 5G), subjected the samples to MS analysis. Supplementary Fig. 5H shows the Arg49 GO-derived adduct gave a Δ40 amu (with water loss), demonstrating that the modification exits in vivo.

### ArqI trimer interface residues facilitate mobilization to the flagellar pole

The atomic structure of ArqI indicated that Arg49 and other inter-trimer residues Arg16 and Pro87 might impact ArqI function. To investigate this, we first expressed and purified ArqI wild-type (WT) protein and R16A, R49Q or P87G variants, and subjected these proteins to size exclusion chromatography (SEC). The resulting chromatogram for the R49Q mutant showed a single peak that mostly paralleled the hexameric WT sample, but also showed a minor dimer peak. In contrast, both P87G and R16A mutants favored a mostly dimeric state (Fig. 3A and Supplementary Fig. 6A). Thus, both Arg16 and Pro87 are critical for hexamer formation via inter-trimeric contacts at physiological salt conditions in vitro (150 mM NaCl)[55]. A crystal structure of the R16A mutant (determined to 1.5 Å resolution; PDB ID 8ECP; Supplementary Table 1) still maintained a hexameric structure (likely due to preferred crystal packing). However, in the R16A structure, Arg49 showed either no electron density or multiple conformers resulting in less inter-trimer contacts (Supplementary Fig. 6B). Together, these data indicate that Arg16 and Pro87, and to a lesser degree Arg49, contribute to hexamer stability.

Next, we used fluorescence microscopy to examine ArqI cellular localization of WT, R49Q, P87G and R16A ArqI superfolder GFP (sfGFP) fusions using a controlled expression system. mCherry was used for constitutive expression to obtain accurate cell boundaries, enabling us to determine ArqI-sfGFP cell localization. As a control we also expressed another ABM domain from *P. aeruginosa*, PA3390, which shares a 33% identity with ArqI, but has alternate residues at Arg49, Pro87 and Arg16 positions (Supplementary Fig. 6C). Following arabinose induction, ArqI migration to the flagellar pole was observed, culminating in distinct puncta with wild-type ArqI, which was not observed in the PA3390 control (Fig. 3B, C). The R49Q mutant demonstrated less polar migration but still formed some distinct puncta at 8 h post-arabinose induction, whereas R16A closely resembled the control where no puncta were observed (Fig. 3B, C). Strikingly, the P87G mutant migrated to both poles (Fig. 3B). After we performed a more unbiased assessment of polar localization over time that incorporates the entire population of cells rather than selection by eye (see Methods), it became evident that the R49Q mutant was indeed impaired in its ability to localize to the pole in comparison to wild-type ArqI (Fig. 3C). A similar polar migration was also observed with GO induction driving a copy of ArqI-mScarlet-I under control of its native (*arqI-gloA2*) promoter (Fig. 3D). Supplementary Figs. 6D and 6E show that protein expression and growth in rich broth, respectively, were similar between the WT and mutant strains, ruling out misfolding artifacts of ArqI variants or the PA3390 ABM domain in vivo.

Finally, we used the **F**luorescence **R**ecovery **A**fter **P**hotobleaching method (FRAP) to study the quaternary structure of ArqI in live *P. aeruginosa* cells. WT ArqI diffused less rapidly than the mutant alleles, consistent with the WT ArqI having a larger oligomeric state (Fig. 3E; SI). When we take into consideration our in vitro assessment of the ArqI oligomeric state (Fig. 3A), these results support the hypothesis that ArqI polar migration requires hexamer formation through key inter-trimer residues. Importantly, these data demonstrate that the sfGFP fusion did not affect hexamer formation in vivo, and therefore gave confidence that other, far smaller epitope tags we have used in

this work (e.g., FLAG, HA and His tags), would likely not influence ArqI oligomeric formation nor function.

### ArqI expression suppresses Pseudomonas Quinolone Signal (PQS) biosynthesis

As ArqI seemed to have little influence on GO resistance (Fig. 1C, D) we hypothesized it could play a role in an unidentified signaling process initiated by GO induction. We noticed that after overnight (arabinose-induced) induction of ArqI, cultures were void of the normal green coloration; a phenomenon that results from high levels of the toxin pyocyanin and siderophore pyoverdine (Fig. 4A)[56,57]. Organic extraction of both pyocyanin and pyoverdine indeed showed a substantial decrease in concentration when ArqI was expressed compared to controls (Fig. 4B, C). Since both pyocyanin and pyoverdine levels are known to be controlled by PQS, a critical quorum-sensing molecule that controls iron acquisition and virulence in *P. aeruginosa*[58,59], PQS was extracted from cultures grown under the same conditions and assayed by thin layer chromatography. PQS was undetectable only when ArqI was expressed (Fig. 4D), suggesting that ArqI is a strong repressor of cellular PQS levels. Further, a *P. aeruginosa* PQS reporter system we constructed (Fig. 4E) showed that expression of ArqI resulted in a significant decrease in PQS, but not with expression of the PA3390 negative control ABM domain protein (Fig. 4F). Similarly, a potent repression of cellular PQS concentrations was observed when we expressed R49Q, P87G, and R16A mutants (Fig. 4F), indicating that ArqI mediated PQS repression was independent of its oligomeric state.

### ArqI directly interacts with PqsA; the first enzyme in PQS biosynthesis

We wanted to understand the mechanism behind PQS biosynthesis inhibition by ArqI. As a first step we performed a yeast-two-hybrid (Y2H) screen against a random fragmented library derived from the PAO1 genome (Hybrigenics Services, Courcouronnes, France). This screen produced a potential interacting clone of PqsA (residues 164-413); the first enzyme in the PQS biosynthesis pathway and responsible for catalyzing the conversion of anthraniloyl-AMP to anthraniloyl-CoA (Fig. 4G)[60].

To verify the ArqI-PqsA interaction we first employed intrinsic tryptophan fluorescence quenching. Unlike PqsA, ArqI lacks tryptophan residues and therefore would have no tryptophan emission. Thus, we could titrate ArqI into purified PqsA without fear of contaminating the PqsA tryptophan emission[61]. After assessing the correct excitation and emission wavelengths to produce low background from other aromatics (Supplementary Fig. 7A–D), increasing concentrations of purified WT ArqI protein was titrated into a fixed amount of either full-length PqsA protein (PqsA$_{FL}$), or the N-terminal domain of PqsA (PqsA$_{NTD}$) - the domain postulated by our Y2H results to bind ArqI (Fig. 4G). Both PqsA constructs showed a dose-dependent decrease (quenching) of intrinsic tryptophan fluorescence emission as measured at 340 nm with the addition of ArqI (Supplementary Fig. 7E, F). A $K_d$ of 21 μM for the ArqI-PqsA$_{NTD}$ interaction and a $K_d$ of 43 μM for the ArqI-PqsA$_{FL}$ interaction was then calculated (Fig. 4H). Because background fluorescence using tryptophan quenching can sometimes lead to artifactual results, we then labeled PqsA protein with dansyl, which further red-shifts the fluorescent signal of the protein of interest and eliminates background tryptophan fluorescence. Paralleling our results from intrinsic tryptophan quenching, addition of increasing amounts of ArqI to PqsA$_{NTD}$ resulted in an unambiguous, dose-dependent quenching of the dansyl signal (Fig. 4I, Supplementary Fig. 7G). An average ArqI-PqsA$_{NTD}$ $K_d$ was then calculated to be 1 μM (Fig. 4I), which we deemed to be more accurate than the tryptophan quenching due to the more prominent peak displayed from dansyl labeling (Supplementary Fig. 7G).

To confirm that PqsA interacts with ArqI in the *P. aeruginosa* cell, we developed a BioID protocol for *P. aeruginosa*, ideal for low affinity

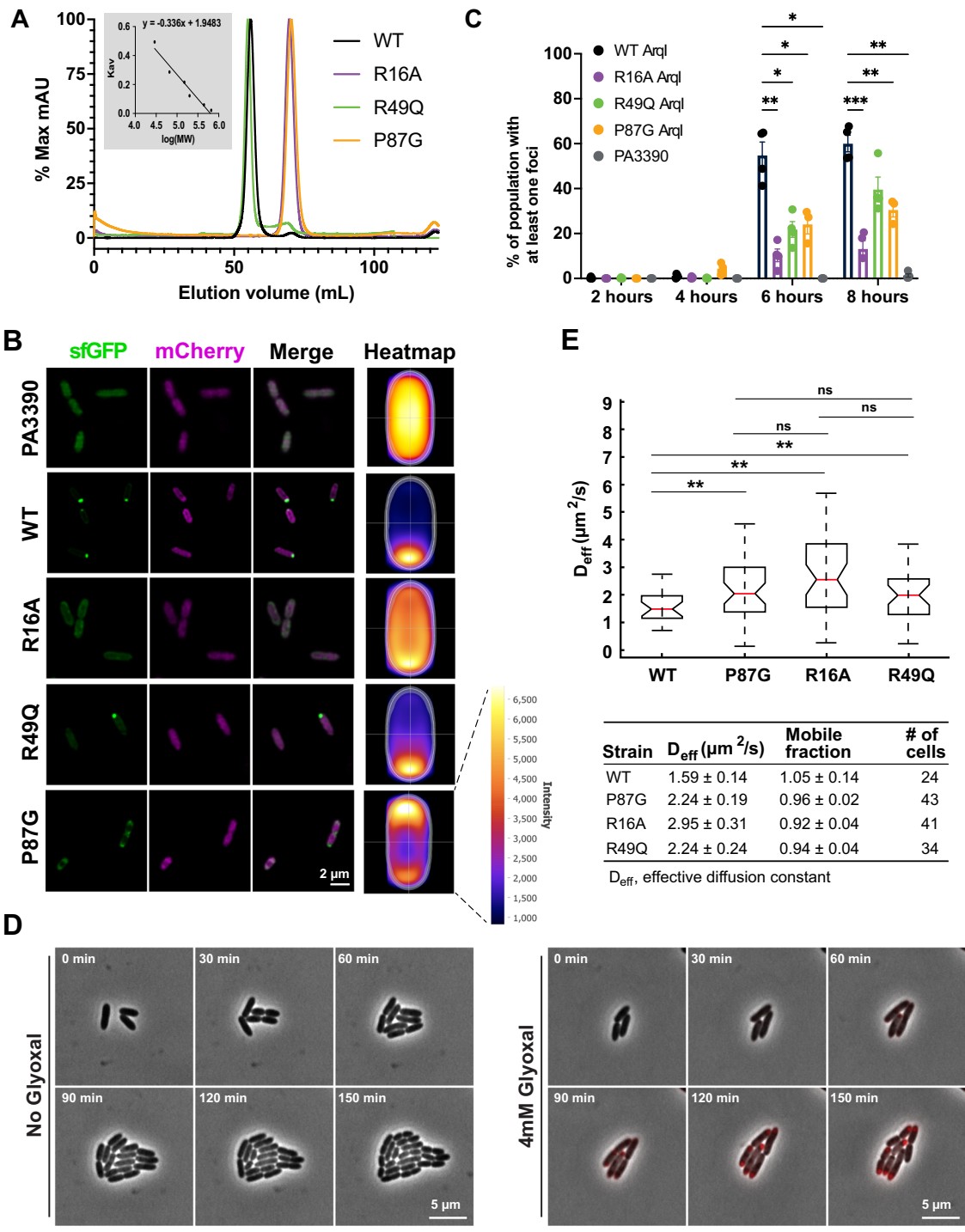

**Fig. 3 | ArqI exhibits a polar localization that is dependent on its hexameric quaternary structure. A** Size exclusion chromatography (SEC) of purified WT ArqI and point mutant variants showing in solution predominantly hexameric (WT and R49Q) and dimeric (R16A and P87G) quaternary structure. Inset: calibration curve of SEC column used for molecular weight estimation. **B** Super resolution fluorescence microscopy images and accompanying fluorescence intensity heat maps of sfGFP-tagged ArqI WT and ArqI point mutant variants eight hours post-induction. The PA3390 ABM domain was used as a negative control for (lack of) polar localization. mCherry was constitutively expressed to visualize the entire cell. Fluorescence intensity heatmaps of the sfGFP channel were generated using MicrobeJ[104]. $n = 4$ biological replicates for all groups except for P87G ArqI ($n = 3$ biological replicates). **C** Visual quantification of polar localization in the experiment in *B* using MicrobeJ (expressed as the counted percentage with at least one foci normalized to total counted cells). A minimum of 500 bacteria total were analyzed per strain and

per timepoint. $n = 4$ biological replicates for all groups except for P87G ArqI ($n = 3$ biological replicates). **D** Time-lapse fluorescence microscopy of mScarlet-I-tagged ArqI polar localization upon induction with GO under the control of its native promoter. Cells were spotted onto LB 1.5% agarose pads with or without 4 mM GO and imaged every 30 min for 2.5 h. Representative images shown of 3 independent experiments. **E** FRAP analysis of WT and ArqI variants showing changes in in vivo effective diffusion constants ($D_{eff}$). $n = 3$ biological replicates. For panels *C* and *E*: Bars indicate arithmetic mean. Whiskers denote standard deviation. *$P < 0.05$; **$P < 0.01$; *** $P < 0.001$ by 2-way ANOVA and Tukey's multiple comparisons test (*C*) or two-sided *t*-test pair wise comparison and Bonferroni correction (*E*). Significant $P$ values were calculated for *3c* (left to right) as (6 h) 0.0088, 0.0252, 0.0102, 0.0326 (8 h) 0.0006, 0.0094, 0.0016. Significant $P$ values for *3e* (left to right): 0.0027, 0.0002, 0.0085. Source data are provided as a Source Data file.

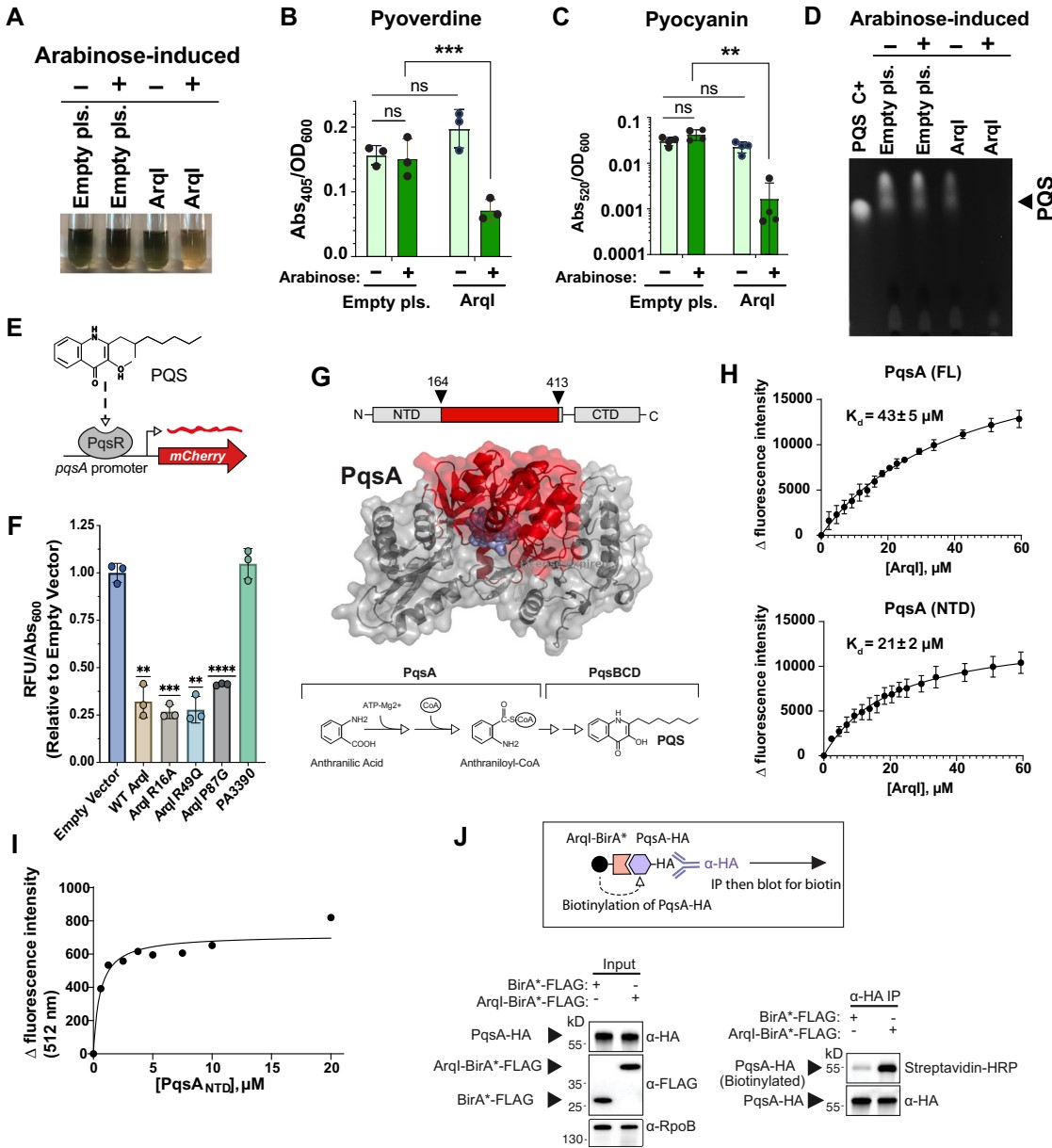

**Fig. 4 | ArqI inhibits PQS production and directly interacts with PqsA. A** Cell-free supernatants of 24-h cultures of WT MPAO1 harboring either pSB109 empty plasmid control or pSB109-FLAG-ArqI. AqrI expression was induced (+) or not induced (−) with arabinose. **B** Quantification of pyoverdine and **C** pyocyanin in supernatants of three biological replicates of experiment shown in *A*. $n = 4$ biological replicates. Bars indicate arithmetic mean. Whiskers denote standard deviation. **D** Thin-layer chromatography (TLC) analysis of extracted PQS from 24-h cultures. Purified PQS (C+) was used as a positive control and standard. **E** Schematic diagram of the $P_{pqsA}$-mCherry reporter construct used as a PQS biosensor (this work). **F** Effect of WT ArqI, ArqI variants, or PA3390 expression on $P_{pqsA}$-mCherry reporter activity at 24 h post ArqI induction. The y-axis was calculated as relative fluorescent units (RFU) to the empty vector control. $n = 3$ biological replicates. Bars indicate arithmetic mean. Whiskers denote standard deviation. **G** Above: Schematic diagram of PqsA fragment (residues 164-413) identified by Y2H as interacting with ArqI. Below: Cartoon representation of the full length PqsA structure generated with the deposited N-terminal domain (PDB code 5OE4;[67]) and AlphaFold[66]. The ArqI-interacting surface is highlighted in red and substrate shown as blue spheres. The reaction catalyzed by PqsA and ensuing PQS biosynthesis is shown below. **H** Intrinsic tryptophan quenching of ArqI interacting with PqsA in vitro. Excitation

(ex) wavelength set at 295 nm and emission (em) scanned at 320-400 nm range. Plot of changes in tryptophan fluorescence intensities (read at 330 nm) of PqsA full-length protein (FL; top panel) and PqsA N-terminal domain (NTD; bottom panel) titrated with increasing ArqI concentrations. Plots are mean values of 4 trials. $K_d$ was determined by fitting in binding saturation: One site – specific binding model of GraphPad Prism 10. Shown are the mean $K_d \pm$ SEM values. (**I**) One representative of three replicates of dansyl-labeled ArqI quenched by addition of purified PqsA$_{NTD}$. Fluorescence intensity changes were measured at 512 nm. Curve fitting of these data yielded an average dissociation constant ($K_d$) of $1.0 \pm 0.3$ μM ($\pm$ SEM). **J** Above: Schematic representation of the BioID experiment. Left panel: Western blot analysis of HA-tagged PqsA from *P. aeruginosa* MPAO1 cell lysate after co-expression of either ArqI-BirA*-FLAG or a BirA*-FLAG control. RpoB was used as a loading control. Right panel: Biotin blot analysis of anti-HA immunoprecipitated (IP) proteins from lysates in the left panel. Representative blots shown of three independent experiments. For panels *B, C,* and *F*: \*$P < 0.05$; \*\*$P < 0.01$; \*\*\*$P < 0.001$ by two-Way ANOVA with Šídák's multiple comparisons test (*B, C*) or one-sample *t* test (*F*). For *4b* the *p* value was calculated as 0.0003 and *4c* 0.0065. For *4f* (left to right) 0.0052, 0.0008, 0.0031, <0.0001. ArqI contains an N-terminal FLAG tag in *A-D, F*. PA3390 contains a C-terminal FLAG-tag in *F*. Source data are provided as a Source Data file.

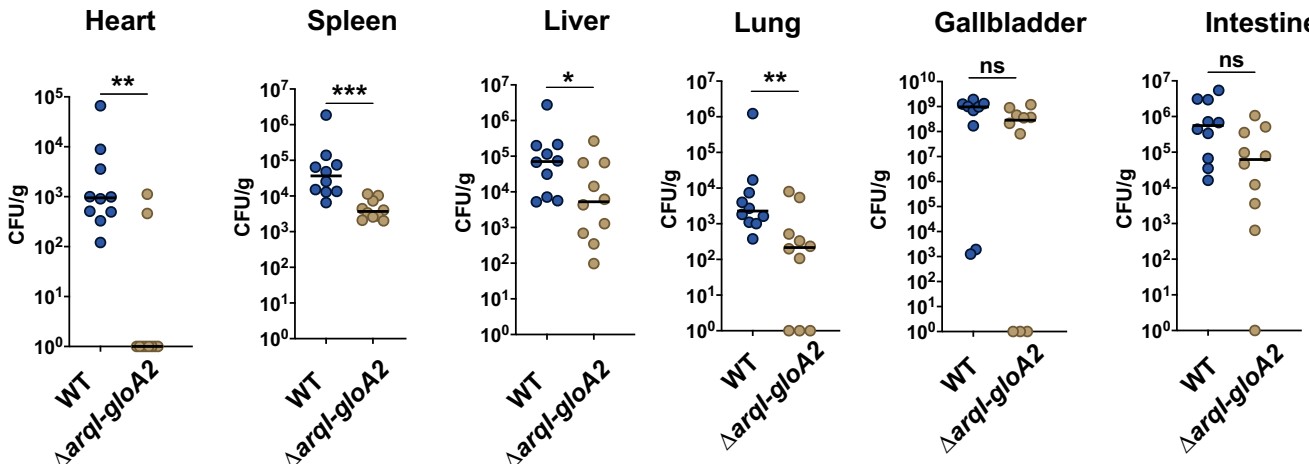

**Fig. 5 | Sepsis infections. The *arqI-gloA2* operon is required for robust colonization of the heart, lung, liver and spleen during bacteremia.** BALB/c mice (n = 10, 2 × 5 mice replicates) were infected by tail vein injection with -1 × 10⁶ CFU of the indicated strains. Organs were harvested and homogenized 24 hpi and bacteria enumerated from serial dilutions. Data points on the x-axis indicate that no CFU were recovered. Statistics: *P < 0.05; **P < 0.01; ***P < 0.001 by way of a two-sided Mann-Whitney test. P values for wild-type versus *arqI-gloA2* deletions are heart (0.0007), spleen (<0.0001), liver (0.0355), lung (0.0089), gallbladder (0.0941), intestine (0.5240). Source data are provided as a Source Data file.

or transient interactions often missed by more classic IP approaches[62]. BioID has been used extensively with mammalian systems but has thus far seen limited use in prokaryotes[63,64], and has never been used with *P. aeruginosa*. An ArqI-BirA*-FLAG fusion protein or a BirA*-FLAG control were co-expressed with HA-tagged PqsA protein in *P. aeruginosa*. After sufficient time to allow BirA*-mediated biotinylation, PqsA-HA was immunoprecipitated (IP) first to avoid *P. aeruginosa* background biotinylated proteins. This step was then followed by detection of biotinylation in the IP purified PqsA-HA (Fig. 4J, above). Results shown in Fig. 4J (below) revealed that only with ArqI-BirA* fusion expression was robust biotinylation of PqsA observed, providing supportive evidence that these two proteins interact in *P. aeruginosa*.

Finally, we examined if the direct interaction of ArqI with PqsA resulted in an inhibition of the PqsA anthranilate coenzyme A (CoA) ligase enzymatic activity. Purified WT ArqI protein was added in 10-molar excess of purified PqsA protein with CoA, ATP, and anthranilate. PqsA enzymatic product (anthraniloyl-CoA) was measured at 375 nm absorbance as previously described[65]. Data indicated that addition of ArqI did not affect anthraniloyl-CoA levels in the reaction (Supplementary Fig. 7H). Thus, the interaction of ArqI with PqsA may inhibit PQS synthesis either by blocking additional PqsA interactions, or by an unknown indirect mechanism.

### Structural prediction of ArqI-PqsA interaction

ArqI was predicted by Y2H to interact with PqsA residues 164-413 (Fig. 4G). In the absence of a PqsA-ArqI atomic structure, we exploited two different programs to predict this interaction; HDOCK[54] and AlphaFold3[66]. Dimeric or hexameric ArqI were assessed for their interactions with the deposited tetrameric PqsA structure[67]. Both programs produced predictions that were in good agreement with our Y2H interface. Interestingly, AlphaFold3 failed to model the ArqI hexamer. However, when dimeric ArqI was modeled with tetrameric PqsA, several different conformations of the complex were generated, one of which produced a structure with the ArqI dimer centered around the PqsA Y2H-predicted interacting interface (in yellow; Supplementary Fig. 8A, B). When the hexameric ArqI was docked onto the tetrameric PqsA in HDOCK, it produced a structure where the hexamer also interacted at the predicted Y2H interface (yellow; Supplementary Fig. 8C, D), and shared several of the interacting residues seen with the dimeric ArqI-PqsA predicted structure. Importantly, dimeric ArqI

binding to PqsA is in agreement with our reporter data in Fig. 4F, which shows that both WT hexamer and dimeric mutants of ArqI are all potent inhibitors of PQS biosynthesis.

### CoA ligase-ABM and glyoxylase-ABM single polypeptide proteins exist and are conserved in nature

Because ArqI was co-expressed in an operon with a glyoxalase (*gloA2*) and was shown to physically interact with the CoA ligase PqsA, we wanted to explore if ABM-CoA ligases and ABM-glyoxalases existed in nature as single proteins. Such a finding would support the hypothesis that these protein families maintain functional conservation.

The Pfam database[68] contains 315 different domain architectures that possess at least one ABM domain (Pfam identifier PF03992). The PqsA enzyme more broadly falls into the acyl-CoA ligase family (Pfam identifier PF00501). Results from an InterPro domain search revealed 5 domain architectures and 43 proteins within these defined architecture groups that contained both an ABM domain and AMP-binding domain (acyl-Co ligase) (listed in **SI**). Conversely, the Pfam identifier PF00549 encompassing the CoA-ligase family showed no associations with ABM domains. A search for glyoxalase domains (Pfam identifier PF00903) with an ABM domain resulted in 3 domain architectures and 24 proteins (listed in **SI**). Taken together, these findings suggest that glyoxalases and PqsA related ligases have indeed co-evolved with ABM domains to presumably carry out conserved biological functions yet to be determined.

### The *arqI-gloA2* operon is required for *P. aeruginosa* sepsis

Because GO-exposure signals a host response or concerted change in gene expression (*e.g.* NO and ROS detoxification genes, T2SS associated genes (Vre (PUMA3) / Hxc gene clusters), and the master small RNA virulence regulator *sicX* required for acute infection[31]) and recent literature suggests that aldehyde remediation plays a key role during infection in both bacterial and fungal pathogens[11–15,69], we wanted to know if the ArqI-GloA2 axis was involved in host survival. To address this issue, we used an acute mouse sepsis infection model (BALB/c mice) with WT MPAO1 and our *arqI-gloA2* operon deletion mutant. Results 24 h after tail vein infection revealed that the Δ*arqI-gloA2* mutant exhibited a statistically-significant drop in bacterial CFUs in the heart, spleen, lung and liver (Fig. 5), which is largely reflective of existing blood titers[70]. Conversely, CFU recovery from the gallbladder

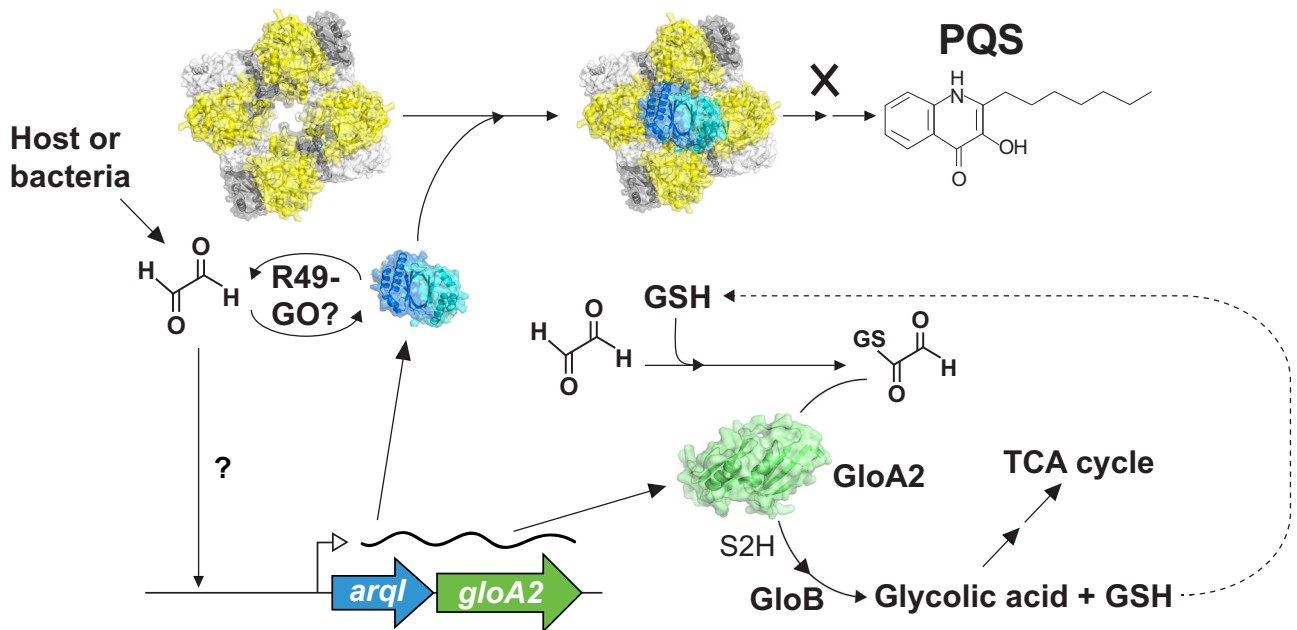

**Fig. 6 | Schematic diagram of ArqI-GloA2-PQS axis hypothesis.** GO accumulates in the bacterial cell from either intrinsic metabolic production form the bacteria, or comes from the host cell. ArqI is posttranslationally-modified on Arg49 by GO, the biological consequence of which remains unknown. GO induces transcription of the *arqI-gloA2* operon through an as-of-yet unidentified GO-specific transcription factor. ArqI protein can then interact with PqsA as either a hexamer or dimer, inhibiting PQS biosynthesis and effectively blocking iron uptake, biofilm formation and immune modulation (PQS known functions). GSH reacts with GO to yield an intermediate which is then acted on by GloA2 to make S-2-hydroxyethylglutathione (S2H). GloB then converts S2H into glycolic acid, while liberating GSH to be recycled back into the GO remediation process. Glycolic acid can then eventually be converted to pyruvate and enter the TCA cycle.

and intestine did not reveal any statistical differences. These results show that, for the first time, genes involved in the GO-specific response are required for infection in a pathogen.

## Discussion

Although all life possesses robust GO response and remediation pathways, it remains unclear in a natural setting how millimolar quantities of GO and other aldehydes might accumulate as metabolic side products. Recently the idea of host cells producing these naturally occurring toxic compounds to eliminate pathogens has been gaining attention[11,14,71,72]. Here we provide the first comprehensive analysis of GO-specific toxicity responses in a microbe (summarized in Supplementary Fig. 9) and discovered the first GO-specific operon; *arqI-gloA2*, which protects the cell from GO-mediated toxicity and signals through direct interaction with PqsA – the first dedicated enzyme in the PQS biosynthesis pathway (Fig. 6). Importantly, in this work we show that the *arqI-gloA2* operon is required for survival in blood-rich organs, suggesting that GO-mediated signaling and remediation is important for host-pathogen interactions.

### ArqI posttranslational modifications and residue conservation

A surprising finding was the Arg49 (GO-derived) PTM in the crystal structure, the precise chemistry of which we hope to elucidate in future studies. Presently, we know that Arg49 is modified by the presence of GO in the bacterial cell and has modest effects on hexamer stability and the ability of ArqI to migrate to the flagellar pole (Fig. 3). A comprehensive study of all deposited ABM domains points to conservation of Arg49 in diverse microbial species, some Archaea, fungi and lower eukaryotes (Supplementary Fig. 10). Whether Arg49 might play a conserved functional role remains to be elucidated.

Although ABM domains were initially discovered (and named) for their roles in antibiotic biosynthesis[43], they have since been shown to play more diverse roles in cellular metabolism with the discovery of their involvement in the enzymatic degradation of heme[52,53]. Our previous work with ArqI (PA0709) demonstrated a role in heme regulation

and heme binding, however its structure published here clearly shows the canonical ArqI binding pocket is too small to accommodate such a large molecule, and therefore we must consider that the direct heme binding we observed in our previous work might not translate to in vivo roles.

In the absence of an obvious ligand, we examined the LsrG family of ABM domains—the closest ArqI structural homologs—whose roles include modulating cell-cell communication of the AI-2 signal by facilitating is catabolism[49]. LsrG family proteins are also postulated to have alternative roles in the degradation of diverse aromatic compounds or may function in redox cycles related to quinone metabolism. At present we can only speculate as to the precise ligand and presumptive GO-chemistry associated with ArqI. Interestingly, the two residues with a PTM in the ArqI structure (Cys68 and Arg49) are not conserved within the greater LsrG family, which is therefore suggestive of an alternative role for ArqI - despite the ability to model LsrG substrates into the ArqI presumed ligand binding pocket (Supplementary Fig. 3). This suggests that ArqI evolved from an LsrG ancestor to accommodate an alternative ligand and functional role in GO signaling. It remains to be seen how expansive the functional roles ABM domains play in nature, as the larger, parent family of ABM binding domains is understudied compared to that of, for example, the Per-Arnt-Sim (PAS) domain[73].

### ArqI as an inhibitor of PQS biosynthesis

Here we show that the polar migration of ArqI is dependent on its hexamer oligomeric state (facilitated by Arg16, *cis*-Pro87, and to a certain extent Arg49; Fig. 3), but not its inhibition of PQS accumulation (Fig. 4F). Although we do not yet understand the mechanism of ArqI GO-induced polar migration, we have modeled both hexamer and dimer interactions with PqsA to show similar regions of interaction (Supplementary Fig. 8), thus both quaternary structures appear to be capable of preventing PqsA function. On the other hand, how ArqI mediates cellular PQS levels remains enigmatic as ArqI does not inhibit PqsA enzymatic activity when tested in vitro (Supplementary Fig. 7).

Alternative mechanisms by which ArqI might inhibit PqsA function would include the physical disruption of other PqsA protein/protein interactions. Indeed, PqsB and PqsC require one another for solubility and form a stable complex[74], however whether PqsA interacts with PqsB and/or PqsC to form a larger complex is not known. Another possible mechanism that ArqI might use to control PQS levels is through the direct binding and degradation of PQS (or a PQS precursor) itself. Alternatively, such a molecule could act as an allosteric regulator of ArqI activity, along with GO, to influence its interaction with PqsA and PqsA inhibition within the bacterial cell. We also must consider that inhibition of PQS biosynthesis at the early stage of PqsA involvement would also result in the concomitant downstream inhibition of other related 4-hydroxy-2-alkylquinoline (HAQ) molecules that range from other quorum sensing compounds such as 2′ amino-acetophenone (2-AA), to ones that directly inhibit the host respiratory chain such as 2-heptyl-4-hydroxyquinoline N-oxide (HQNO)[75–77].

The molecular mechanism of GO-induced ArqI and its subsequent interaction with PqsA is confounded by the larger question of when and to what extent PQS is produced in vivo[78]. One intriguing possibility is that ArqI affects PQS production in certain tissues in vivo to regulate biofilm formation and the state of colonization versus the planktonic phenotype, whereas GloA2 is simply required for remediation of host cell produced GO. Ultimately, the spatiotemporal ligand-induced regulation of both PQS and ArqI-GloA2 axis induction will have to be examined during infection, all of which presently remains unmapped.

## Perspectives on the ArqI-GloA2 axis in the context of infection

In this work we have shown that, when produced, ArqI is able to inhibit PQS production in the *P. aeruginosa* cell (Fig. 4). On the other hand, a spatiotemporal map of how/if this interaction translates into regulation of PQS biosynthesis by the ArqI-GoA2 axis during infection, and importantly the natural inducer(s) of the GO response, remain enigmatic. Although we do not show direct evidence of the ArqI-GloA2-PqsA axis involvement in blunting phagocytic assaults, this work and the literature provide sufficient evidence to seriously explore this as a possibility. In support, a recent report has eloquently described macrophages exploiting MGO as an antimicrobial to control a broad-range infections[79]. In addition, PQS can act as a chemoattractant for neutrophils in culture[80]. It would thus follow that preventing PQS production in response to host aldehyde signals in certain tissues would be in the best interest of *P. aeruginosa* to survive host assaults. In addition, phagocytes rely on ROS and RNS (*e.g.* NO) to eliminate bacteria, a process that is exacerbated by the presence of iron. The reaction of iron and ROS can then result in oxidation of biomolecules to produce GO. Indeed, our data show that GO exposure results in a dramatic repression of siderophore, heme and free iron uptake systems, processes that PQS also positively controls (Supplementary Fig. 9)[78].

As mentioned above, an especially novel and interesting result from our RNA-seq data showed a marked increase in three taurine α-ketoglutarate (α-KG) dioxygenases, responsible for catalyzing the conversion of the amino acid taurine to sulfite and aminoacetaldehyde in the presence of oxygen, α-KG and iron[21,22]. Interestingly taurine, which is only synthesized by the host/vertebrate cells, has been implicated as a host signal necessary for colonization of gut microbiota[81]. Moreover, taurine is the most abundant amino acid in leukocytes such as neutrophils where it can reach up to a staggering 50 millimolar in concentration[82]. In leukocytes taurine readily reacts with hypochlorous acid to form taurine-HCl, which directly inhibits nitric oxide (NO) synthase (iNOS) to curtail phagocyte NO poisoning[83]. Our RNA-seq data here suggest GO as a potential signal that plays a key role taurine uptake from host cells, which could have several functions including a potential use in blunting GO and/or acting as a nutrient source.

Recently the idea that aldehydes generated by phagocytes can act as antimicrobials has been gaining traction; however, there is a paucity of examples that directly demonstrate such a connection. These include studies implicating MGO depletion of GSH affecting glyoxalase activity and the intracellular survival *Listeria monocytogenes*, as well as elimination of a glyoxalase from *Streptococcus pyogenes* resulting in increased neutrophil killing[11,72]. Other work has highlighted macrophages using aldehydes as an innate immune defense mechanism[14,71]. When combined, these studies and our work here indeed lend credence to the argument that bacteria are equipped to sense the presence of these aldehydes and mount their own defense by shutting down the uptake of phosphate and iron (through ArqI inactivation of PQS biosynthesis), utilizing the host's intracellular taurine and other sulfur sources to replenish sulfur pools and then capitalize on these acquisitions to detoxify the host aldehyde defense mechanism. On the integrated role of PQS in these processes, although PQS regulates activities that are crucial for infection (*e.g.* iron acquisition), its precise role and spatiotemporal expression during infection remains somewhat controversial[78]. Thus, future studies will have to interrogate the connection between PQS and the ArqI-GloA2 axis during infection.

Our infection data in Fig. 5 suggests the roles of GO-induced genes might be tissue, or even cell type specific given the importance of the *arq-gloA2* operon in the blood-rich heart, spleen, lung and liver, but to a much lesser extent the gallbladder and intestine. Given what we know about *P. aeruginosa* and sepsis this would further support the role of the ArqI-GloA2 axis in host phagocyte remediation. Indeed, recent data show there is a large bottleneck in the elimination of bacteria prior to arrival in the gallbladder[84]. However, once in the gallbladder, whatever bacterial cells were successful in colonizing this organ are then able to replicate to exceedingly high levels (in this work ~1 x 10$^9$; Fig. 5). These bacteria, in turn, seed the intestines, which, along with an expected reduction in immune system activity in these organs, might be the reason we observed no significant difference in the gallbladder and the intestines when comparing WT to the *arqI-gloA2* mutant.

Finally, the RNA-seq data implicated GO as an inducer of the *vre* (PUMA3) regulated T2SS *hxc* gene cluster, whose role in virulence has been established[40]. However other than its upregulation during phosphate starvation, the inducer of this T2SS has remained enigmatic (Supplementary Fig. 9)[39]. We did observe that the T2SS excreted hemolytic phospholipase PlcH– whose role is to liberate host phosphatidylcholine to diacylglycerol (DAG)[85], which is then presumably internalized and processed by the *glp* operon (also GO induced; Supplementary Fig. 9)[41,86,87]. Indeed, the Glp system was one of the most upregulated operons after GO exposure, but interestingly only at the later 1-h timepoint (Fig. 1A; Supplementary Fig. 1; SI), and has been implicated in control of infection due to its MGO byproduct[88]. Why the Glp pathway, and presumably MGO production, is upregulated in the presence of GO remains to be deciphered. These data warrant further examination as to how host aldehydes may trigger Hxc/PclH production and the glycerol uptake and catabolism system to exploit primary lung surfactant and epithelial cell-based lipids as a carbon source.

In sum, we have discovered the first GO-specific response operon in bacteria. Our work suggests that aldehydes such as GO, which have recently been proposed as a means by which host cells kill engulfed bacteria[11,14,71,72], act to induce a sulfur and phosphate starvation response, the latter of which induces T2SS Vre/Hxc virulence axis. We show that the presence of GO induces the expression of ArqI to then block PQS production, an action which would presumably prevent additional iron intake and by association iron-driven ROS and more GO formation (Supplementary Fig. 9). It will be of great future interest to determine if detection of host aldehydes signals pathogens through ArqI to initiate colonization and infection, and how this response might differ between GO and MGO exposures. Indeed, to facilitate such targeted aldehyde responses, there must be transcription factors

able to sense GO and other aldehydes with precision that remain to be discovered.

## Methods

### Ethical Considerations

Animal studies were approved by the Institutional Animal Care and Use Committee (IACUC) at Loyola University Chicago (Protocol 2021016). Animal procedures were performed according to the Institutional Animal Care and Use Committee Guidebook of the Office of Laboratory Animal Welfare and Public Health Service Policy on Humane Care and Use of Laboratory Animals. Animal studies were carried out in compliance with the Animal Research: Reporting of In Vivo Experiments (ARRIVE) guidelines (https://arriveguidelines.org). Animals were humanely euthanized following guidelines approved by the American Veterinary Medical Association (AVMA).

### Statistics and Reproducibility

Animal numbers and statistical analyses were calculated according to ref. [89]. Only female mice were used in the study, and sex was not considered in the current study design. Prior work using this infection model has demonstrated no sex-dependent differences in phenotypes[89]. For general statistical analyses and reproducibility, all experiments were done at least in triplicate unless otherwise noted. When appropriate, data were analyzed using Prism (GraphPad Software). We use a one-way ANOVA test with a single variable, and with more than one variable a two-way ANOVA test. For statistical analyses between two samples, we use a student's $t$-test pairwise comparison. Specific statistical tests used, the number of subjects tested in each experiment and notations of significance are described in the figure legends or within the individual figure panel. A $P$ value of 0.05 or less was considered significant for all analyses. No data were excluded from the analyses. The experiments were not randomized. The Investigators were not blinded to allocation during experiments and outcome assessment.

### Bacterial strains and growth conditions

Bacterial strains used in this study can be found in **SI**. *P. aeruginosa* and *E. coli* cultures were grown in Luria-Bertani media (LB, Difco; Beckton Dickinson, Franklin Lakes, NJ) for standard growth and maintenance. Liquid cultures were grown at 37 °C with 230 rpm shaking on an orbital shaker unless otherwise stated. Culture density was monitored using a Genesys 150 UV-Vis spectrophotometer (Thermo Fisher Scientific, Waltham, MA) at a wavelength of 600 nm. Solid media was solidified with 1.5% agar (bacteriological; VWR, Solon, OH). **S**uper **O**ptimal broth with **C**atabolite repression (SOC) medium was made with 20 g/L tryptone, 5 g/L yeast extract, 0.5 g/L NaCl, 10 mM $MgCl_2$, 10 mM $MgSO_4$, 2.5 mM KCl, and 10 mM glucose (final concentrations). M9 minimal media (6.78 g/L $Na_2HPO_4$, 3 g/L $KH_2PO_4$, 0.5 g/L NaCl 1 g/L $NH_4Cl$, pH 7.4) was supplemented with 0.2% glycerol 100 μM $CaCl_2$, 100 μM $FeCl_3$, and 2 mM $MgSO_4$ after autoclaving. **P**seudomonas **I**solation **A**gar (PIA; Sigma Aldrich, St. Louis, MO) was supplemented with 20 mL/L glycerol for selection of *P. aeruginosa* exconjugants. Sucrose containing plates were made by the addition of 5% sucrose to LB agar after autoclaving. Antibiotic concentrations used in selection of *E. coli* resistance determinants were as follows: 15 μg/mL Gentamicin (Gm), 50 μg/mL carbenicillin (Cb), 50 μg/mL kanamycin (Kan), and 50 μg/mL streptomycin (Sm). Antibiotic concentrations for *P. aeruginosa* were as follows: 30 μg/mL Gm, 250 μg/mL Cb, and 2,000 μg/mL Sm. For inducible constructs, isopropyl-ß-D-1-thiogalactopyranoside (IPTG; GoldBio, St. Louis, MO) was added to both in liquid culture and agar plates at a concentration of 0.1-1 mM, and arabinose to a concentration of 0.02-0.2% w/v.

### Construction of plasmids

All restriction enzymes were purchased from New England Biolabs (NEB), Ipswitch, MA. To construct a low-copy replicating plasmid for reporter constructs and complementation experiments, pSB109[47] was chosen as a suitable backbone. To remove all elements other than the origin of replication and the gentamicin-resistance gene, the arabinose-inducible promoter and the original multiple cloning site (MCS) of pSB109 were removed by digestion with SphI and SacI followed by gel extraction of the origin-containing DNA fragment. T4 DNA polymerase (NEB) was then used to blunt the DNA fragment. A linear DNA cassette synthesized by Integrated DNA Technologies (IDT; Coralville, IA) containing the MCS from mini-CTX[90], as well as a 5′ sequence complementary to the pSB109 sequencing forward primer (**SI**) was then blunt-ligated into the vector using T4 DNA Ligase (Promega, Madison, WI) to yield plasmid pCC21.

Construction of inserts for ligation was done using the melt and reanneal method as per ref. [91] unless otherwise noted. To construct the pE-SUMO-ArqI plasmid, the ArqI ORF was PCR amplified from *P. aeruginosa* MPAO1 genomic DNA using two primer pairs to generate BsaI complementary overhangs: BsaI-ArqI-F1/BsaI-ArqI-R2 and BsaI-ArqI-F2/BsaI-ArqI-R1. The PCR products were ligated into BsaI digested pE-SUMO (LifeSensors, Malvern, PA).

To construct pCPD-PqsA, the coding sequence of the *pqsA* gene was amplified from *P. aeruginosa* MPAO1 using primer pairs pCPD-PqsA-F/pCPD-PqsA-R and inserted into SmaI digested pCPD[92] using Gibson Assembly (NEB) according to the manufacturer's protocols[93]. The $PqsA_{NTD}$ (amino acids 1-399) construct was ligated into the pQLinkG2 vector according to a published protocol[94].

To construct the $P_{arqI}$ reporter pCC21-$P_{arqI}$-mScarlet-I, the 145 bp upstream regulatory region of ArqI was PCR amplified using primer pairs SacI-$P_{arqI}$-F and XbaI-NcoI-$P_{arqI}$-R (**SI**), followed by digestion with SacI and XbaI (New England Biolabs) to produce complementary overhangs. The product was then ligated into pCC21 digested with SacI and XbaI to create pCC21-$P_{arqI}$. An mScarlet-I gene block codon optimized for *Streptococcus pneumoniae* was synthesized by IDT and used as a PCR template and amplified with primer pairs NcoI-mScarlet-I-F1/KpnI-mScarlet-I-R1 and NcoI-mScarlet-I-F2/KpnI-mScarlet-I-R2. The resulting product was then ligated into NcoI/KpnI digested pCC21-$P_{arqI}$ to yield pCC21-$P_{arqI}$-mScarlet-I.

The low-copy replicating vector pMMB67EH was modified to contain an enhanced ribosome-binding site and FLAG-tag sequence for easy tagging of proteins for expression. A linear DNA fragment synthesized by IDT, SacI-pSB109-RBS-FLAG-HindIII, containing the enhanced RBS from pSB109, as well as a 3′ FLAG-tag sequence was digested with SacI and HindII and ligated into SacI/HindIII digested pMMB67EH to yield pMMB67EH-FLAG. The PqsA coding sequence, omitting the native stop codon, was then PCR amplified from the MPAO1 chromosome using primer pairs SphI-PqsA-F1/PstI-PqsA-R1 and SphI-PqsA-F2/PstI-PqsA-R2. The PCR product was then melted, reannealed, and then ligated into SphI/BamHI digested pMMB67EH-FLAG to yield pMMB67EH-PqsA-HA.

For *arqI-gloA2* single genes and whole operon complements, the arabinose-inducible plasmid pSB109 was utilized. To construct pSB109-FLAG-ArqI, the *arqI* gene was PCR amplified from MPAO1 using primer pairs NcoI-FLAG-ArqI-F1/NdeI-ArqI-R2 and NcoI-FLAG-ArqI-F2/NdeI-ArqI-R1. The resulting products were ligated into NcoI/NdeI digested pSB109. To construct pSB109-gloA2-HA, the *gloA2* gene without its native stop codon was amplified from MPAO1 using primer pairs NcoI-gloA2-F2/PstI-gloA2-R2 and NcoI-gloA2-F1/PstI-gloA2-R1. The resulting product was ligated into NcoI/PstI digested pSB109, inserting the gene in frame with the included HA-tag and stop codon in pSB109. To construct pSB109-ArqI-FLAG-gloA2-HA, the *arqI* gene and upstream regulatory region was PCR amplified from MPAO1 using primers BamHI-145bp-UP-ArqI-F and ArqI-FLAG-Link-R. The *gloA2* gene and upstream intergenic region was PCR amplified with primers ArqI-FLAG-Link-F and HindIII-gloA2-HA-R. The PCR products were then fused using overlap extension PCR and the product used as a PCR template with primers BamHI-145bp-UP-ArqI-F2 and HindIII-gloA2-HA-

R. This product was then digested with BamHI and HindIII and ligated into BamHI/HindIII digested pUC18T-mini-Tn7T-Gm[95]. The *arqI-gloA2* genes were then amplified from this plasmid using primer pairs NcoI-ArqI-F1/PstI-gloA2-R2 and NcoI-ArqI-F2/PstI-gloA2-R1 and ligated into NcoI/PstI digested pSB109. pSB109-PA3390-FLAG was constructed similarly to pSB109-FLAG-ArqI with primer pairs NcoI-FLAG-PA3390-F1/NdeI-PA3390-R2 and NcoI-FLAG-PA3390-F2/NdeI-PA3390-R1.

To construct pSB109-ArqI-sfGFP for protein localization experiments, the *arqI* gene and the 145 bp upstream regulatory region was amplified from the MPAO1 chromosome with primer pair BamHI-145bp-ArqI-F/Linker-ArqI-R. Super folder GFP (sfGFP) with an N-terminal domain-breaking linker was amplified from the construct described in ref. 96 with primer pairs sfGFP-Linker-F/HindIII-sfGFP-R. The DNA pieces were then fused using overlap extension PCR and the product gel extracted. The product was then used as a template for PCR amplification with primer pairs pair BamHI-145bp-ArqI-F/HindIII-sfGFP-R, digested, and subsequently ligated into BamHI/HindIII digested pCC21 to yield pCC21-P$_{arqI-145bp}$-ArqI-sfGFP. The ArqI-sfGFP fusion construct was then amplified from pCC21-145bp-ArqI-sfGFP using primer pairs NcoI-ArqI-F1/NdeI-sfGFP-R2 and NcoI-ArqI-F2/NdeI-sfGFP-R1 and subsequently ligated into NcoI/NdeI digested pSB109 to yield pSB109-ArqI-sfGFP. pSB109-PA3390-sfGFP was constructed similarly using primers listed in **SI**. pCC21-P$_{arqI}$-ArqI-mScarlet-I was constructed in a similar method using primer pairs BamHI-145bp-ArqI-F2/Linker-ArqI-R, mScarlet-I-Linker-F/HindIII-mScarlet-I-R2, BamHI-145bp-ArqI-F1/HindIII-mScarlet-I-R2, and BamHI-145bp-ArqI-F2/HindIII-mScarlet-I-R1.

To construct the pSB109-ArqI-BirA*-FLAG plasmid, the *arqI* gene was amplified using primer pair NcoI-ArqI-F1/BirA-Link-ArqI-R. The *birA** gene with a 5′ flexible linker (GGGGSGGGGSGGGGS) was amplified from an *S. pneumoniae* codon-optimized gene fragment of the *Aquifex aeolicus* VF5 BirA R40G catalytic mutant, synthesized by Genscript (Piscataway, NJ), with primer pairs ArqI-Linker-BirA-F/NdeI-BirA-R1. The two resulting PCR products were fused using overlap extension PCR, and the resulting gel extracted fragment was used as a PCR template with primer pairs NcoI-ArqI-F1/NdeI-BirA-R2 and NcoI-ArqI-F2/NdeI-BirA-R1. The PCR product was then ligated into NcoI/NdeI digested pSB109. To construct pSB109-BirA*-FLAG, the *birA** gene was amplified using primer pairs NcoI-BirA-F1/NdeI-BirA-R2 and NcoI-BirA-F2/NdeI-BirA-R1. The PCR product was then ligated into NcoI/NdeI digested pSB109.

Site-directed mutagenesis of plasmids was carried out by a modified Quick Change method[97]. Briefly, plasmids were used as a PCR template with Platinum SuperFi II polymerase (Thermo Scientific) with mutagenic primers listed in **SI**. PCR reactions were run for 25 cycles before DpnI digestion and transformation into *E. coli* DH5α cells. All mutations were verified by Sanger sequencing at GeneWiz (Azenta Life Sciences, Burlington, MA) or whole plasmid next-generation sequencing (Plasmidsaurus, Eugene, OR).

To construct mini-CTX1-Gm-P$_{A1/04/03}$-mCherry and mini-CTX1-Gm-P$_{pqsA}$-mCherry, a linear DNA fragment containing the mCherry gene flanked by a 5′ HindIII cut site, the RBS from pSB109, a NcoI site that consists of the mCherry start codon, and a 3′ KpnI cut site was synthesized by IDT. The synthesized DNA fragment was digested with HindIII and KpnI and ligated into HindIII/KpnI digested mini-CTX1-Gm. The P$_{A1/04/03}$ promoter was synthesized by IDT as complementary ssDNA oligos that when annealed created 5′ BamHI and 3′ NcoI sticky ends[91]. The oligos were annealed and ligated into BamHI/NcoI digested mini-CTX1-Gm-mCherry to yield mini-CTX1-Gm-P$_{A1/04/03}$-mCherry. For the P$_{pqsA}$ reporter construct, the regulatory region upstream of the *pqsA* gene coding region was amplified using primer pairs BamHI-P$_{pqsA}$-F1/HindIII-P$_{pqsA}$-R2 and BamHI-P$_{pqsA}$-F1/HindIII-P$_{pqsA}$-R2 and the resulting product ligated into BamHI/NcoI digested mini-CTX1-Gm-mCherry to yield mini-CTX1-Gm-P$_{pqsA}$-mCherry. In addition, mini-

CTX1-Gm-P$_{A1/04/03}$-sfGFP was constructed by digesting mini-CTX1-Gm-P$_{A1/04/03}$-mCherry with NcoI/KpnI to remove the mCherry gene. The sfGFP gene was then amplified using primer pairs NcoI-sfGFP-F1/KpnI-sfGFP-R1 and NcoI-sfGFP-F2/KpnI-sfGFP-R2 and subsequently ligated into the digested plasmid.

### Electroporation of *P. aeruginosa*

To introduce replicating plasmids in *P. aeruginosa*, 1 mL of overnight culture of *P. aeruginosa* in LB media inoculated from single colony was pelleted at 10,000 *g* for 2 min. Cells were washed twice with 1 mL of 300 mM sucrose before resuspending in 100 μL of 300 mM sucrose. 30–50 ng of purified plasmid DNA was then added and the cell suspension and transferred to a 2 mm gap electroporation cuvette (Thermo Fisher Scientific). Cells were pulsed on a Bio-Rad Gene Pulser XCell (Bio-Rad Laboratories, Hercules, CA) with the following conditions before rapid addition of 1 mL of LB media to the cuvette: 2500 voltage (V), 25 microfarads (μF), 200 Ohms. The cell suspension was then transferred to a 2 mL microfuge tube and recovered for one hour at 37 °C and 230 rpm shaking before dilution and plating onto the appropriate antibiotic-containing plates.

### Insertion of mini-CTX1-Gm and pUC18-mini-Tn7T-Gm derived constructs into *P. aeruginosa*

For insertion at the neutral *attB* site using mini-CTX1-Gm derived vectors, the mating competent strain *E. coli* SM10 was used. Briefly, *E. coli* SM10 cells containing the mini-CTX1-Gm vector to be delivered were scraped from an overnight plate and mixed on a plain LB agar plate with *P. aeruginosa* cells scraped from an overnight plate that had been incubating at 42 °C for at least 4 h. The conjugation was allowed to proceed for 4-8 h at 37 °C before the cells were scraped and resuspended into 1 mL of LB. Dilutions of the suspension were plated on PIA agar medium + 30 μg/mL Gm. Single colonies were selected and grown overnight and subsequently electroporated with plasmid pFLP2 and plated onto LB + 250 μg/mL Carb to remove the Gm-resistance cassette. To cure the strain of the pFLP2 plasmid, the strain was then streaked onto LB + 5% sucrose plates for *sacB*-mediated negative selection. For mini-Tn7 delivery, the aforementioned protocol was used with the following changes: for conjugation of the pUC18-mini-Tn7-Gm plasmid, a triparental mating was utilized which included *E. coli* SM10 pTNS3 which delivers the transposase necessary for construct integration into the chromosome. All antibiotic sensitivities and resistances were verified by patching. The *attB* and *att*Tn7 insert sites were verified by Sanger sequencing (GeneWiz).

### Generation of *P. aeruginosa* deletion mutants

Deletions were generated using the suicide vector pKNG101 or pEX18Gm as described in references[98,99] with minor modifications. Briefly, 500 bp upstream of the DNA targeted for removal, as well as the first three codons of the gene of interest (GOI) were PCR amplified with Upstream-Forward and Upstream-Reverse primers. The last 3 codons and 500 bp downstream of the GOI were amplified with Downstream-Forward and Downstream-Reverse primers. Upstream-Reverse and Downstream-Forward primers were designed with complementary regions to each other at the 3′ and 5′ ends, respectively, to allow for overlap extension PCR or Gibson Assembly to create the "mutator fragment". For pKNG101 constructs, the resulting mutator fragment was then PCR amplified for restriction-less PCR cloning as described in ref. 91. Briefly, F1/R2 F2/R1 primer pairs were used to amplify the mutator fragment and introduce 5′ BamHI and 3′ SpeI complementary overhangs. The products were annealed together and phosphorylated overnight with T4 Polynucleotide Kinase (Promega, Madison, WI) before ligation into BamHI/SpeI digested pKNG101 using T4 DNA

Ligase (Promega). pKNG101 derived mutator plasmids were maintained in *E. coli* CC18 *λpir*.

The mutator plasmid was conjugated from *E. coli* CC18 *λpir* or SM10 *λpir* into *P. aeruginosa* MPAO1 by triparental mating with *E. coli* 1047 cells containing plasmid pRK2013[100]. Briefly, a sweep of fresh colonies from both of the two *E. coli* strains were mixed on an LB plate and incubated at 37 °C for 4 h. A sweep of fresh MPAO1 colonies that had been incubated at 42 °C for four hours was then mixed with the *E. coli* strains and incubated for an additional 4 h. The spot was then resuspended in 1 mL of LB and dilutions plated on PIA + Sm plates to select for MPAO1 exconjugants. Sm-resistant colonies were streaked onto LB plates supplemented with 5% sucrose and grown at RT for 2 days to select for the second crossing-over event. Colonies were then screened by PCR using Check Forward and Reverse primers (**SI**) for successful deletion of the GOI. Mutations were then verified by Sanger sequencing using the Check Forward and Reverse primers.

### *P. aeruginosa* growth assays
To monitor growth over time, 1 mL of overnight cultures was centrifuged at 10,000 $g$ for 2 min. The cells were resuspended in fresh LB media and pelleted again before resuspension in 1 mL fresh media. $OD_{600}$ was read and cultures were normalized to an $OD_{600} = 0.05$ in appropriate media. For growth assays performed in 96-well plates, 200 µL of culture was aliquoted in duplicate wells. Plates were incubated and monitored in a BioTek H1 Synergy multimode plate reader (BioTek, Winooski, VT) at 37 °C with 567 cpm linear shaking. Absorbance at 600 nm was read every 15 min for 24 h.

### Fluorescence reporter assays
To measure mScarlet-I fluorescence, MPAO1 strains carrying reporter plasmids were subcultured 1:100 from overnight cultures into 5 mL of fresh LB media in 50 mL conical tubes. Cultures were incubated at 37 °C and 230 rpm shaking for 2 h before addition of test compound(s). Cultures were then incubated for an additional 3 h before transfering to ice for 10 min to stop growth. mScarlet-I reporter strains were read directly in media as background fluorescence was minimal at the wavelengths used. 200 µL of culture was read in triplicate in a black-walled 96-well plate (655096; Greiner Bio-One, Stonehouse, UK). A red (excitation 575/15 nm, emission 635/32 nm, dichroic mirror 595 nm) filter cube was used for fluorescence readings in a BioTek H1 Synergy multimode plate reader. Fluorescence readings were normalized to the absorbance at 600 nm.

### RNA isolation, sequencing and analysis
For sample preparation to conduct RNA-seq experiments, 1 mL of overnight culture of *P. aeruginosa* was pelleted 10,000 $g$ for 2 min and resuspended in 1 mL of fresh LB media, pelleted and resuspended again in 1 mL of LB (i.e., washed twice in LB). Cells were then inoculated into 12 mL of LB media at an $OD_{600}$ of 0.05 in 125 mL Erlenmeyer flasks. Cultures were incubated at 37 °C and shaken at 230 rpm for 2 h before addition of glyoxal. Samples of approximately 2 x 10^8 cells were taken at 15 min and 1-h post addition and immediately mixed with 2 volumes RNAProtect Bacteria Reagent (Qiagen, Valencia, CA), followed by incubation at RT for 5 min. Samples were then centrifuged at 5,000 $g$ for 10 min, supernatants aspirated, and pellets stored at -80 °C. All samples were done in biological triplicate.

To isolate RNA, RNEasy Mini RNA isolation kit (Qiagen, Valencia, CA) was used according to the manufacturers protocol. Briefly, pellets from above were thawed on ice. Bacteria were lysed by resuspension in 200 µL of TE buffer supplemented with 15 mg/mL lysozyme and 20 µL Proteinase K (Qiagen) followed by incubation at RT for 10 min. 700 µL of buffer RLT and 500 µL of ethanol was added to each sample before application to a RNeasy Mini column and centrifugation at 8000 $g$ for 30 s. Columns were washed with 350 µL RW1 before on-column DNAse treatment. 10 µL of DNase I diluted in 70 µL buffer RDD (Qiagen,

Valencia, CA) and applied to each column and incubated at room temperature (RT) for 30 min. 350 µL of buffer RW1 was then applied to the column and incubated 5 min at RT. Columns were washed twice with 500 µL buffer RPE before a centrifugal drying step (8000 $g$ for 2 min). RNA was eluted in two, 50 µL washes with nuclease-free water. RNA quality and quantity were assessed by nanodrop and by electrophoresis on a 1% bleach agarose gel[101].

RNA-seq and differential gene expression analysis was performed by GeneWiz. For RNA sequencing, samples were depleted for rRNA and strand-specific RNA sequencing of 5-10 million reads per sample was implemented. Transcriptional start sites were found by visualizing the aligned RNA-seq reads against the genome using the Integrated Genome Viewer[102]. STRING analysis and output was then employed by using genes that were differentially regulated 3-fold or greater after glyoxal treatment as compared to untreated cells. RNA-seq data as well as STRING outputs can be found in Supplementary Information (SI).

### Fluorescence microscopy for protein subcellular localization
Overnight cultures of *P. aeruginosa* MPAO1 *attb*::P$_{A1/04/03}$-mCherry harboring sfGFP-tagged ArqI under the control of an arabinose-inducible promoter in pSB109 were subcultured 1:100 in fresh LB medium supplemented with 0.02% arabinose. A low concentration of arabinose (0.02%) was used to prevent overexpression that could lead to imaging artifacts[103]. Cultures were incubated at 37 °C and with 230 rpm shaking and samples taken every 2 h. Cells were fixed by first chilling them on ice for 10 min before pelleting and resuspending in an equal volume of 4% paraformaldehyde for 15 min at RT. Samples were centrifuged and pellets resuspended in an equal volume of PBS for imaging.

For imaging, 5 µL of PBS resuspended cells were added to a poly-L-lysine coated chambered cover glass (C4-1.5H-N, Cellvis, Sunnyvale, CA) containing 495 µL of PBS. Cells were allowed to settle onto the bottom of the chamber for 15 min before imaging. Imaging was performed on a Zeiss 880 Airyscan microscope with a 63x 1.4 NA objective (Zeiss, Oberkochen, Germany) on the "Airyscan SR" acquisition setting at RT. The constitutive mCherry (561 nm laser) signal of the cells was used to identify cells for imaging as well as for analysis. The ArqI-sfGFP channel was acquired using the 488 nm argon laser. Representative fields containing 50-200 bacterial cells were taken in technical triplicate and biological triplicate. Images were analyzed using the MicrobeJ plugin for ImageJ[104]. The mCherry channel was used to detect bacterial cells, with a threshold of bacteria that are over 1 µm in length to eliminate cells that are not lying flat within the focal plane. For foci counting, the 'maxima' function was used on the sfGFP channel, gating on maxima that were brighter than 1000 AU. The number of foci-containing bacteria was then divided by the total number of bacteria detected to generate the percent of the bacterial population with foci present. To generate fluorescence intensity heatmaps, the straighten, shape, and polarity functions of MicrobeJ were enabled, gating on cells that are greater than 1 µm in length. Polarity was set by the pole with the greatest mean intensity in the sfGFP channel (INTENSI-TY.ch1.mean). Plots were generated by the 'Shape Plot' function using the sfGFP channel intensity.

For time-lapse imaging, a Ti2-E inverted microscope with X-Cite XYLIS XT720S Broad Spectrum LED Illumination System equipped with an Okolab stage-top incubation chamber to maintain 37 °C, a DS-Qi2 CMOS camera, and a CFI Plan Apochromat Lambda D 100X oil objective was utilized. Agarose pads for imaging were made according to ref. 105 with modifications. Briefly, a 125 µL Gene Frame (Thermo Scientific) was affixed to a microscopy slide. 50 µL of molten 1.5% agarose LB medium +/− 4 mM GO was spotted inside the gene frame, and a second slide placed on top until the agarose solidified. Bacteria were subculture in LB medium from overnight cultures for 2 h at 37 °C, with 230 rpm shaking before diluting 1:2 in fresh LB medium. 2 µL spots of diluted culture was then spotted onto the agarose pads and allowed

to dry before affixing a coverslip on top. The slides were allowed to acclimate in the imaging chamber at 37 °C for 15 min before imaging. Images were taken every 30 min for 2.5 h, and subsequently analyzed in ImageJ.

## Fluorescence recovery after photobleaching (FRAP)

Bacteria, grown overnight in LB broth with 30 µg/mL Gentamicin, were back-diluted 1:100 into EZ-rich media[106] and grown for ~2.5 hours with 0.02% w/v L-arabinose with shaking (250 rpm) at 37 °C. Slides were prepared for imaging by placing bacteria on 1.5% agarose EZ-rich pads, allowing the droplet to dry before covering them with a #1 coverslip and sealing them with Valap (1:1:1 mixture of vaseline, lanolin and paraffin). All FRAP experiments were performed with an inverted custom-built wide-field microscope at 23 °C (Princeton University). Images were obtained using a 1.49 NA 100x Nikon objective and 2x optical magnification, and captured with an Andor EM-CCD camera chilled to −80 °C. Each experiment consisted of 80 frames imaged at a kinetic cycle of 0.0409 s and an exposure of 0.04 s. The frames were allocated as 5 pre-bleach frames, 10 bleaching frames, and 65 recovery frames. For the bleaching, individual cells were exposed to a focused FRAP laser beam (488 nm with a 525/50 nm emission filter 525) for 0.5 s at one quarter cell-length. For each strain, FRAP experiments were performed in triplicate with 20 cells imaged each day for a total of 60 cells. The resulting data was analyzed with custom MATLAB software following the analysis detailed in ref. [107].

## Western blot analysis

To assess protein expression, cells were pelleted at 10,000 $g$ for 2 min and resuspended in B-PER (Thermo Scientific) lysis buffer (supplemented with 100 µg/mL lysozyme, 20 µg/mL DNase I, and 20 µg/mL RNase A) according to the equation: volume of lysis buffer (µL) = volume of culture (mL) x $OD_{600}$ of culture x 100. After a 15-min incubation at RT with rotation, lysates were centrifuged at 16,000 $g$ for 10 min to pellet any insoluble material. Lysates were separated by SDS-PAGE and transferred to PVDF membranes (EMD Millipore). Membranes were blocked for 1 h at RT with 5% milk in PBS-Tween 20 (PBS-T; PBS + 0.05% Tween 20). Membranes were then probed with primary antibody for 1 h at RT followed by washing with PBS-T three times for 5 min each. Membranes were subsequently probed with HRP-conjugated secondary antibody for 1 h at RT before washing with PBS-T three times for 15 min per wash. Blots were developed using Pierce ECL Western Blotting substrate (Thermo Scientific) and captured on an Azure 400 imaging system (Azure Biosystems, Dublin, CA). The following antibodies were diluted in 5% milk in PBS-T to the indicated concentrations: Mouse anti-FLAG M2 (1:10,000, F1804, Sigma Aldrich), rabbit anti-HA (1:10,000, H6908, Sigma), rabbit anti-GFP (1:10,000, A-6455, Invitrogen), mouse anti-RpoB (8RB13, Biolegend, San Diego, CA), HRP-conjugated goat anti-mouse (1:5000, Jackson ImmunoResearch, UK), HRP-conjugated goat anti-rabbit (1:5000, Jackson ImmunoResearch). For streptavidin-HRP blots, membranes were blocked in 5% BSA in PBS-T. Streptavidin-HRP (1:10,000, AB7403, Abcam) was diluted in 5% BSA in PBS-T for blotting.

## Glyoxal toxicity plate assay

Experiments were performed as previously described with minor modifications[108]. Briefly, to assess survival and growth against GO stress, overnight cultures of *P. aeruginosa* were subcultured 1:100 in fresh LB media and incubated at 37 °C with 230 rpm shaking for 2 h until cells reached mid-logarithmic phase. Cells were then placed on ice for 10 min to stop growth before normalizing cell density to an $OD_{600}$ of 0.01 in LB medium. Samples were then serially diluted 10-fold 5 times in LB medium in a 96-well plate before spotting 10 µL of each dilution onto LB agar plates containing the appropriate antibiotic, inducing compound, and test compound(s) additions. Plates were then incubated overnight at 37 °C before imaging the following day.

## Immunoprecipitation (IP) of FLAG-tagged proteins for mass spectrometry (MS)

FLAG-tagged proteins were IPed from *P. aeruginosa* MPAO1 cells using Anti-FLAG M2 magnetic beads (M8823, Sigma). Briefly, 1 mL of MPAO1 pSB109-ArqI-FLAG overnight cultures were centrifuged 10,000 $g$ for 2 min. Supernatant was removed and cells were washed twice with M9 minimal medium. 500 mL of M9 minimal medium supplemented with 30 µg/mL gentamicin and 0.2% L-arabinose was inoculated to an $OD_{600}$ of 0.1 in 2 L flasks. Cultures were incubated 37 °C, 230 rpm until an $OD_{600}$ of ~0.4 was reached. Cultures were then harvested by centrifugation at 5000 $g$ for 10 min before removing supernatant and resuspending the resulting cell pellet in 1 mL of lysis buffer (50 mM Tris pH 7.5, 150 mM NaCl, mini cOmplete protease inhibitor (Roche, Basel, Switzerland), 20 µg/mL DNase I, and 20 µg/mL RNase A). Samples were then sonicated for 4 x 10 s bursts with a Branson Sonifier S-450 (Branson Ultrasonics Corp., Danbury, CT) equipped with a 0.75 inch microtip. Samples were pelleted at 21,000 $g$ for 45 min at 4 °C. Supernatants were harvested by centrifugation and protein concentration read by Bradford reagent. Protein concentrations were normalized by dilution with lysis buffer before applying to 100 µL beads pre-equilibrated in lysis buffer. Samples were incubated on a rotisserie at RT for 1 h. Beads were then washed three times for 5 min each in wash buffer (lysis buffer without DNase I/RNase A). Beads were resuspended in 100 µL of 1X Laemmli buffer and boiled at 94 °C for 10 min. 30 µL of this boiled sample was then resolved by SDS-PAGE to confirm the IP of ArqI-FLAG and for band excision for mass spectrometry.

## Mass Spectrometry of FLAG-ArqI IP from *P. aeruginosa*

Immunoprecipitated FLAG-ArqI protein samples were separated by electrophoresis on a 4-12% Bis-Tris gradient gel (Invitrogen). The gel was then fixed by incubation in 50% methanol/10% glacial acetic acid for 15 min and stained with Coomassie Brilliant Blue for 30 min at RT. Following sufficient destaining of the gel in 10% methanol/7.5% glacial acetic acid, protein bands of interest were carefully excised and added to sterile 1.5 mL Eppendorf tubes. The gel pieces were washed as follows: (1) 100 mM ammonium bicarbonate, 600 RPM, 37 °C, 30 min; (2) 1:1 (vol/vol) ammonium bicarbonate/acetonitrile, 600 RPM, 37 °C, 15 min; (3) 100% acetonitrile, 37 °C, 15 min. The samples were then reduced and alkylated by subsequent incubations in 250 mM DTT and 50 mM iodoacetamide, respectively. Trypsin digestion was performed with 5 µg trypsin/sample in 100 mM ammonium bicarbonate at 37 °C for 18 h. Tryptic peptides were collected in a sterile Eppendorf tube and the gel pieces then washed in 50% acetonitrile/5% formic acid to retrieve additional peptides. The combined peptide fractions were dried by vacuum centrifugation at RT and then reconstituted in 3% acetonitrile/0.1% formic acid for analysis by LC-MS/MS. Approximately 100 ng of peptides were separated by liquid chromatography in a 25 cm C18 column (Pep-Map) and then analyzed by tandem mass spectrometry on an LTQ Orbitrap XL (Thermo Fisher Scientific). The raw LC-MS/MS data was searched against the *Pseudomonas aeruginosa* PAO1 proteome using the Peaks Bioinformatics software.

## Mass Spectrometry of ArqI crystallography preparation

The same protein preparation used to create the ArqI crystals (below) was heated (95 °C for 5 min) with lysis buffer (100 µL, 12 mM sodium lauroyl sarcosine, 0.5% sodium deoxycholate, 50 mM triethylammonium bicarbonate (TEAB)). The samples were treated with tris(2-carboxyethyl) phosphine (10 µL of 55 mM in 50 mM TEAB, 30 min, 37 °C) followed by treatment with chloroacetamide (10 µL, 120 mM in 50 mM TEAB, 30 min, 25 °C in the dark). They were then diluted 5-fold with aqueous 50 mM TEAB and incubated overnight with Sequencing Grade Modified Trypsin (1 µg in 10 µL of 50 mM TEAB (Promega), followed by the addition of an equal volume of ethyl acetate/trifluoroacetic acid

(TFA, 100/1, v/v). After vigorous mixing (5 min) and centrifugation (13,000 g, 5 min), the supernatants were discarded, and the lower phases were dried in a centrifugal vacuum concentrator. The samples were then dissolved in acetonitrile/water/TFA (solvent A, 100 μL, 2/98/0.1, v/v/v) and loaded onto a small portion of a C18-silica disk (3M, Maplewood, MN) and placed in a 200 μL pipette tip. Prior to sample loading the C18 disk was prepared by sequential treatment with methanol (20 μL), acetonitrile/water/TFA (solvent B, 20 μL, 80/20/0.1, v/v/v) and finally with solvent A (20 μL). After loading the sample, the disc was washed with solvent A (20 μL, eluent discarded) and eluted with solvent B (40 μL)[109]. The collected eluent was dried in a centrifugal vacuum concentrator.

The eluants were reconstituted in water/acetonitrile/FA (solvent E, 10 μL, 98/2/0.1, v/v/v), and aliquots (5 μL) were injected onto a reverse phase nanobore HPLC column (AcuTech Scientific, C18, 1.8 μm particle size, 360 μm x 20 cm, 150 μm ID), equilibrated in solvent E and eluted (500 nL/minute) with an increasing concentration of solvent F (acetonitrile/water/FA, 98/2/0.1, v/v/v: min./% F; 0/0, 5/3, 18/7, 74/12, 144/24, 153/27, 162/40, 164/80, 174/80, 176/0, 180/0) using an Eksigent NanoLC-2D system (Sciex, Framingham, MA)). The effluent from the column was directed to a nanospray ionization source connected to a high-resolution orbitrap mass spectrometer (Q Exactive Plus, Thermo Fisher Scientific) acquiring mass spectra in a data-dependent mode alternating between a full scan (m/z 350–1700, automated gain control (AGC) target 3 x 10⁶, 50 ms maximum injection time, FWHM resolution 70,000 at m/z 200) and up to 10 MS/MS scans (quadrupole isolation of charge states ≥ 2, isolation width 1.2 Th) with previously optimized fragmentation conditions (normalized collision energy of 32, dynamic exclusion of 30 s, AGC target 1 x 10⁵, 100 milliseconds maximum injection time, FWHM resolution 35,000 at m/z 200). The raw data was analyzed in Proteome Discoverer 2.4 with appropriate chemical modifications.

## Mass Spectrometry of *E. coli* purified ArqI

An aliquot of 50 μg of protein extract was solubilized in 100 ul of 8 M urea/50 mM Ammonium bicarbonate (ABC), pH 8.0 solution, reduced with 5 mM DTT for 30 min, and alkylated with 15 mM iodoacetamide for 20 min in the dark. The protein solution was then diluted (1:4) with 50 mM ABC to reduce the urea concentration and digested overnight at 37 °C with MS grade Chymotrypsin (1: 50, Thermo Fischer, catalog # 90056). Digestion was stopped with formic acid (0.1% final concentration), centrifuged and the supernatant was used to purify peptides using C18 spin columns (G-Biosciences, catalog # 786-931) as per manufacture protocol. Eluted desalted and purified peptides were dried using a speed vacuum and resolubilized in 40 μl of 2.5% acetonitrile and 0.1% FA for loading into the mass spectrometry device. 500 ng of purified peptides were loaded onto a Vanquish Neo UHPLC system (Thermo Fisher) with a heated trap and elute workflow with a c18 PrepMap (5 mm) trap column (P/N 160454) in a forward-flush configuration connected to a 25 cm Easyspray analytical column (P/N ES802A rev2) 2 μ,100 A, 75 μm x 25 with 100% Buffer A (0.1% formic acid in ddH₂O) and the column oven operating at 40 °C. Peptides were eluted over a 90 min gradient using 80% acetonitrile, and 0.1% formic acid (buffer B), going from 2% to 10% over 10 min, to 45% over the next 66 min, then to 99% over 10 min, and then kept at 95% for 14 min, after which all peptides were eluted. For a 150 min run, peptides were eluted over a 150 min gradient using 80% acetonitrile, and 0.1% formic acid (buffer B), going from 2% to 8% over 10 min, to 45% over 115 min, then to 65% over 20 min, and then kept at 95% for 10 min, after which all peptides were eluted.

Spectra were acquired with an Orbitrap Eclipse Tribrid mass spectrometer with FAIMS Pro interface (Thermo Fisher Scientific) running Tune 3.5 and Xcalibur 4.5. For all acquisition methods, the spray voltage was set to 1900 V, and the ion transfer tube temperature was set at 300 °C, FAIMS switched between CVs of −45 V, −

55 V, and −65 V with cycle times of 2.0, 1.5, and 1.0 respectively. MS1 spectra were acquired at 120,000 resolutions with a scan range from 375 to 1500 m/z, normalized AGC target of 300%, and maximum injection time of 50 milliseconds, S-lens radio frequency (RF) level set to 30, without source fragmentation and datatype positive and profile; Precursors were filtered using monoisotopic peak determination set to peptide; included charge states, 2-7 (reject unassigned); dynamic exclusion enabled, with n = 1 for 60 s exclusion duration at 10 ppm for high and low. DDMS2 scan using isolation mode Quadrupole, Isolation Window (m/z): 1.6; activation Type set to HCD with 30% Collision Energy (CE), OrbiTrap as a detector with 30 K resolution, AGC target set to 50,000; maximum Injection Time for 54 milliseconds, microscans: 1 and data type set to Centroid.

## MS/MS data analysis of *E. coli* purified ArqI protein

Raw data were analyzed using Proteom Discoveror 2.5 (Thermo Fisher) using Sequest HT search engines. The data were searched against the *E. coli* Uniprot protein sequence database supplemented with *P. aeruginosa* recombinant SUMO-ArqI protein; "MGHHHHHHGSLQD SEVNQEAKPEVKPEVKPETHINLKVSDGSSEIFFKIKKTTPLRRLMEAFAKR QGKEMDSLTFLYDGIEIQADQTPEDLDMEDNDIIEAHREQIGGSGGGMTY HVLVQFDVPSDKREAFAAAGLFDANGSLQNEPGTLRFEVIRDENNRNRFY LDEVYEDEAAFLQHCRNETIARFYELIDSYAFGPLFLFKGYRVEG"

The search parameters included precursor mass tolerance of 10 ppm and 0.02 Da for fragments, allowing 2 missed trypsin cleavages, oxidation (Met) and acetylation (protein N-term) and methylglyoxal hydroimidazalone-1/+54.011 Da (K, R), carboxymethylArginine/+58.013 Da (R), glyoxal hydroimidazalone-1/+40.011 Da (K, R), glyoxal AGE/+21.984 Da (R) and Dihydroxyimidazolidine/+72.021 Da (R) as variable modifications, and carbamidomethylation (Cys) as a static modification. Percolator PSM validation was used with the following parameters: strict false discover rate (FDR) of 0.01, relaxed FDR of 0.05, maximum ΔCn of 0.05, and validation based on q-value. Precursor Ions Quantifier settings were Peptides to Use: Unique + Razor; Consider Protein Groups for Peptide Uniqueness was set as True; Precursor Abundance Based On Intensity; Normalization based on Total Peptide Amount; Scaling Mode set as none, Protein Abundance calculated by the summed intensity of connected peptides.

## Protein Purification

SUMO-tagged proteins were expressed in either *E. coli* LOBSTR or T7 Express lysY^(i/q) strains. Overnight cultures were subcultured 1:100 into fresh LB medium supplemented with 50 μg/mL kanamycin and grown at 37 °C, with 230 rpm shaking until an OD₆₀₀ of ~0.8 was reached. Protein expression was induced with addition of 1 mM IPTG and the cultures were incubated at 18 °C with 150 rpm shaking overnight before harvesting by centrifuging cultures at 5000 g for 10 min. Pellets were stored at −80 °C before purification.

To purify His₆-SUMO-tagged ArqI, the cell pellet was resuspended in 25 mL/L culture lysis/wash buffer (50 mM Tris-HCl pH 8.0 150 mM NaCl, 50 mM imidazole, 1 mM beta-mercaptoethanol (βME), and 1 mM phenylmethylsulfonyl fluoride (PMSF)) and sonicated for 5 x 30 s pulses using a Branson Sonifier S-450 sonicator. The sample was centrifuged 18,000 g for 45 min before filtering the supernatant through 0.45 μm syringe filter to remove any remaining cell debris. The supernatant was then applied to a 2-4 mL bed of Ni-NTA resin (Qiagen) equilibrated in lysis/wash buffer and allowed to flow through by gravity. The resin was then washed with 100 mL of lysis/wash buffer before elution in 10−20 mL of elution buffer (50 mM Tris-HCl pH 8.0, 150 mM NaCl, 300 mM imidazole, 1 mM βME). Fractions were evaluated by SDS-PAGE and Coomassie staining before pooling and dialyzing overnight at 4 °C against 4 L of ArqI buffer (50 mM Tris-HCl pH 8.0, 150 mM NaCl, 1 mM βME). The SUMO-tag was then cleaved from ArqI by addition of purified ULP1 at a ratio of 1:10 ULP1:SUMO-ArqI by weight and then incubated at RT for 1 h. The cleaved His₆-SUMO

protein was then removed by passing the sample over 1 mL of Ni-NTA (Qiagen) resin three times by gravity flow. To further remove impurities, SEC was utilized on a Cytiva Äkta Pure 25 chromatography system. Purified ArqI was concentrated to ~10 mg/mL using 10 kDa cutoff centrifugal filters (Amicon; Sigma-Aldrich) before centrifugation at 20,000 $g$ for 10 min to remove any aggregated protein. The sample (2 mL) was then applied to a HiLoad Superdex 75 16/600 pg (Cytiva) column pre-equilibrated in ArqI buffer at a flow rate of 1 mL/minute at RT. Before pooling fractions, peak fractions were collected and analyzed by SDS-PAGE and Coomassie staining to identify fractions with the highest abundance and purity of ArqI. The resulting ArqI protein was concentrated by centrifugal filtration before snap freezing in liquid nitrogen and storing at -80 °C. At this stage ArqI protein was >95% pure and protein concentrations were determined by Braford reagent (Bio-Rad Laboratories, Hercules, CA) using a bovine serum albumin (BSA) standard curve.

The 6x-His-CPD-tagged full-length PqsA was expressed in *E. coli* T7 Express lysY$^{i/q}$ with the addition of 1 mM IPTG overnight at 18 °C. The cell pellet was lysed by sonication in 50 mM TrisHCl pH 8.0, 300 mM NaCl, 1 mM βME, 1 mM PMSF buffer. The 6x-His-CPD-PqsA protein was captured by a Ni$^{2+}$-NTA (Qiagen) column equilibrated with the wash buffer (lysis buffer without PMSF) and eluted using 300 mM imidazole in wash buffer. Eluted proteins were dialyzed against wash buffer using 6-8 kDa MWCO dialysis tubing (Spectra/Por, Spectrum Laboratories, Inc.) overnight at 4 °C. The 6x-His-CPD tag was cleaved by adding 5X molar excess of phytic acid and incubating for 4 h at RT. Cleaved tags and uncleaved proteins were removed by Ni$^{2+}$-NTA and MonoQ (Cytiva) columns (as described above). PqsA full-length protein was then further purified using a Superdex 75 (Cytiva) column in 50 mM TrisHCl pH 8.0, 150 mM NaCl, 1 mM βME buffer. Purified proteins were frozen using liquid N$_2$ and stored at −80 °C.

The GST-tagged PqsA$_{NTD}$ was expressed in T7 Express lysY$^{i/q}$ competent *E. coli* cells (New England Biolabs) by adding 1 mM IPTG and then incubated at 18 °C with 150 rpm shaking applied overnight. Cells were lysed by sonication in 50 mM TrisHCl pH 8.0, 300 mM NaCl, 1 mM βME, 1 mM PMSF buffer. GST-tagged PqsA$_{NTD}$ protein was captured in glutathione resin (GenScript) and eluted with 10 mM reduced glutathione. The GST-tag was cleaved with TEV protease and PqsA$_{NTD}$ purified using Superdex 75 size exclusion column (Cytiva) in 50 mM TrisHCl pH 8.0, 150 mM NaCl buffer.

For selenomethionine (SeMet)-labeled protein, the above protocol was used with the following changes. Cells were subcultured into LB, and before induction of protein expression, cells were harvested by centrifugation at 10,000 $g$ for 10 min. Cells were then washed using 25 mL/L culture with M9 medium and resuspended in a final volume of 25 mL M9 medium. M9 media was prepared as detailed in 'bacterial strains and growth conditions' with the following changes: substitution of 0.2% glycerol for 0.4% w/v glucose, addition of 10 μg/mL thiamine, 75 μg/mL SeMet, 50 μg/mL leucine, 50 μg/mL isoleucine, 50 μg/mL valine, 100 μg/mL phenylalanine, 100 μg/mL lysine, 100 μg/mL threonine, and omission of FeCl$_3$. Cells were then inoculated into the same volume of M9 media as in the original LB culture volume. Protein induction was initiated with the addition of 1 mM IPTG (final concentration) and cultures incubated overnight at 18 °C with 150 rpm shaking before harvesting the next day via centrifugation.

## Modeling of PqsA structure

A PqsA structure of the ligand binding (N-terminal) domain in complex with anthraniloyl-AMP is available (PDB code 5OE4;[67]). This structure was aligned with a AlphaFold[66] generated structure of the full-length PqsA sequence to align anthraniloyl-AMP with the full-length prediction (where this ligand was absent).

## ArqI docking

To investigate how ArqI and PqsA are likely to interact, we utilized two docking software platforms: AlphaFold3[66] and HDOCK[54]. In both platforms, docking simulations were carried out for tetrameric PqsA and dimeric or hexameric ArqI. Interestingly, the AlphaFold server was unable to successfully predict the crystallographic hexameric ArqI structure in the binding simulations, however, there was one promising model of dimeric ArqI bound to tetrameric PqsA that had an interaction surface that supports the yeast two hybrid data. Utilizing the tetrameric PqsA model from this result, another round of docking experiments was performed in HDOCK[54], testing for binding of the crystallographic ArqI hexamer. The majority of the simulated binding events for the AF-PqsA tetramer and ArqI hexamer support the Y2H data.

## Small ligand docking

To investigate likely small molecule binding modes in the ArqI ligand binding pocket, docking experiments were performed in HDOCK[54]. Docking experiments were performed in HDOCK with LsrG (PDB codes 2GFF and 3QMQ) and ArqI dimer for methylglyoxal (MGO) and 4,5-dihydroxy-2,3-pentanedione (DPD). For these small molecules, the binding site was essentially identical across proteins and aligned well to an acetate molecule that is modeled in the ArqI ligand binding pocket.

## Diagnostic size exclusion chromatography (SEC)

To estimate protein size and quaternary structure by SEC, a Cytiva Äkta Pure 25 chromatography system equipped with a HiLoad Superdex 75 16/600 pg or Superose 6 Increase 10/300 GL (Cytiva) column was utilized. The columns were equilibrated in 50 mM Tris-Cl pH 8.0, 150 mM NaCl, 1 mM βME or 50 mM Tris-Cl pH 7.4, 500 mM NaCl, 1 mM βME. A total volume of 2 mL at 10 mg/mL protein concentration was loaded for size estimation. Column calibration was carried out using 300 μL at 70 μM of Gel Filtration Marker Kit (MWGF1000; Sigma-Aldrich—carbonic anhydrase 29 kDa, albumin 66 kDa, alcohol dehydrogenase 150 kDa, ß-amylase 200 kDa, apo-ferritin 443 kDa) at a flow rate of 0.4 mL/minute at RT.

## Crystal structure determination

For crystallography native and SeMet-substituted ArqI was expressed and purified as described above. For crystallization experiments ArqI was purified in 50 mM Tris-HCl pH 8.0, 50 mM NaCl and 1 mM βME, and screened for crystallization by sparse matrix crystallization screens at a concentration of 15 mg/mL.

SeMet-ArqI crystals were detected in JCSG + 19 (0.1 M sodium acetate pH 4.6, 8% PEG 4000) at RT. Crystallization conditions were optimized to 50 mM sodium acetate pH 5.0, 10% PEG 4000 with a protein to well drop ratio of 2:1. After one day of growth the crystals were cryoprotected in mother liquor with 20% glycerol and cryocooled for data collection. A Se-(Single wavelength Anomalous Diffraction (SAD) dataset was collected at the Stanford Synchrotron Radiation Lightsource (SSRL) 12-2 at the peak absorbance wavelength for incorporated SeMet residues (E = 12662 eV). The SAD dataset was indexed and integrated using XDS (versions January 26, 2018 to January 10, 2022)[110], and scaled in AIMLESS 0.7.4[111]. Phases were solved and a model was generated in Crank-2 in CCP4[112]. The resulting model was refined in phenix.refine (versions 1.14-3260 to 1.20.1-4487)[113] and coot (versions 0.8.9.1 to 0.9.8.3)[114]. The Se-SAD ArqI structure was solved in the P 2$_1$2$_1$2 unit cell with 6 subunits in the asymetric unit (ASU) and diffraction resolution to 2.0 Å. All subsequent datasets were solved by molecular replacement in Phaser with phases supplied from the Se-SAD-derived dimer[115], and the resulting model was refined as discussed above.

Native ArqI crystals were grown and optimized from the same condition as SeMet-substituted ArqI protein. A native dataset was

collected from crystals grown in 50 mM sodium acetate pH 5.0, 12% PEG 4000. Crystals were cryocooled in mother liquor supplemented with 15% glycerol before data collection at ALS 5.0.2 with X-rays at E = 12398 eV. The native dataset was solved in a C 222₁ unit cell with 6 subunits in the ASU, and with resolution to 2.0 Å. As in the Semet-ArqI crystals, electron density showed glyoxal modifications on Arg49 in several of the subunits (C, D, and E, in this structure) and βME-modifications for all Cys68 residues. Modifications were modeled and refined into the coordinate file. Using MS we identified three possible arginine adducts (dihydroxyimidazolidine, hemiaminal or carboxymethyl), and the electron density only accommodated the linear adducts—hemiaminal or carboxymethylarginine. Because these adducts are not easily distinguished experimentally, both hemiaminal or carboxymethylarginine adducts were modeled and refined alongside the unmodified arginine residue. The resulting occupancy at position C-Arg49, D-Arg49, and E-Arg49 is: 0.4, 0.4, and 0.6 for unmodified; 0.4, 0.5, and 0.3 for the hemiaminal adduct; 0.2, 0.1, and 0.2 for carboxymethylarginine, respectively.

Due to the involvement of Arg16 at the hexamer trimeric interface, an R16A variant of ArqI was generated by site directed mutagenesis. ArqI-R16A was expressed and purified as described above. The protein was subjected to crystallization screening and was found to crystallize using a Hampton Research Crystal Screen, condition 18 (0.1 M sodium cacodylate pH 6.5, 0.2 M magnesium acetate tetrahydrate, 20% PEG 8000). Following optimization, crystals grown in 0.1 M sodium cacodylate pH 6.5, 0.2 M magnesium acetate and 30% PEG 8000 were cryocooled in mother liquor supplemented with 15% glycerol for data collection at SSRL 12-2. Following data processing, the protein was found to crystallize in a H3 unit cell with 2 subunits in the ASU with X-ray diffraction to 1.5 Å resolution. βME -modified Cys68 residues were modeled into the coordinate file as discussed above.

Finally, due to the presence of βME-modifications on the previous datasets, ArqI was expressed and purified as described above, excluding the presence of βME in solubilization and purification buffers. The βME-free protein was then subjected to crystallization trials as described above and crystallized in PACT crystallization screen condition 26: 0.1 M PCB buffer pH 5.0, 15% PEG 1500. The crystals were optimized and crystals from 0.1 M PCB pH 5.5, 15% PEG1500 were cryocooled and subjected to X-rays at SSRL 12-2. Following processing, the crystals were found to form a P 2₁ lattice with 12 subunits in the ASU and X-ray diffraction to 2.1 Å. Data collection and refinement statistics can be found for wild-type ArqI and the R16A mutant in Supplementary Table 1.

### Negative stain electron microscopy

Purified ArqI with the following sequence after protease cleavage from SUMO (SGGGMTYHVLVQFDVPSDKREAFAAAGLFDANGSLQNEPGTLRF EVIRDENNRNRFYLDEVYEDEAAFLQHCRNETIARFYELIDSYAFGPLFLFK GYRVEG) was prepared for negative stain analysis as per ref. 116. To prepare samples for negative stain EM analysis 1.5 μl of ArqI at 1.5 mg/ml (in 0.75 mL, 10 mg/mL in 50 mM Tris-HCl pH 8.0, 50 mM NaCl buffer) was diluted ten-fold. Carbon-coated copper EM grids (400 mesh Cu-film, Electron Microscopy Sciences, Hatfield, PA) were glow-discharged in a Solarus plasma cleaner (Gatan, Pleasanton, CA) for 20 s at 10 W in air and subsequently were held using reverse anti-capillary tweezers, carbon side up, and a 4 μl drop of the diluted ArqI sample was applied at RT and after 1 min blotted with Whatman 1 filter paper. The grids were then serially transferred to 15 μl drops of 1.5% uranyl acetate solution after 5, 10, 15, and 30 s, followed by blotting after the last transfer and finally allowing the grid to air dry. EM data were collected on a JEOL 3200FS transmission electron microscope (JEOL, Peabody, MA) operating at 300 keV with a Gatan K3 Direct Electron Detector (Gatan, Pleasanton, CA) using the program SerialEM[117] at a nominal magnification of 40,000x with a pixel size of 0.92 Å at the

specimen level and using a defocus range of -0.5 μm to -1.5 μm. A total of 1054 movies were collected. Data were processed using RELION 3[118] as follows. Movies were corrected with MotionCorr2 using 5 x 5 patches and the Contrast Transfer Functions (CTF) were calculated with ctffind4. Particles were picked automatically using the Laplacian of Gaussian method using a maximum diameter of 150 Å, yielding 121,129 particles form the 1,1054 images. During extraction the particles were down sampled to a 3.68 Å pixel size (final box size 60 pixels x 60 pixels). Four successive rounds of 2D classification were used to remove noise and bad quality particles, eventually yielding 57,667 good particles. Visual inspection of the 2D class averages show 2-fold and possibly 3-fold symmetric classes and hence several possible point groups were tested in subsequent steps. Initial ab initio models with C1, C2 and D3 symmetry were generated. 3D refinement imposing C1, C2, and D3 symmetry show very similar volumes, but the C2 refinement yielded smoother Fourier Shell Correlation (FSC) curves. As the C2 symmetry is also consistent with the 2D class averages and the volumes, this symmetry was imposed on the volume. The nominal resolution of the final volume was -10 Å using the FSC Gold Standard criteria and -14 Å using the 0,5 correlation criteria (Supplementary Fig. 3F). Visual inspection of the volume suggested a resolution closer to the latter.

### BioID verification of ArqI-PqsA interaction

WT MPAO1 harboring pSB109-ArqI-BirA*-FLAG or pSB109-BirA*-FLAG and pMMB67EH-PqsA-HA were subcultured from overnight cultures 1:100 into 50 mL LB + 50 μM biotin. Cultures were incubated for 4 h at 37 °C and shaken at 230 rpm before induction of protein expression by addition of 0.02% w/v arabinose and 100 μM IPTG. Cultures were incubated for 4 additional hours before cells were harvested by centrifugation at 4,800 g for 10 min. Cells were then washed once with 10 mL PBS and then resuspended in 1 mL lysis buffer (PBS + 100 μg/mL lysozyme + mini cOmplete protease inhibitor (Roche) and sonicated for 3 x 15 s pulses on a Sonics Vibracell at 60% amplitude. Lysates were clarified by centrifugation at 21,000 g for 10 min and protein concentration read by Bradford assay.

10 μg of mouse anti-HA IgG (Clone 16B12, Biolegend) was diluted in 200 μL PBS-T and bound to 50 μL of Invitrogen Protein G Dynabeads for 30 min at RT in a reaction with rotation. Beads were then washed once in 500 μL PBS-T before addition of 500 μL of MPAO1 lysate at 10 mg/mL protein concentration and incubated for 1 h at RT with rotation. Lysate was removed and beads were washed three times in 500 μL of PBS-T. To elute protein, beads were resuspended in 20 μL of Laemmli buffer and boiled for 10 min. Proteins eluted were analyzed by SDS-PAGE fractionation and western blotting (as above).

### Pyoverdine and pyocyanin measurements

1 mL of overnight culture was centrifuged 10,000 g for 2 min and the cells washed twice in 1 mL of fresh LB medium before resuspension in 1 mL of fresh LB. OD₆₀₀ was measured and the strains were then subcultured in LB + 30 μg/mL Gm +/− 0.2% arabinose to an OD₆₀₀ of 0.05 in 125 mL flasks. Cultures were incubated at 37 °C and with 230 rpm shaking for 24 h before transferring to ice for 10 min. OD₆₀₀ was measured before cultures were centrifuged at 4800 g for 10 min and the supernatant collected and filtered through a 0.22 μm syringe filter.

Pyoverdine was measured in culture supernatants as described in ref. 119. Briefly, 200 μL of supernatant was transferred to black-walled 96 well plates and the absorbance read at 405 nm. Values were normalized to the OD₆₀₀ reading of the culture. Pyocyanin measurements were carried out as in ref. 120 with minor modifications. Briefly, 5 mL of culture supernatant was extracted with 3 mL of chloroform. 2 mL of the bottom organic layer was then re-extracted with 2 mL of 0.2 N HCl. The reddish-pink organic layer was then removed, and its absorbance read at 520 nm.

## PQS extraction and thin layer chromatography (TLC)

*P. aeruginosa* was cultured as described in the pyoverdine and pyocyanin measurements section (above). PQS extraction and TLC was performed essentially as described in ref. 121. Briefly, 300 μL of culture was extracted with 900 μL of acidified ethyl acetate. 450 μL of the organic phase was completely evaporated until its contents were dry on the benchtop and then subsequently resuspended in 10 μL of 1:1 ethyl acetate:acetonitrile. TLC plates were prepped by soaking in them in 5% $KH_2PO_4$ for 30 min and activated at 100 °C for one hour. 2 μL of PQS extracts and 1 μg of purified PQS as a control were spotted onto the plates and resolved using 17:2:1 methylene chloride:acetonitrile:1,4-dioxane. Plates were visualized and photographed under UV light.

## Intrinsic tryptophan quenching

A 200 μL purified full-length PqsA or the PqsA N-terminal domain (PqsA$_{NTD}$) in buffer (50 mM Tris-HCl pH 8.0, 150 mM NaCl) were added to a 96-well, all black microtiter plate (non-treated, Thermo Scientific Nunc F96 Microwell) at a fixed concentration of 5 μM, respectively. A buffer-only control was likewise prepared. The same volume of ArqI stock solution (1160 μM in 50 mM Tris-HCl pH 8.0, 150 mM NaCl) was titrated in both the buffer control and protein sample to generate 17 data points with concentration ranging from 0 to 59.4 μM. The mixture was allowed to equilibrate for 15 min after every addition of ArqI before reading. A BioTek Cytation 5 plate reader (Agilent) was used with an excitation wavelength of 295 nm and emission scan setting of 320-400 nm according to ref. 61. The generated protein fluorescent traces were corrected by subtracting the corresponding ArqI-titrated buffer traces. Change in maximum fluorescence intensities at 332 nm (PqsA$_{FL}$) and 330 nm (PqsA$_{NTD}$) were plotted against ArqI concentration. Equilibrium constants ($K_d$) were calculated by fitting the plots in GraphPad Prism 10 using the Binding Saturation: One Site-Specific binding model.

## Dansylation of ArqI and quenching by PqsA addition

Purified ArqI wild-type hexamer for dansylation and quenching were carried out as described in ref. 122. Protein buffer was exchanged to 10 mM MOPS, pH 8.4. A 50 mM stock solution of dansyl chloride (Sigma-Aldrich, MO, USA) in DMSO was used. To 100 μM ArqI, an equal concentration of 100 μM dansyl chloride solution was added. The reaction was incubated at room temperature, covered with aluminum foil, and gently agitated using a rotary mixer for 1.5 h. The reaction was then quenched by buffer exchange with 10 mM Tris-HCl pH 8.0 using an Amicon centrifugal filter with a 10 kDa MWCO (MilliporeSigma, MA, USA). The dansylated ArqI (DNS-ArqI) was then further purified using a Superdex 75 size-exclusion column. Binding Assay DNS-ArqI with PqsA$_{NTD}$: A 10 μM stock solution of dansylated ArqI (DNS-ArqI) was prepared in 50 mM TrisHCl pH 8.0, 150 mM NaCl. PqsA$_{NTD}$ solutions with concentrations ranging from 0 to 40 μM were also prepared in the same buffer. For the microplate binding assay, 50 μL of DNS-ArqI and 50 μL of each PqsA$_{NTD}$ solution were mixed, resulting in a final concentration of 5 μM DNS-ArqI and 0 to 20 μM PqsA$_{NTD}$. The mixtures were incubated for 1 h at room temperature. Fluorescence measurements were performed using a BioTek Synergy H1 plate reader (Agilent Technologies, CA, USA) with an excitation wavelength of 330 nm and an emission range of 400–630 nm. The change in fluorescence intensity at the peak wavelength of 512 nm was plotted against PqsA$_{NTD}$ concentration to generate the binding curve. The affinity constant ($K_d$) was determined from three replicates using the One-site Specific binding model in GraphPad Prism (GraphPad Software, MA, USA).

## Sepsis infection model

All procedures were performed in accordance with the guidelines approved by the Loyola University Chicago Animal Care and Use Committee as described in protocol #2021016. Female BALB/c mice aged 6 to 8 weeks were purchased from The Jackson Laboratory. Mice were housed under ABSL-2 containment at the Loyola University Chicago Comparative Medicine research facility. *P. aeruginosa* isolates were subcultured from overnight growth at a 1:10 dilution for 3 h in MINS media and were adjusted to 1 x $10^6$ CFU/50 μL in phosphate-buffered saline (PBS). Mice were restrained using a TV-150 Tailveiner® (Braintree Scientific). A lateral caudal vein was dilated with a heat lamp, and 50 μl of the bacterial suspension was administered intravenously using a 30-gauge needle. After 24 h, infected mice were euthanized by carbon dioxide inhalation. Organs were excised, weighed, and homogenized in PBS using a Benchmark® Bead Blaster 24. Bacterial burden was enumerated by plating serial dilutions on LB agar containing 5 μg per mL of irgasan to select for *P. aeruginosa*.

## Bioinformatics

Sequence alignments were generated by using Clustal Omega[123] and uploaded into EScript 3.0[124] for visual output.

To identify the conservation of residue Arg49 we downloaded all ABM domain sequences in the UnitProt database (PF03992) and then aligned them using the HMMER Biosequence online analysis tool[125]. These sequences were then further curated to eliminate duplicate species using BioPython and R. Sequences were then further aligned using MAFFT v7.388[126] and tree files generated using FastTree v2.1.11[127], curated to add (NCBI) taxonomy IDs, converted to Newick format and uploaded for display using iTOL[128]. Results produced 441 ABM-domain containing sequences from unique species that possess an Arg49 equivalent. ABM domains with Arg49 equivalent residues were found predominantly in bacterial phyla, but also some fungi, Archaeal, algal, and animal phyla. Arg49-containing ABM domains are listed in **SI**.

To examine the *arqI-gloA2* operon and surrounding gene genomic synteny, the Basic Local Alignment Search Tool (BLAST)[129] was used to query the ArqI nucleotide sequence for homologs in a collection of 444 *P. aeruginosa* complete genomes obtained from the Pseudomonas Genome Database[130]. Thirty-five hundred nucleotides flanking the "subject start" and "subject end" BLAST hit indices (encompassing genes PA0705 through PA0714 in strain PAO1) were used as coordinates to generate trimmed Genbank files for each of the 444 strains using genbank_slicer.py[131]. Identical and variable regions across all trimmed Genbank files were identified using the programs SPINE and AGEnt, respectively[132]. The distribution of these variable regions across all strains was determined using ClustAGE[133]. Jaccard similarity calculations were performed on the ClustAGE output using R Statistical Software (v4.2.1; R Core Team 2022)[134] with the vegdist command from the vegan 2.6-4 package[135]. Strains with a Jaccard similarity index > 0.9 were grouped together as having the same synteny. A gene cluster comparison figure was generated from representative strains using Clinker[136].

## Reporting summary

Further information on research design is available in the Nature Portfolio Reporting Summary linked to this article.

## Data availability

PDB codes for structures are available in Supplementary Table 1 and have been deposited in the RCSB Protein Data Bank (https://www.rcsb.org). Mass spectrometry data have been deposited in ProteomXchange partner, MassIVE (massive.ucsd.edu) under identifiers MSV000094205 and MSV000097777. The RNA-seq data generated in this study have been deposited in the SRA database under accession code PRJNA1256497. Source data are provided with this paper.

## Code availability

ABM domain sequences containing an Arq49 residue were curated using BioPython and R. Code has been deposited in Zenodo (DOI: 10.5281/zenodo.15311339), https://zenodo.org/records/15311339. The

code used for the FRAP experiments can be found at: https://github.com/BrattonLab/arqi-frap.

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

## Acknowledgements

Funding for this work was provided by NIH NIAID grant R01AI135960 and NIGMS grant GM141230 to A.T.U. This work was supported in part by two National Science Foundation (NSF) awards through the Center for the Physics of Biological Function (PHY-2210346 and PHY-1734030) to J.W.S., B.P.B., and D.V.M. Partial support of this project was provided by VUMC Discovery Scholars in Health and Medicine Program to B.P.B. In addition, this work was supported by the NIH grant 5R01AI163119 to J.P.A. and NIH grant R01EY034239 to A.R.K., and the Human Frontier Science Program grant (RGY0080/2021) to B.S.T. Research was also supported by NIH R35-GM118108 to A.M. We acknowledge the help from the Northwestern University Structural Biology Facility. Support from the R.H. Lurie Comprehensive Cancer Center of Northwestern University to the Structural Biology is acknowledged. Research reported in this publication was supported by the Maximizing Investigators' Research Award (MIRA R35) from the National Institute of General Medical Sciences (NIGMS) of the National Institutes of Health (NIH) #R35GM138183 to J.R.B. We would also like to acknowledge NIH grants HL136737, HL172492 and HL175964 to J.A.K and the American Heart Association grant 35170045 to T.G.M. We acknowledge NIH grant T32 HL105346 to L.W.R. We also want to thank Loyola University Chicago for funds supporting this research. We thank Dr. Peter Larsen at the Loyola Genomics Facility for helpful discussions with RNA-seq data and statistics. We thank Benjamin Katz at the UCI Chemistry Mass Spectrometry Facility, and also the staff at the Advanced Light Source at Berkeley National Laboratories (ALS) and the Stanford Synchrotron Radiation Lightsource (SSRL). We would like to acknowledge the Program in Lung Biology at the University of Alabama at Birmingham. This work is dedicated to the memory of Dr. Bernard ("Bernie") Weisblum.

## Author contributions

C.J.C., D.G.G., B.J.C. and M.T., D.V.M., B.P.B., J.W.S., B.S.S., A.G., J.R.B., A.F.M., A.R.K., J.P.A., C.W.G., J.W.S. and A.T.U. designed the research. C.J.C., D.G.G., B.J.C. M.T., D.V.M., B.P.B., D.E.S., V.L.T., T.G.M., L.W.R., M.C.M., A.F.M., J.P.W., A.R.K., J.P.A., C.P. performed the research. A.T.U. conceived the project and wrote the original draft. C.J.C., D.G.G., B.J.C., C.W.G., J.P.A. contributed to writing the manuscript.

## Competing interests

The authors declare no competing interests.

## Additional information

[1]Department of Microbiology and Immunology, Loyola University Chicago, Maywood, IL, USA. [2]Department of Molecular Biology and Biochemistry, University of California, Irvine, Irvine, CA, USA. [3]Department of Medicine, Division of Pulmonary, Allergy and Critical Care, Department of Medicine, University of Alabama at Birmingham, Birmingham, AL, USA. [4]Lewis-Sigler Institute of Integrative Genomics, Princeton University, Princeton, NJ, USA. [5]Molecular, Cellular and Developmental Biology, Yale University, New Haven, CT, USA. [6]Department of Pathology, Microbiology and Immunology, Vanderbilt University Medical Center, Nashville, TN, USA. [7]Department of Cell and Developmental Biology, Vanderbilt University, Nashville, TN, USA. [8]Vanderbilt Institute for Infection, Inflammation, and Immunology, Nashville, TN, USA. [9]Department of Physics, Princeton University, Princeton, NJ, USA. [10]Department of Molecular Pharmacology and Biological Chemistry, Northwestern University, Chicago, IL, USA. [11]Center for Structural Biology Facility, Northwestern University, Chicago, IL, USA. [12]Department of Cell and Molecular Physiology, Loyola University Chicago Stritch School of Medicine, Maywood, IL, USA. [13]Department of Plant and Agroecosystem Sciences, University of Wisconsin-Madison, Madison, WI, USA. [14]School of Life Sciences, University of Nevada Las Vegas, Las Vegas, NV, USA. [15]Department of Chemistry and Biochemistry, University of California, Los Angeles, Los Angeles, CA, USA. [16]The Pasarow Mass Spectrometry Laboratory, David Geffen School of Medicine, The Jane and Terry Semel Institute for Neuroscience and Human Behavior, UCLA, Los Angeles, CA, USA. [17]Department of Biology, Loyola University Chicago, Chicago, IL, USA. [18]Department of Molecular Biosciences, Northwestern University, Evanston, IL, USA. [19]Department of Pharmaceutical Sciences & Molecular Biology & Biochemistry, University of California Irvine, Irvine, CA, USA. [20]Present address: Department of Pathology, Microbiology and Immunology, Vanderbilt University Medical Center, Nashville, TN, USA. [21]Present address: Department of Structural Biology, St. Jude Children's Research Hospital, Memphis TN 38105, USA. [22]Present address: Department of Molecular, Cellular, and Developmental Biology and BioFrontiers Institute, University of Colorado Boulder, Boulder, CO, USA. ✉e-mail: atulijasz@uabmc.edu

