## [Transparent Peer Review file · Nature Communications]

A glyoxal-specific aldehyde signaling axis in *Pseudomonas aeruginosa* that influences quorum sensing and infection

Corresponding Author: Dr Andrew Ulijasz

Version 0:

Reviewer comments:

Reviewer #1

(Remarks to the Author)

Our comments to the authors responses have been prefaced with ###

Reviewer #1 (Remarks to the Author):

NMICROBIOL-24030754

Summary

This is a fascinating story on the identification of a glyoxal (GO) responsive pathway in the Gram-negative bacterium *Pseudomonas aeruginosa* (Pa), a widespread opportunistic pathogen. The authors hypothesized that Pa encounters GO during infections and sought to look at the transcriptional response to GO. The authors found several loci had altered expression upon GO exposure, with the most strongly induced gene being PA0709 (renamed ArqI). Remarkably, this locus was specifically induced by GO and not closely related aldehydes including methylglyoxal (MG). The authors solved the structure of ArqI and found two posttranslational modifications (PTM) including BME and GO itself on specific residues. They further showed that ArqI represses PQS synthesis and also can localize to bacterial poles. arqI is co-expressed with gloA2, which appears to be needed for resistance to GO; furthermore, the data presented suggest GloA2 is the prime source of resistance against GO.

The most interesting aspect of this story is the identification of a promoter that specifically responds to GO and that the produced protein, ArqI, can itself get a GO PTM. The authors have collected a lot of data but didn't quite succeed in figuring out what ArqI actually does. The authors claim ArqI is required for expression of pqsA as well as inhibiting activity of its gene product. They provide some compelling data that ArqI and PqsA interact and that ArqI can go to the poles in a GO-inducible manner. These are mostly descriptive and don't give a picture of what is going on with this protein or how it makes Pa less fit in a sepsis infection model. There is a lot of speculation in the Discussion that goes back to the GO RNA-Seq data but none of the ideas presented here was tested, and none can be conclusively linked to ArqI itself.

We would first like to thank this and the other reviewers for spending valuable time to go over our manuscript in depth, which in our view has resulted in a greatly improved paper.

As the reviewer points out, we still do not fully understand the role ArqI plays in glyoxal toxicity and virulence. Because these in vivo/mechanistic experiments are quite complex and involve what amounts to another PhD project entirely, we will not pursue the in vivo mechanism at this time. This current embodiment of work encompasses 6 years of research and in our view gives an important step in understanding GO-specific responses, and characterizes a novel PTM and protein system involved in the GO response. How these responses are induced by the immune system in vivo (if it is in fact the immune system) is (again in our view) beyond the scope of these studies.

An RNA-Seq analysis of the arqI mutant would likely be incredibly informative to help begin to understand why this locus is

important for pathogenesis, and now GO might affect downstream gene expression.

As suggested by the reviewer, we did do this experiment, but it revealed little changes in RNA levels between WT and mutant strains with GO addition. This could be due to how ArqI is posttranslationally modified and therefore might not be easily detected by looking at RNAseq differences. Another explanation would be that ArqI is required during infection and need a complete host environment to enact its phenotypic roles.

Between WT and Δ arqI mutant, genes that exhibited the greatest changes in RNA specific to ArqI were a Molybdenum binding protein (upregulated 2.7-fold in the arqI deletion strain; PA3441), and two transporters close to one another in the genome; one of unknown function (PA5548; 3.8-fold) and one alpha-ketoglutarate (PA5530; 6.5-fold). BauA, involved in the degradation of beta-alanine (upregulated 4-fold in the arqI deletion). We are unsure of how these data would further support the current publication and so we have not included it in the updated manuscript

###We are satisfied with this

The title of the paper is somewhat over the top. The authors do not show in any way why this ArqI locus is important for virulence, which could be due to more than one thing.

To better reflect the major findings of the paper we have changed the title to:

"The glyoxal response in *Pseudomonas aeruginosa* reveals a novel, glyoxal-specific signaling axis that influences quorum sensing and infection."

###We are satisfied with this edit

Major points

While there is a lot of data in this manuscript, there are some statements that were over-reaching. The final sentence of their Intro makes very sweeping statements without enough data to back it up. The authors also need to more clearly explain what they think ArqI is actually doing given they cannot directly link its function to pathogenesis.

Noted - we have redone the manuscript considerably to dampen speculation and focus more on the findings, i.e. we show the bacterial response to GO and the molecular consequences of the ArqI-GloA2 induction mainly on the microbe side (save the sepsis data in Fig. 5).

Although we will not pursue the mechanism here, we talk about in the discussion section what ArqI might be doing in the context of PqsA regulation during infection, and especially how the host might use GO to kill bacteria/induce such as response.

The relevant discussion section parts are subheadings:

"ArqI as an inhibitor of PQS biosynthesis" – line 582

And the next subsection:

"Perspectives on the ArqI-GloA2 axis in the context of infection"

###We are satisfied with this edit

The introduction focuses a lot on what is known about GO production and detoxification in eukaryotes, with some minor references to this same knowledge in prokaryotes. I would have liked to see more of what's known of GO detoxification and its relevance in prokaryotes and *Pa* in particular. This would help add relevance to the study.

We agree and thank the reviewer for pointing this out. Most of the provenance with GO/MGO toxicity and signaling has come from eukaryotic research (as might already know). It is thus important to establish this for a broad readership. We agree there was not enough about the prokaryotic literature and have therefore added onto the introduction descriptions of prokaryotic data. For example, describing the NEM system (starting with line 111).

###We are satisfied with this edit

Opening the story with the general GO response takes away from the main point of the paper, which is the characterization of the most strongly regulated genes, PA0709-gloA2 and the gene product ArqI. Given that there's no data following up the GO-induced transcriptome beyond the arqI-gloA2 locus, it doesn't seem appropriate to front-load the paper with these data.

Along these lines, how GO affects the general transcriptome is very hand-wavy, which is fine, but this discussion belongs in the Discussion. In particular there is emphasis on sulfur and phosphate starvation, taurine uptake and ROS mitigation as measures used to minimize GO damage or respond to GO.

We have now largely attenuated the global RNA-seq Intro discussion and moved much of it to the Discussion section. However, we did leave text in the results to go over the more salient data with the RNA-seq as this is (to the best of our knowledge) the first comprehensive GO-specific RNA-seq study in a microbe. In our opinion, there are some novel and

interesting points here to make before focusing on the main operon – arqI-gloA2.

###We are satisfied with this edit

Was there overlap in the 25 min and 1 hr transcriptome? A table that categorizes the RNASeq results would be helpful for the long lists provided in lines 111-149.

Yes, there was both significant overlap (104 genes) and significant differences. We now provide a table as suggested that denotes the unique GO-exposed 1 hour timepoint genes, and also overlap between the 15 min/1 hour timepoints. We have added this text (line 147):

Line 147:

“Several genes were differentially expressed with GO treatment: 104 differentially expressed genes were shared between the 15 minute and 1-hour timepoints and 43 genes were unique to the 1-hour or "late expressed" timepoint. Surprisingly, in many cases the 15 minute and 1 hour GO treated data exhibited opposite expression of certain genes/operons, indicating a truly dynamic response and metabolic readjustment to GO remediation over time (Fig. 1a; Supplementary Information (SI)). STRING 1 was used to compare related gene families that were differentially expressed greater than 3-fold (Extended Fig. 1b-g; SI).”

We have also included some sentences within the RNA-seq description that denotes 15 min versus 1 hour timepoints such as:

Line 179:

Upon GO exposure, a global repression in iron uptake and utilization pathways was also observed, which included siderophore (pyoverdine), Fe³⁺-PQS transporter (fptA), and pyoverdine synthesis genes pvdH (PA4221; only at the 1-hour timepoint - see SI), ferric iron uptake operon hitAB (PA4687/PA4688), ferrous iron-dependent dioxygenase piuC, heme synthesis and catabolism genes (hemN and hemO, respectively), and bacterial ferritin bfrB. Additionally, we saw a strong downregulation of the phosphate uptake and sensing pst system 2,3 (at the 1-hour timepoint only), as well as molybdenum uptake icmP genes 4.

###We are satisfied with this edit

Fig 3 b and c are confusing. Fig 3b shows the fluorescence data for strains expressing mutant forms of ArqI. The representative images depict R49Q migrates to the poles and P87G may migrate to both poles. Yet quantification in Fig. 3c shows no difference between R49Q and P87G as its plotting the percent of population with foci. I think the quantification muddies the fact that there appears to be different migration patterns between the two ArqI mutants. Further, it is unclear if the heatmap in Fig. 3b is a collective average of all sampled cells or a representative image.

Fig. 3c is an average of representative cells which were selected by eye (50- 500; see Materials and Methods section).

As for the quantification of percent cells with foci (Fig. 3b), that quantification is only the percent of cells within the entire population with at least one foci. The program we wrote for this searches for foci that exceed a certain intensity threshold- which was set intentionally by us. Then the program takes the number of cells that have at least one foci and divides by the total number of cells detected. Therefore, the quantification does not discriminate on the spatial localization within the bacteria or the number of foci, but rather solely if a cell has at least one or not. This difference in quantification versus visual selection in Fig. 3c might account for the differences seen. The main point as we see it is the R49Q mutant exhibits different migratory behavior than wild-type, but not to the extent of the R16A or P87G mutants.

To better reflect these differences the X axis of 3c has been changed to: "percent of population with at least one foci." Differences between R49Q and wild-type can also be seen in the FRAP experiment (Fig. 3E).

We also added this description to the text:

Line 350:

“After we performed a more unbiased assessment of polar localization over time that incorporates the entire population of cells rather than selection by eye (see Methods), it became evident that the R49Q mutant was indeed impaired in its ability to localize to the pole in comparison to wild-type ArqI (Fig. 3c).”

###We are satisfied with this edit

Fig. 4: Protein-protein interactions and impact on gene expression: The authors make the argument that ArqI represses PQS biosynthesis by repressing pqsA, but also that ArqI represses function by binding to PqsA. The yeast 2 hybrid experiment was described as a "screen", but there was no screening involved. Rather, it seems Y2H was used to look at protein-protein interactions. However, there was no actual Y2H data shown from what I can tell and nothing in the methods. The reader is referred to Fig. 4 g but that just has a schematic of the domains used for Y2H analysis (also, there is inconsistent formatting throughout the text that should be fixed. For example, "Y2H" and "YTH" are both used for yeast two hybrid). It seems the

commentary about these absent data should be removed as it doesn't seem to add much to the story.

The Y2H screen was performed by a company (Hybrigenics) in possession of the library. We have removed the commentary, as suggested by the reviewer. However, it is essential that we state the origin of our interaction findings. We have changed all to read "Y2H" for consistency. The relevant text begins at line 389.

###We are satisfied with this edit

Along these lines, the BioID experiments are not well presented. Given that bacteria, including Pa, have their own BirA proteins that are usually the dominant thing to be detected with streptavidin, I was confused as to how these experiments were done so cleanly. I think a schematic or just better description of how these were done would be useful for most readers. Furthermore, line 1078 "...verification..." isn't really what was done. I think you were testing or looking for the interaction? Overall, the Y2H and BioID experiments seemed somewhat "clunky" and not particularly informative if the relevance of the interaction between ArqI and PqsA isn't determined. Maybe it could just be presented better.

The reviewer is spot on. And we are well aware if intrinsic Biotin Ligases, which has been the reason (before this paper) that no one used this method in prokaryotes. To get around this issue we first immunoprecipitated (IPed) PqsA. In this way when we blot for biotin (second step), the background biotin PTMs do not show up (as they were washed out in the initial IP of PqsA) and cloud the interaction results. We choose this method because most protein-protein interactions are quite weak, yet still biologically relevant, and this method takes advantage of its high sensitivity and ability to identify interacting partners within the low to mid micromolar Kd range (the range of this precise interaction).

A new schematic has been provided (Fig. 4J, above) as the reviewer suggests to help readers understand how the interaction was shown.

###We are satisfied with this edit

Fig. 5: The term "clean deletion" isn't really useful and is rarely accurate (see the supp table of strains).

The term "clean deletion" has been replaced by "deletion" throughout the text.

###We are satisfied with this edit

Given that it wasn't completely complemented supports the possibility of polar effects.

To the review's suggestions we have re-done our GO toxicity on these strains with enumeration (Fig. 1C, D).

###We are satisfied with this edit

Lines 319-322 discussion about the incomplete complementation isn't really correct. While I agree that incomplete complementation could be due to effects from variations in expression or even just having the complementing allele encoded elsewhere in the chromosome, one can't connect the in vitro GO sensitivity levels to in vivo defects.

While creating complements for the in vivo studies we were concerned with the plasmids being lost during sepsis w/o proper antibiotic selection, so we opted for a chromosome, single copy integrative strategy. We went back and tested these complements and determined they were basically not complementing (based on GO toxicity resistance). The reason for this likely a lack of expression of the gene at the foreign – which is common in *P. aeruginosa* genetics. Indeed, the CTX site for complementing has had issues before for many other labs and, like most complementing systems is not perfect.

Therefore, we now simply provided data only on the operon deletion (arqI-gloA2) versus wild-type w/o complements in Fig. 5. Although not ideal, we believe this show that this operon and/or downstream effects of this operon are critical to infection.

###We are satisfied with this edit

It is likely that ArqI has other affects unrelated to GO resistance in vivo (which is almost certainly the case).

Yes, we agree to this statement. One very likely possibility its effect on PQS, which before this work was not known to be associated with GO / aldehydes

.

Furthermore, complementation with gloA2 alone had the most GO resistance (fig. 1c) and while this as explained by higher gloA2 expression levels in the complemented strain (extended fig. 3b), it was not quantified at all; the text says that there was "partial restoration of expression levels" but there's no expression data (mRNA) presented.

See comments above on complements.

There are immunoblots for tagged, recombinant proteins but not endogenous ones, so one can't comment on levels of anything.

We agree and have removed any commentary suggesting we can see these differences.

###We are satisfied with this edit

Rather, the data in fig. 1c shows complementation with gloA2, and not arqI, is what's important for GO resistance.

Yes we agree. ArqI does not show any evidence as to being involved in GO resistance (in vitro) as we state many times in the text. New data in Fig. 1C,D proves this, at least in rich media at a single timepoint. It might be doing something else during infection which will have to be demonstrated as part of a future project

###We are satisfied with this edit

It also isn't clear why the authors didn't complement their deletion strain with untagged versions of their genes. Is there a reason why complementation wasn't done with untagged genes?

Yes, we tagged proteins because in the absence of a ArqI antibody we could not measure protein levels of the complemented strains. Measuring protein levels in addition to RNA levels we think is crucial to interpretation of data, but as the reviewer points out a comparison to WT (protein levels) cannot be made here.

###We are satisfied with this edit

The tags could have effects on function.

In our experience short 6-mer peptides rarely influence function. However, it cannot be ruled out. One must have a way of measuring protein levels in the absence of an antibody and tagging the protein is really the only way to do this.

Importantly, we have now added new data: a tag-less ArqI complement to Fig. 1C,D and there still had no effect on GO resistance. If you look at the ArqI structure the N and C termini are positioned well to accommodate a tag, and is therefore not surprising that a short peptide has no effect on phenotypes or polar migration (for the latter we attached a 26 kDa (gfp) protein).

We have therefore added the additional text:

Line 364:

"Importantly, these data demonstrate that the GFP fusion did not affect hexamer formation in vivo, and therefore gave confidence that other, far smaller epitope tags we have used in this work (e.g. FLAG, HA and His tags), would likely not influence ArqI oligomeric formation nor function either."

###We are satisfied with this edit

Importantly, the parental/wt and mutant strains should also carry empty vectors as controls for plasmid effects on various phenotypes. It doesn't seem that this important control was done and could have helped with the interpretation of the data.

We apologize for not making the controls clearer. Many strains labeled in this work as 'wild-type' are actually empty vector controls. The nomenclature was originally there depicting this but was regrettably removed for clarity (it obviously had the opposite effect). To remedy this, we have since changed these WT labels to 'WT EV' (for Empty Vector) to denote the actual strain (see Fig. 1). "Empty vector/plasmid" is/was used in Fig. 4, for example.

###We are satisfied with this edit

Given the current state of the mouse data, it is impossible to conclude that arqI is needed for virulence.

We now focus on WT versus the aqI-gloA2 (operon) deletion. This operon is clearly required for infection. Our collaborator Jon Allen developed the sepsis assay in the lab of Alan Hauser (NW University) and told us the rarely see such as obvious phenotype due to the usual high variability of infection data with *P. aeruginosa*. Of note: With our deletions there are no genetic remnants of the original integrated vector.

The statistical analysis on the animal data is unclear. Were the other strains/complemented strains significantly different from either the parental or mutant strains? Again, if all of the strains were plasmid transformed, the data may have been more robust.

As per above we did not include the complements in this version. Statistics were calculated as per PMID 32156726 (Drs. Jon Allen/ Allen Hauser protocol) and as mentioned in the text.

###We are satisfied with this edit

Some of the language should be softened. For instance, line 313 states "the arqI-gloA2 axis was involved in host cell detection, virulence, and survival" but there is no data that addresses 'host cell detection'. The authors should focus on the strongest data they found.

In general, we have greatly softened (and extensively rewritten) the statements regarding infection and virulence, saving much of this for the discussion section. We will pursue in vivo mechanistic studies in the future.

###We are satisfied with this edit

Minor points

Abstract: The statement that GO is a "novel inducer of virulence" is a bit much, and I don't think it adds anything to the story. "Virulence" is a system/host-dependent term.

We have largely downplayed the original statements claiming this connection and saved this for the discussion section. We do see an induction of one of the two T2SSs that *P. aeruginosa* possesses in response to GO, but this observation only serves as a discussion item now. We have focused the text more on the novelty of a GOspecific response and the paper findings.

###We are satisfied with this edit

line 87: "amino acids" can be deleted.

Done

###We are satisfied with this edit

line 94: MGO can also form spontaneously from DHAP.

we added this (line 104): ", or alternatively can form from DHAP spontaneously"

###We are satisfied with this edit

line 111: "clear" shouldn't be used

This text has been reworked as part of the discussion as suggested. "Clear" has been removed.

###We are satisfied with this edit

line 106 states *Pa* is treated with sub-MIC concentrations of GO but there are no MIC data to support this claim. Along these lines, MG treatment is at 4 mM; was this also sub-MIC? How was this concentration selected?

We have now provided new MIC data with both GO and MGO (Fig. S1A), showing that 4 mM is well below the calculated GO MIC of 12.5 mM.

###We are satisfied with this edit

line 139: "these responses make sense" sounds a bit glib.

changed to (line 186): "Collectively, these responses fit well into our overall hypothesis,"

###We are satisfied with this edit

line 169: "GloA protein" is redundant. And it should be just *gloA* (italicized gene that is expressed).

Fixed.

###We are satisfied with this edit

line 164: should read "...was highly specific to GO."

Line 228 now reads: "These results show that gloA2 plays a highly specific role in GO remediation. On the other hand, at least under these in vitro conditions arqI was shown to have no role in GO resistance."

###We are satisfied with this edit

lines 211-223: glyoxal was spelled out throughout but it is abbreviated everywhere else. SEC only needs to be abbreviated once (see line 230 where it is redefined).

These have been fixed.

###We are satisfied with this edit

line 348 HCl, not HCL

Fixed.

###We are satisfied with this edit

Line 343-344 cites ref. 13 evidence suggesting the taurine dioxygenase homolog ALKBH7 is protective against glyoxal toxicity. Ref. 13 (Kulkarni et al.) instead finds that loss of ALKBH7 confers protection against GO.

We have corrected this with the following text:

Now line 522:

"Interestingly paralleling these functions in prokaryotic systems, in humans the mitochondrial equivalent of TauD, α -KG dependent dioxygenase Alkb homolog 7 (ALKBH7), has also recently been implicated in the regulation of GO metabolism. Indeed deletion of ALKBH7 results in elevated glyoxylase (GLO-1) levels; reversing cellular necrosis and cardiac injury 5, and has been shown to reduce glycation of diabetic patients 6. Although there is currently a lack of experimental evidence, is it nevertheless intriguing to think that pathogens might be hijacking host taurine to blunt aldehyde toxicity already experienced by host cells."

###We are satisfied with this edit

Throughout: Genes should be italicized in the figures. Also, genes are "expressed", not proteins; proteins are "produced". Lines 142- : "several virulence genes (not factors)" *vreI* and *vreA* should be italicized with lower case "v" as they are genes you are talking about. There's a bit of wordiness throughout. The notation of mutants v. complemented strains is not clearly presented in the figures because untraditional labels were used. Proper ASM nomenclature guidelines are a good place to start for standard usage.

We have made these changes/corrections throughout.

###We are satisfied with this edit

Centrifugation isn't "xg", it is usually just "g" (ital). Also the letter x should be replaced with symbols where appropriate.

Have changed all to "g"

###We are satisfied with this edit

Fig 4 b and c should have stats comparing empty plasmid + arabinose to FLAG-ArqI + arabinose, not the current empty plasmid minus arabinose to FLAG-ArqI + arabinose

Doing this (by Two-Way ANOVA with Šidák's multiple comparisons test) results in:

Pyoverdine (4B): no significance in EC 0% vs 0.2% arabinose and $p < 0.001$ (***) Pyocyanin *4C): no significance in EC 0% vs 0.2% arabinose and $p < 0.01$ (**)

These have been added to the figure.

###We are satisfied with this edit

Fig. 6 is overly complicated and should consider numbering sections as it is read in the figure caption.

Great suggestion - thank you. This figure has now been redone with numbering and accompanying legend description.

###We are satisfied with this edit

Extended fig. 5 e,f: I am not convinced this shows a dose dependent response to GO. In 5e there appears to be less ApqI with 2 mM GO.

Agreed, however the higher molecular weight bands are more prominent with 2 mM GO, possibly indicating that at this high concentration there could be higher order proteins being formed (though GO linkages to ArqI interacting proteins). Nevertheless, we have changed the text to read "trending".

###We are satisfied with this edit

Extended fig. 6d should be log scale.

Now Fig. 6e; we have done this and replaced the old figure. #

###We are satisfied with this edit

Line 280 states a 'significant decrease in PQS' and calls out fig. 4f, but there are no stats. Should not say significant in this case.

We have added mutant versions of ArqI to this figure along with statistics.

###We are satisfied with this edit

Several figure panels are pixelated, missing axis labels, or missing stats. Figure legends are not consistent with figure panels. Please verify completeness of figures.

We have ensured these things have been done throughout.

###We are satisfied with this edit

Some examples below:

o Fig. 1 legend mentions stats for panels D and E but there are no stats.

We have added the stats to these figures and description of stats in the legend.

###We are satisfied with this edit

o Fig. 2 a,b is pixelated

o Extended fig. 1 a-d is pixelated

To our knowledge the version that we uploaded was not pixelated, so we are unsure why this has occurred. We will ensure the final figures will not be and apologize if the figures were not legible enough to be properly reviewed.

###We are satisfied with this edit

o Extended fig. 3 g inset, h are missing axis labels

Figure Extended figure 3g has been re-done and 3h has been removed.

###We are satisfied with this edit

Ext. Fig. 1, 3c is unreadable.

Unfortunately, the lettering is as large as we can make it. However, the resolution high enough where you can

zoom in for details.

Figure 3c should be readable from looking at the original pdf submission. We are unsure why this is.

###We are satisfied with this edit

Reviewer #2 (Remarks to the Author):

The manuscript by Corcoran et al., "The glyoxal response in *Pseudomonas aeruginosa* controls quorum sensing, virulence, and infection through a novel regulatory axis," describes the *Pseudomonas aeruginosa* response to glyoxal, a toxic metabolic byproduct of cellular processes by conducting RNAseq transcriptomics in vitro. The in vitro transcriptomic result highlights the glyoxal induction of arqI-gloA2 operon along with differential gene expressions in sulfur catabolism/replenishment, ROS detoxification, Iron uptake/utilization, and selective number of genes involved in virulence. The authors clearly show GO resistance by GloA2, supported by the spot-killing promoter activity of the arqI-gloA2 operon in response to GO exposure. Moreover, the authors have provided strong structural evidence of ArqI quaternary structure (crystallography), oligomerization, and post-translational modification. In addition, the report shows the ArqI connection to the quorum sensing system, MvfR/pqs, by showing the interaction of ArqI with the PqsA, an enzyme of the biosynthetic operon pqsABCDE. The manuscript is well written, and overall, experiments are well organized and presented. My chief concern is the superficiality of the data presented regarding the authors' claims about quorum sensing. The study offers insufficient data on the ArqI relation to QS and the impact of this relationship despite the demonstration of direct interactions between ArqI-PqsA and PQS detection by western-blot up arabinose induction. This part of the work feels disconnected from the rest of the work presented. In any case, PQS is not abundantly produced in human tissues.

Whether ArqI-PqsA interaction impairs PQS biosynthesis and, most importantly, the PqsR/MvfR QS circuit to cause virulence attenuation needs further evidence such as mechanistic analyses, the use of QS mutants, bacterial burden analysis using more appropriate infection models, mortality studies, etc.

For example, in vivo data presented beg the question of why, in mice infected with the ArqI mutant, the CFUs significantly decreased since PqsA is not inhibited.

First, I want to thank this and the other reviewers for taking so much time to critic the manuscript – which will undoubtedly make it a better paper.

We have found that our complements are were not complementing GO toxicity and have therefore removed these from Fig. 5 data. We have added data from other tissues as suggested by the reviewer. We think that the reason behind the arqI-gloA2 axis being required for infection might be due to gloA2 being required for aldehyde remediation (aldehydes we hypothesize are being produced by phagocytes in blood-rich organs).

The individual contribution of ArqI we are still unsure of, in the blood (sepsis) or otherwise (other tissues).

Due to a lack of more in-depth data in this work tying in PQS production in vivo to ArqI expression, we have toned down our rhetoric and refocused the PQS link as an ongoing story. Investigating the very interesting findings with the ArqI-PQS axis with various host cell lines and infection models is out of the realm of this study, as well as interrogating the interesting finding of ArqI controlling T2SS hux system will be the investigation of another upcoming work.

What is the ArqI-PqsA and ArqI-QS relationship in vivo?

This is again a question we will answer in detail in future studies.

To better illustrate how big this question is, and in our opinion why studying this properly would be a whole additional paper, is that it is currently unknown under what natural circumstances glyoxal induces the arqI-gloA2 axis or any other glyoxal responsive system, and therefore by association its presumed control of PQS synthesis (and by association other metabolites in the PQS biosynthesis pathway - as this reviewer points out) during infection. Presently, in order to induce glyoxal GO responses in *P. aeruginosa* (or any other microbe for that matter) you have to add quite a lot of GO externally to the system. From our data and data in the literature we now hypothesize in the discussion phagocytes could be the inducer and should be tested in the future.

Another concern is that *P. aeruginosa*'s response to glyoxal in LB is not representative of the bacterial host interaction. All

data in reference to *P. aeruginosa* response to glyoxal are performed in LB, a rich medium. I will have performed some of the experiments in co-cultures to take into consideration the host impact during infection. I have several comments that I hope will help improve the manuscript.

Although we agree that testing multiple conditions would have been more informative, we went with LB for the following reasons:

1. The cells show consistent growth.
2. We did not have to choose a single minimal medium carbon source that could skew results, as *Pseudomonas* is able to use many different carbon sources.
3. We suspected that the glyoxal response happens regardless of medium used, its toxic to the cells.

Using co-culture is a brilliant idea. This experiment is currently being done with RNA-seq (duo-seq) and has/will produce a massive data set regarding host pathogen interactions. We anticipate these studies to be published in another publication due to the large amount of data generated.

The real experiment that will be done in the future will be using human blood/PMNs with luciferase reporter strains to test for GO-induced (and PQS-induced) induction, and then using these strains for in vivo whole animal imaging. Finding out the spatiotemporal expression of PQS during infection would be a next step/project.

###We are satisfied with this

Title

Line 1 – The title is misleading and needs to be rephrased. This study does not fully demonstrate that glyoxal response in PA controls the QS and virulence.

We agree and have changed the title to:

“A novel glyoxal-specific aldehyde signaling axis in *Pseudomonas aeruginosa* that influences quorum sensing and infection.”

###We are satisfied with this edit

Abstract

Line 68 – It should be mentioned in the abstract that the transcriptional response to GO in PA was analysed in vitro.

Results

Please explain the length of time used to expose *P. aeruginosa* to GO. Why were longer times not used? Growth inhibition, toxicity?

We have added new data showing the MIC of GO and MGO (Extended Fig. 1A). 4 mM was chosen due to it being under the MIC but sufficient for induction of the GO response. There is no growth inhibition over time at this concentration. We used 2 timepoints, 15 min and 1 hour and can see that most genes are expressed shortly after exposure (15 min) with some longer term at 1 hour. Interestingly, some genes are initially upregulated at 15 min and then downregulated at 1 hour. We have added additional text to address these differences (lines 147+).

###We are satisfied with this edit

Lines 75-76: The authors state, “ArqI migrates to the flagellar pole and shuts down biosynthesis of the critical quorum-sensing molecule PQS by forming a complex with PqsA.” Where it is shown that after the ArqI migration to the pole, a complex with PqsA is formed.

PQS inhibition by polar migration has, indeed, not been shown directly in this work. The abstract/text has been rewritten to better reflect the paper's data with conclusions. The way the original manuscript was written suggested something we did not intend (poorly worded sentence structure).

###We are satisfied with this edit

Also, please note that inhibition of PqsA shuts down PQS synthesis and multiple small molecules and virulence factors regulated by the MvfR/pqs QS system.

We have now addressed this and other concerns with a section in the Discussion titled: (line 582): ArqI as an inhibitor of PQS biosynthesis

###We are satisfied with this edit

Fig 1C doesn't provide CFU enumeration.

This has now been done and is shown in Fig. 1D.

###We are satisfied with this edit

Figure 2E: It is not immediately clear whether the glyoxal was co-crystallized with ArqI. Post-translational modification in ArqI that results from glyoxal exposure should clarify whether this modification is a direct result of the biochemical reaction between ArqI and glyoxal."

No glyoxal was added to the media - we simply saw the modification in the crystal when solving the structure (to our surprise). This indicates there must have been enough glyoxal in *E. coli* naturally to see some electron density in 3 of the 6 monomers in the hexamer structure. In essence we were lucky.

Figure 3B: Subcellular localization of ArqI is nicely shown. Misfolded proteins in the form of inclusion bodies or dispensable proteins localize at the pole. ArqI (WT) migration in the flagellar pole seems to be due to the absence of glyoxal, the substrate, therefore localizing at the pole.

We did do this experiment in the original submission, but it might have been missed because we did not emphasize it enough in the original text. Induction with GO allowed migration to the pole of ArqI (Fig. 3d), and this migration can also be achieved with artificial induction by arabinose (Fig. 3b).

###We are satisfied with this edit

The point mutations R16A and P87G indeed seem to be important residues that determine this phenotype. Thus, authors should reconsider their claim that ArqI is "blindly" being dispersed intracellularly in search of glyoxal.

We have added new data showing that these residues are not required for PQS inhibition (Fig. 4f), so it appears that oligomeric state and PQS inhibition are 'uncoupled'. The mechanism here we still do not understand, but new data we provide in Extended Fig. 7 shows that indeed both hexamer and dimer ArqI are predicted to similarly bind the PqsA deposited tetrameric structure – which fits nicely with the Fig. 4 data now. In the discussion section we discuss potential mechanisms.

Finally, we add new data (new figure: Ex Fig. 7h) showing (surprisingly) that ArqI interacting with PqsA does not interfere with the enzymatic activity of PqsA. We therefore propose (in the Discussion section) it could be binding PqsA and disrupting its interaction with another protein(s) – such as PqsB/C etc.

###We are satisfied with this edit

Authors may want to show the subcellular localization of ArqI (WT) in the presence of glyoxal.

As mentioned above this experiment was in the original submission. Figure 3d shows ArqI migrating to the pole after addition of glyoxal in the original submission. This was accomplished by creating a ArqI-mSCARLET-I fusion under the control of the native arqI-gloA2 promoter. We also use a control ABM domain (PA3390) that shares significant homology with ArqI and does not migrate to the pole (arabinose induction – Fig. 3c), nor inhibit PQS accumulation (new data; Fig. 4f).

###We are satisfied with this edit

Figure 4G: Please replace pqsDEBCH with pqsBCD. E is not involved in the synthesis of the small molecules, and pqsH is not part of the pqsABCDE operon.

Thank you for catching this. We fixed it.

###We are satisfied with this edit

Figure 4I: Utilizing BioID method, authors have shown ArqI-PqsA interaction in PA. It would be beneficial for the authors to show this during *in vivo* infection.

This is an excellent idea! However, since BioID has only been used in prokaryotes one other time (and in culture at that) to the best of our knowledge, this would take extensive research to troubleshoot and would be a great publication in-it-of-itself.

Roux, K. J., Kim, D. I., Burke, B. & May, D. G. BioID: A Screen for Protein-Protein Interactions. *Curr Protoc Protein Sci* 91, 19

23 11-19 23 15 (2018). <https://doi.org/10.1002/cpps.51>

Veyron-Churlet, R., Lecher, S., Lacoste, A. S., Saliou, J. M. & Loch, C. Proximity-dependent biotin identification links cholesterol catabolism with branched-chain amino acid degradation in *Mycobacterium smegmatis*. *FASEB J* 37, e23036 (2023). <https://doi.org/10.1096/fj.202202018RR>

Therefore, the suggested *in vivo* assay would have to be performed after a long period of trouble shooting and IACUC submissions/approvals that we believe are well out of the scope of this publication. In addition, it is not even known where and when PQS is expressed during infection, save some reports on its accumulation in chronic CF patients. Thus, to summarize we agree with this idea - but for a future study/work.

###We are satisfied with this edit

I Figure 5A: Authors present bacterial counts in the heart at 24 hpi after tail vein infection. Infecting the vein and measuring CFUs in the heart to conclude bacterial spread is questionable. Authors should adopt IP or SQ infection and present bacterial burdens in several organs (heart, spleen, gallbladder)

We have not adopted the IP or SQ model, but rather added data from other organs in the manuscript from the tail vein infection (gallbladder, spleen). Indeed, in the spleen a major difference is observed between WT and the *arqI-gloA2* deletion. We suspect the differences are due to phagocytes – and this will be tested for a future work describing *in vivo* mechanism.

###We are satisfied with this edit

I Figure 5A: There is a major concern about the authors' claim that ArqI has a "critical" role in the disruption of the QS system. Authors show a significant attenuation of heart CFU counts upon knocking out the *arqI*. The authors claim that ArqI shuts down the PQS biosynthesis. If so, how do you explain the clear attenuation of PA burden in the heart upon deleting *arqI*? In other words, in the absence of ArqI, the PQS biosynthesis should be intact (per the authors' claim), which would then fail to explain the attenuation of the burden supposedly due to the disruption of PQS biosynthesis.

It appears from the literature that the role of PQS during infection might be more tissue- and especially strainspecific. There are likely many signals feeding into the *arqI-gloA2* axis we are unaware of, and importantly, would almost certainly differ between tissues/organs etc. For example, ArqI might be expressed in two different organs, but only in one particular organ it might garner a PTM in the protein structure, which would then affect its activity - despite the protein being present.

As stated before, *in vivo* studies such as these would require years of research to really get at properly and is, in our opinion, outside the current scope of this publication (however exciting). For this current body of work, we have thus dampened claims on the QS connection with ArqI and fully intend to explore in detail the tissuespecific effects of PA infection and the influence of QS by ArqI in the future.

###We are satisfied with this edit

Minor I Line 117: list the genes of bacterial homologs to ALKBH7.

done I Line 119: "ssuD" to "ssuD". done

I Line 310: The sentence "...addition of GO-induced genes reflected..." should be improved. Did the authors "add" the "GO-induced genes"? Remove "addition of" We fixed the text to make more sense.

I Line 334: "it's" to "its".

Done

I Line 415: "...that would further exacerbate glyoxal formation" to "... that would inhibit further glyoxal formation".

Done

I Line 988: insert a space.

Done

I Line 1279: The figure legend says, "ArqI is required for robust colonization of the heart, lung, and spleen during bacteremia", but the figure only displays bacteremia in the heart. Remove "lung and spleen".

We have now added lung and spleen data and thus kept these statements in.

Reviewer #3 (Remarks to the Author):

In response to the manuscript “The glyoxal response in *Pseudomonas aeruginosa* controls quorum sensing, virulence, and infection through a novel regulatory axis”. The authors use RNA seq to identify a transcriptional glyoxal response, where they identify a two-gene operon highly upregulated – PA0709 and *gloA2*. They then use X-ray crystallography and other techniques to structurally characterise ArqI (PA0709), identifying that the protein is modified by glyoxal, and forms a hexamer. They finally show that the protein (usually) localises at the flagellar pole and inhibits PQS synthesis in relation to Quorum Sensing. Please find below my comments on this manuscript.

We would first like to thank this reviewer and others for meticulously going over this work to improve the manuscript. Many new figures and data have been added to address issues below.

One of the strong issues is the structural similarity between PA0709 and LsrG. In the fig. 6 schematic, PA0709 is postulated to inhibit PqsA and thus block quorum sensing. I find it somewhat disingenuous that the manuscript avoids discussion of the role of ArqI – to hydrolyze QS molecules. Have the authors considered/ruled out this possibility for PA0709. Either way, this needs to be a prominent feature of the analysis and the discussion.

We have tested the ability of PA0709 (ArqI) to influence PqsA enzymatic activity using an enzymatic assay with PqsA substrates (anthranilic acid plus CoA) and found that PA0709 addition had no influence in this process (new data Extended Fig. 7h). However, this does not rule out possible enzymatic activity of ArqI on PQS itself, other quinones or PQS precursors etc. At present we are screening compounds to identify a possible substrate of ArqI. We have also modeled LsrG substrates (DPD) into the ArqI presumed binding pocket (Extended Fig. 4c-f). The possibilities of a ligand for ArqI are now in the discussion section, starting with the section: ArqI as an inhibitor of PQS biosynthesis (line 582).

###We are satisfied with this edit

The manuscript has several lines of characterizing PA0709 – showing hexamer formation, polar localization and glyoxal modification – none of these play a role in the fig. 6 schematic, demonstrating to me that the authors haven't decided unpicked the significance of this characterization in terms of the phenotype.

We do not know the significance of the hexamer/dimer and polar localization at the moment, but ArqI and its inhibition of PqsA/PQS is shown in Fig. 6; affecting steps 1, 2 and 3 > and then subsequently many downstream steps. We hope with the new numbering system the role in the ‘big picture’ for ArqI/GloA2 is more clear now.

###We are satisfied with this edit

Line 122 – fix “e.g. ref”

fixed

###We are satisfied with this edit

Line 153 – indicate here at first instance that *gloA2* is PA0710.

Done

###We are satisfied with this edit

Line 179 – the description of symmetry “trimer of dimers” is an informal way of describing D3 symmetry. As the structure displays true D3 symmetry, this should be stated using this language.

We have changed to this: “The crystal structure of ArqI revealed a D3 symmetry (“trimer of dimers” hexamer) (Fig. 2a; Extended Fig. 3d).”

We are satisfied with this edit

Line 180 – “the hexamer oligomerization was further verified by a low-resolution EM structure.” – It would be useful to at least have the FSC in the supplementary for these data, but moreso further details as is standard for EM determination.

We have included a new figure (Extended fig. 3f) with and FSC plot.

###We are satisfied with this edit

Line 181 – “size exclusion chromatography” – Ext Fig 3G is uninformative. The calibration curve needs axis titles.

We have redone Extended figure 3g, and even expanded it. Axis titles have been added.

###We are satisfied with this edit

Its also unclear as to what the hexamer and dodecamer labelling is for. Was there a dodecameric peak? There was not a dodecamer peak and this was originally entered for reference. It has since been removed.

###We are satisfied with this edit

It might be more interpretable to have the calibration proteins marked on the gel filtration chromatogram with actual molecular weights, to make it easier to confirm that ArqI is inferred as hexameric.

We have added this to Extended Fig. 3g.

###We are satisfied with this edit

Line 181 – Intact protein mass spectrometry – The labels on Ext Fig 3H seem to be in the wrong place. Also the figure is of low quality so the molecular weights can't be read. It would be useful to have the predicted molecular weights and calculated MWs given so that it is clear they match.

We revisited the mass spec data and found it was not analyzed correctly. We have decided to remove this figure in this new version of the manuscript as there is plenty of existing evidence of a hexamer.

###It is a shame that this has been removed, the mass spec data would have been an effective way of proving true hexamer formation. However, we are happy that the other data do support the presence of a hexamer, in the absence of this data.

Line 182-184 DALI is not informative of oligomer state. I recommend running both TopSearch (be careful that input PDB has correct TER cards, use both ASU and biological modes) and FoldSeek Multimer, which are much more apt in probing the PDB for relatives of common interfaces. You may get the same null result but it is worth being thorough.

Thank you for the suggestions. We ran TopSearch with 8ECX: The majority of the hits were octamers. We found two hexamers (pdb ID 1X7V (PA3566 protein) and 2GVK (DyP protein)), but these hexamers look nothing like ArqI – they both are trimers of dimers that form a ring (i.e. have a central cavity). FoldSeek, when limited to bacteria, gave the other ArqI structures and then the same hits as Topsearch (NIPSNAP proteins like pdb ID 1VQS) – i.e. octamers. We have not included these results into the manuscript.

###We are satisfied with this edit

Line 185 – “The basic homodimer was most similar to characterized LsrG proteins”. The RMSDs here should state how many residues/C α s are involved in the analysis.

We were constrained for space in the original submission but have added this text back in (lines 267+).

###We are satisfied with this edit

They also indicated that the LsrG proteins are incredibly similar to ArqI. For this reason, it is surprising that no mention is given to their function, given that they are involved in quorum sensing.

We thank the reviewer for pointing this out. We now extensively discuss this similarity in the text now in both the Results section (lines 267-296) and the Discussion section (lines 567-580). We discuss how ArqI might have evolved from LsrG proteins. For example, the MGO structure is a part of the natural ligand of LsrG (P)DPD (new Extended Fig. 4f). Using HDOCK we model into the 'active pocket' of ArqI MGO, DPD and Acetate (AC) as GO was too small.

###We are satisfied with this edit

Also, do they contain the conserved residues described for ArqI?

No, not the modified ones (Arg49 and Cys72), which originally one of the reasons we did not mention a similarity to LsrG family members. These types of binding proteins can have a few residues that change within a pocket which completely chance the substrate binding/function. However other residues, such as Pro87/Gly86, are conserved and found in many (not only LsrG proteins) ABM proteins. LsrG and ArqI comparisons can be seen in the new and expanded version of Extended Fig. 4.

###We are satisfied with this edit

Line 191 – Mutating Pro87 to Gly would have greater implications than preventing Gly86's interaction. The cisbonded Gly-Pro is clearly an important structural feature and also ensures L88 gets buried. Mutation of this Pro to Gly would likely produce a drastic affect structurally, and affect that whole strand, probably affecting at least to the sandwiching Phe residues. I recommend adding a line noting this potential disruption.

We agree here and have added a line addressing this issue in the text. We also have added a figure showing the “Phe gate” (new Figure 2c). See lines 260-265. Interestingly the Pro87>Gly mutation resulted in ArqI localizing to both poles rather than one. At present we have no idea why this happens.

###We are satisfied with this edit

Line 206 – “only seen in subunits C,D and E” – It would be useful to see the electron density of the other chains (are they not modified/is the density absent).

We have added an extra part to Extended Fig. 5b showing the unmodified Arg49 electron density of subunits A,B,F.

###We are satisfied with this edit

Extended figure 5 – Coot screen grabs (Extended Figure 5) are not great quality and could be improved with more modern molecular graphics software.

We believe these figures are of sufficient quality for supplemental figures.

###I still believe that supplemental figures should be of the highest quality.

Line 208 – “possibly assisting in further stabilization of the dimer-dimer contacts” is a non-intuitive way to describe this interaction as this is essentially what creates the trimeric interface of the D3 hexamer. Describing it in a trimer-interface terminology would better highlight the role of this modification in the hexamer.

We now describe this as a trimer interface throughout.

###We are satisfied with this edit

Figure 2 - it would be useful to have a clearer picture of other interactions at this trimeric interface – not just those from the PTM. Are there other residues that provide hydrophobic packing/H-bonding/electrostatic interactions? (e.g. F85) On the whole, there could be more of a structural interrogation in Fig 2.

We have added text describing the possible roles of Gly86-Pro87 and their interrogation of Phe85, which was not mentioned previously in the text. For example, see lines See lines 260-265. Much of this was removed previously due to word limits.

###We are satisfied with this edit

Line 209-211 – It might be useful to explain that all three adducts were modelled into the final model.

We now mention ‘All potential arginine GO-derived adducts’; line 307.

###We are satisfied with this edit

Line 222 – “Extended Fig 5h shows the Arg49 glyoxal derived adduct gave a delta40 amu (with water loss), demonstrating that the modification exists in vivo”. If this is the case then the actual physiological adducts are not those structurally characterised in this manuscript. Why is there no discussion on how the interactions would be different and/or modelling of these adducts into the structure?

Thank you for bringing up this point. We have removed the comment “exists in vivo” as during the immunoprecipitation and processing of the samples, or purification process and crystal formation, the chemistry of these modifications can be affected and result in different adducts depending upon the procedure. Of note, the natural progression of a glyoxal modification is it forms the imidazole-derived (e.g. imidazolone) ring and then will break to the linear form; which is the one that (at least partially) demonstrated electron density in the crystal. The main point we are making in this work is the Arg49 is modified with glyoxal, however seeing precisely what this modification is in vivo is not completely supported by our data here. There is likely complex chemistry that occurs within this active pocket which we will interrogate in another publication.

We have added text describing the above as succinctly as possible given the word limits. Lines 298-323 are dedicated to this discussion in the Results section. We also mention this issue in the Discussion but do not go into detailed and speculative explanations as to the reason for this discrepancy (line 552-553).

###We are satisfied with this edit

Does this protein still form hexamers with the alternate adducts? Native MS, SEC, SEC-MALS or EM of the FLAG-Arql protein would confirm this.

Further interrogation of this adduct would likely be time consuming and complex as we suspect there might be chemistry specific to in vivo conditions. Presently it appears with our crystal structure of the modified WT protein and data in Fig. 3 that the hexamer is retained. Although not as good as taking the time to perform extensive EM experiments to show the hexamer is still formed with a small 6-mer peptide addition, microscopy / FRAP shows that the hexamer is retained with a 26kD sfGFP protein fusion to Arql (see Fig. 3) – which we think is sufficient evidence for the arguments we are making in the text.

Line 225 – again “dimer-dimer” interface might be better described as the trimer interface.

We have changed dimer-dimer to trimeric interface throughout the text.

Line 239 – Because of the reasons outlined above, Arg49 hasn't been convincingly shown to have a role in hexamer stability.

The data show that Arg49 does affect hexamer stability as shown in Fig. 3, albeit not as much as Arg16/Pro87. It might depend on the extent of the PTM on Arg49 but this proved difficult to show in this work. Nevertheless, emphasis/claims of Arg49 contribution to hexamer stability have been dampened throughout the text. Because the Arg49 mutation appears to affect polar migration (and by association its oligomeric state) differently between quantification in Fig. 3c versus b, we added text to describe this discrepancy (lines 350-354). Further, our FRAP data shows statistical significance between the WT and R49Q mutant (Fig. 3e).

###We are satisfied with this edit

Line 240 – does tagging the protein with GFP affect oligomerisation? This is an important control.

We believe the FRAP results, which are an in vivo assessment of the size, sufficiently answers this question.

###FRAP assays are an intracellular assay, and it only shows the diffusion rate of the molecule. You cant say with any certainty what the molecular weight or oligomeric state is of that protein complex. I think the assay should be performed with the proper controls, rather than using proxy controls.

Line 227-232 - Figure 3A is again difficult to interpret due to lack of ability to confirm the MW using the calibration chart. MW standards shown on the chromatogram would allow confirmation of the predicted MW of the later elution peak.

We have inserted MW predictions into Fig. 3a, and put a new chromatogram in Extended Figs 3g and 6a along with theoretical versus calculated masses. Arg49 and Pro87 mutant peaks align perfectly with either WT or Arg16 mutants. We have run these samples on several column over the years and we are very confident in their dimer versus hexamer masses.

###We are satisfied with this edit

Also, it's surprising that a mutation of R49 has little effect on the oligomeric state of the protein, and this is not properly discussed in the results.

As stated above we have added discussion around Arg49 and the hexamer. We would describe this as having a moderate effect, and could of course be due to the percentage of this residue having a PTM in the cell – we think this could be the reason.

###We are satisfied with this edit

Is the R49Q mutant mimicking the interactions of the glyoxal?

No, it is just non-reactive. It would not be considered a mimetic.

###We are satisfied with this

Was crystallisation of the R49Q mutant unsuccessful to see this?

We have not crystallized this mutant, only R16A because it does result in close to 100% dimer. R49Q only a small amount of dimer.

###We are satisfied with this

Has an R49A mutant been tested? (in this assay and subsequent assays using ArqI mutants) This would have prevented any additional contributions from alternative side chains to assess how a non-modifiable sidechain would affect hexamerization.

This is a very good point. We did not test this mutant because the Glutamine better mimics the Arg structure and we thought it would : (i) not be as disruptive to the protein and (ii) not be reactive to glyoxal (a Ala mutant would obviously also not be reactive).

Although using an R49A mutant would be a good idea, we felt that further mutating this and other mutants will not sufficiently add to the outcome of the paper. ArqI structure and function will be further interrogated in future publications.

###We are satisfied with this

Line 267-279 – do any of the utilized mutants (point mutants, as opposed to gene KO) effect QS?

Short answer is no. We have added new data to address this in Fig. 4f. Surprisingly hexamer or dimer both still inhibit QS. We have shown in Extended Fig. 7 that both oligomeric states predicted to bind the PqsA structure. Thus, oligomeric state and QS are uncoupled. We have also mentioned this throughout the text now.

###We are satisfied with this edit

General – it is worth attempting a AlphaFold co-fold of PA0709 and PqsA and reporting outcome (even if null) in the discussion. The discussion would also benefit from noting there are several proteins that contain both entities (PA0709 domains and PqsA-like domains) in the same polypeptide (I did a PFAM search of uniprot and found 61, there will be more).

PA0709 dimer PqsA monomer:

<https://golgi.sandbox.google.com/fold/79ca2dfd958d8436>

PA0709 dimer PqsA dimer:

<https://golgi.sandbox.google.com/fold/75e9defd47bc97d>

Very helpful comment here. We have added (as stated above) new data both AlphaFold and HADDOCK generated ArqI-PqsA predictions with great results. Both Dimer and Hexamer are predicted to bind with high confidence.

We have generated a list PqsA-like domains with ABM protein fusions (found in nature) and provided them in a new SI file. Also, there is text to go with it. Indeed, we have added a whole section on this interesting finding, thank you (lines: 452+ - dedicated Results section to this topic).

###We are satisfied with this edit

X-ray Table needs CC half values for all data and high resolution shell.

Included in an updated Table 1. These data were mistakenly left out of the original submission – thank you.

###We are satisfied with this edit

Figure 3E – This figure should also be rotated slightly to show that there are indirect interactions through the buried HOH246. This water is currently obscured by the glyoxal.

This Figure has been changed accordingly. Thank you.

###We are satisfied with this edit

1 Szklarczyk, D. et al. The STRING database in 2023: protein-protein association networks and functional enrichment analyses for any sequenced genome of interest. *Nucleic Acids Res* 51, D638-D646 (2023).

<https://doi.org/10.1093/nar/gkac1000>

2 Nikata, T. et al. Molecular analysis of the phosphate-specific transport (pst) operon of *Pseudomonas aeruginosa*. *Mol Gen Genet* 250, 692-698 (1996). <https://doi.org/10.1007/BF02172980>

3 Lamarche, M. G., Wanner, B. L., Crepin, S. & Harel, J. The phosphate regulon and bacterial virulence: a regulatory network connecting phosphate homeostasis and pathogenesis. *FEMS Microbiol Rev* 32, 461473 (2008).

<https://doi.org/10.1111/j.1574-6976.2008.00101.x>

4 Wang, T. et al. *Pseudomonas aeruginosa* T6SS-mediated molybdate transport contributes to bacterial competition during anaerobiosis. *Cell Rep* 35, 108957 (2021). <https://doi.org/10.1016/j.celrep.2021.108957>

5 Kulkarni, C. A. et al. ALKBH7 mediates necrosis via rewiring of glyoxal metabolism. *Elife* 9 (2020).

[https://doi.org:10.7554/eLife.58573](https://doi.org/10.7554/eLife.58573)

6 Esmaeili, F., Maleki, V., Kheirouri, S. & Alizadeh, M. The Effects of Taurine Supplementation on Metabolic Profiles, Pentosidine, Soluble Receptor of Advanced Glycation End Products and Methylglyoxal in Adults With Type 2 Diabetes: A Randomized, Double-Blind, Placebo-Controlled Trial. *Can J Diabetes* 45, 39-46 (2021).

[https://doi.org:10.1016/j.cjcd.2020.05.004](https://doi.org/10.1016/j.cjcd.2020.05.004)

Reviewer #2

(Remarks to the Author)

This study identifies a global glyoxal (GO) response in *Pseudomonas aeruginosa* and characterizes the first GO-specific operon, *arqI-gloA2*. Through RNA-seq analysis, the researchers found that this operon, consisting of glyoxalase (*gloA2*) and a previously uncharacterized antibiotic monooxygenase (ABM) domain protein (renamed *ArqI*), is highly induced by GO. *ArqI* was observed to migrate to the flagellar pole upon GO exposure and unexpectedly suppressed the production of the *Pseudomonas* Quinolone Signal (PQS) quorum-sensing molecule. Structural analysis revealed a unique hexamer and a novel GO-derived arginine modification, and functional assays demonstrated that *ArqI* directly interacts with *PqsA*, the first enzyme in PQS biosynthesis. Finally, a sepsis infection model confirmed the *arqI-gloA2* operon is involved in the survival in the host. This is an exciting study that not only uncovers a novel bacterial response to GO but also links aldehyde stress to quorum sensing and perhaps virulence. The discovery of a unique post-translational modification and *ArqI*'s structural features provides valuable insights into bacterial stress adaptation.

The authors have adequately addressed the reviewers' concerns. However, I still have some major points regarding the article.

- In Figure 5, the sepsis data remains inconclusive in supporting the authors' claim of a GO-specific response. This is because two independent processes appear to be occurring in parallel: (1) *GloA2*-mediated GO resistance and (2) *ArqI*-mediated quorum sensing (QS) response. Since only the deletion mutant of the entire operon was used, it remains unclear whether the observed effects are due to *ArqI*, *GloA2*, or both. While the authors have moderated their claims by focusing on the deletion mutants, this result is still insufficient to determine the specific contribution of each gene to virulence.
- Although the authors have moved the GO-response gene expression section to the discussion, it still feels somewhat disconnected from the main *ArqI-gloA2* findings, and overwhelming. This disconnect is evident even in the title and abstract, where it is not mentioned. While it is certainly valuable to discuss the role of the GO-response, the current discussion remains too speculative and lacks experimental validation, making it feel somewhat unfocused and diluting the main valuable message of the study.
I believe this interpretation is worth keeping but should be significantly shortened and made more concise, avoiding excessive speculation. Additionally, in the figure 6, the main finding regarding *PqsA/ArqI-GloA2* is too small and difficult to see, despite being a key message. It would be beneficial to enlarge and emphasize this finding while moving some of the broader interpretations based on RNA-seq to a supplementary figure.
- Overall, over-interpretation can diminish the novelty and impact of the findings. Moving some of this contents - particularly the virulence data in Figure 5 and the summarized scheme of GO-response based on RNA-seq in Figure 6 - to the supplement could help maintain the study's focus and strengthen its clarity.
- Throughout the manuscript, *arqI-gloA2* should be italicized.
- In Line 87 and other instances: the phrase "during the colonization of blood-rich organs" is used. Since the manuscript specifically mentions blood-rich organs and refers to sepsis, can the authors distinguish whether the bacteria colonized a specific organ? Additionally, did the authors monitor CFU levels in the blood? If not, using "survival" instead of "colonization" would be a more appropriate term.
- line 114: *NemR* responds to both MGO and GO. Please also cite the following relevant references:
J Bacteriol. 2010 Aug;192(16):4205-14. doi: 10.1128/JB.01127-09.
Mol Microbiol. 2013 Apr;88(2):395-412. doi: 10.1111/mmi.12192.
- line 130: *Aql* >> *ArqI*
- line 141: For clarity, the strain information should be added at the beginning:
"(MICs) in LB." >> "(MICs) for the *P. aeruginosa* MPAO1 strain in LB."
- line 221: In Figure 1C and its legend (Line 1547), 6 mM GO is stated, while the main text mentions 4 mM GO. This discrepancy should be checked and corrected for consistency.
- Figure 1C: Although the authors stated that *ArqI* does not affect GO resistance, the spot titer assay suggests otherwise. GO resistance appears to be slightly enhanced, by at least 10-fold, in both *::Flag-arqI* and *::arqI* strains, which seems significant and should not be overlooked. This should be discussed in more detail.
By presenting quantified data in Figure 1D, the authors argue that *ArqI* has no effect on GO resistance. If this is the case, the GO panel in Figure 1C should be replaced with a clearer image to avoid misinterpretation. Otherwise, as currently shown, readers may perceive that *ArqI* confers GO resistance.
Additionally, to align with the virulence test, it would be valuable to assess GO/MGO resistance using single in-frame deletion strains. While the authors expressed the respective genes in a double deletion background to dissect their specific roles, testing individual deletions could further clarify their contributions to GO resistance.
- Figure 1C: The *arqI-gloA2* deletion should also be labeled with "EV" or, as in Figure 1D, marked as "(+EV)" for consistency.
- Figure 1CD: Is *::Flag-arqI* correct, or should it be *::arqI-FLAG*? Please verify and ensure consistency.

- Fig 4J legends:

Line 1646: Should FLAG-Arql-BirA* be corrected to Arql-FLAG-BirA*? Please verify for consistency.

Line 1647: FLAB >> FLAG

- Figure 5G: The protein structure displays gray-printed words, which should be removed for clarity.

- Line 445: HDOCK , >> HDOCK, please remove space

- Extended Figure 8: The label "C" is incorrect and should be corrected to "B".

- Line 663: mutant.t >> mutant. : remove t

- Fig 5: 'A' should be removed

- Line 739 C (bold) >> C (bold with underlined)

- Line 755~: The "Construction of Plasmid" section is overly detailed. Information such as primer pairs, restriction sites, and similar details can be summarized in the primer/plasmid table instead.

- Line 901: 'Clean' can be deleted

- Line 1119: PAO1 or MPAO1?

- Line 1197: *Pseudomonas aeruginosa* >> *P. aeruginosa* (italic)

- Line 1250-1260: B-mercaptoethanol >> BME (as already abbreviated in line 1225)

Reviewer #3

(Remarks to the Author)

This study reports the identification and characterization of a novel glyoxal (GO)-specific transcriptional response in *Pseudomonas aeruginosa*, centering on a previously uncharacterized two-gene operon, *arql-gloA2*. Using RNA-seq and functional assays, the authors demonstrate that this operon is uniquely responsive to glyoxal, not methylglyoxal or other aldehydes, and is the most highly upregulated genomic region under GO exposure. The operon encodes *gloA2*, a glyoxalase involved in aldehyde detoxification, and *arql*, an ABM domain-containing protein herein named Aldehyde-responsive quorum-sensing Inhibitor (Arql).

A key novel finding is that Arql modulates virulence by inhibiting *Pseudomonas* Quinolone Signal (PQS) production through a direct interaction with PqsA, the first enzyme in the PQS biosynthesis pathway. The atomic-resolution crystal structure of Arql reveals a unique D3-symmetric hexameric assembly with a previously undescribed GO-derived post-translational modification on a conserved arginine residue (Arg49). Functional analysis shows that this modification, along with key residues involved in oligomerization (Arg16, Pro87), influences Arql localization to the bacterial flagellar pole—a process hypothesized to be involved in spatial regulation of signaling.

Furthermore, the study demonstrates that deletion of the *arql-gloA2* operon significantly impairs bacterial colonization in a murine sepsis model, particularly in blood-rich organs. This finding implicates the operon in host adaptation and survival during systemic infection.

Collectively, this work defines the first GO-specific aldehyde signaling axis in bacteria that integrates environmental stress sensing with quorum-sensing regulation and virulence, revealing new molecular targets for antimicrobial development.

This is a dense, and comprehensive study. I feel that any concerns raised by previous reviewers were met and I do not have any new concerns resulting from the revised manuscript.

Reviewer #2 (Remarks to the Author):

We would first like to thank this reviewer for spending so much time to improve our manuscript for publication.

This study identifies a global glyoxal (GO) response in *Pseudomonas aeruginosa* and characterizes the first GO-specific operon, *arqI-gloA2*. Through RNA-seq analysis, the researchers found that this operon, consisting of glyoxalase (*gloA2*) and a previously uncharacterized antibiotic monooxygenase (ABM) domain protein (renamed *ArqI*), is highly induced by GO. *ArqI* was observed to migrate to the flagellar pole upon GO exposure and unexpectedly suppressed the production of the *Pseudomonas* Quinolone Signal (PQS) quorum-sensing molecule. Structural analysis revealed a unique hexamer and a novel GO-derived arginine modification, and functional assays demonstrated that *ArqI* directly interacts with *PqsA*, the first enzyme in PQS biosynthesis. Finally, a sepsis infection model confirmed the *arqI-gloA2* operon is involved in the survival in the host. This is an exciting study that not only uncovers a novel bacterial response to GO but also links aldehyde stress to quorum sensing and perhaps virulence. The discovery of a unique post-translational modification and *ArqI*'s structural features provides valuable insights into bacterial stress adaptation.

The authors have adequately addressed the reviewers' concerns. However, I still have some major points regarding the article.

- In Figure 5, the sepsis data remains inconclusive in supporting the authors' claim of a GO-specific response. This is because two independent processes appear to be occurring in parallel: (1) *GloA2*-mediated GO resistance and (2) *ArqI*-mediated quorum sensing (QS) response. Since only the deletion mutant of the entire operon was used, it remains unclear whether the observed effects are due to *ArqI*, *GloA2*, or both. While the authors have moderated their claims by focusing on the deletion mutants, this result is still insufficient to determine the specific contribution of each gene to virulence.

We agree, the results shows that the operon is important. This is reflected in the text. By the 'GO response' it is implied that this GO responsive operon is important, which includes an enzyme that mediates GO and *ArqI* which acquired a GO-derived PTM.

- Although the authors have moved the GO-response gene expression section to the discussion, it still feels somewhat disconnected from the main *ArqI-gloA2* findings, and overwhelming. This disconnect is evident even in the title and abstract, where the it is not mentioned. While it is certainly valuable to discuss the role of the GO-response, the current discussion remains too speculative and lacks experimental validation, making it feel somewhat unfocused and diluting the main valuable message of the study.

I believe this interpretation is worth keeping but should be significantly shortened and made more concise, avoiding excessive speculation.

At this point we think these differences are largely due to writing style. We agree that the paper should be more focused on the ArqI findings - and did this to some extent in our newly revised manuscript. We have refocused the discussion on ArqI-GloA2, where we start out describing ArqI-GloA2 and our findings, then leave a section at the end to speculate on host-pathogen interactions. Large amounts of Discussion text were eliminated in the process, including the text describing NosZ/copper speculation (original paper lines 526-548). We feel this is about as far as we can go without losing the overall description / discussion of the findings.

Additionally, in the figure 6, the main finding regarding PqsA/ArqI-GloA2 is too small and difficult to see, despite being a key message. It would be beneficial to enlarge and emphasize this finding while moving some of the broader interpretations based on RNA-seq to a supplementary figure.

This figure has been moved to supplemental (Fig. S9), and a more refined schematic of ArqI-GloA2 function has been put in its place (Fig. 6).

- Overall, over-interpretation can diminish the novelty and impact of the findings. Moving some of this contents - particularly the virulence data in Figure 5 and the summarized scheme of GO-response based on RNA-seq in Figure 6 - to the supplement could help maintain the study's focus and strengthen its clarity.

We feel Fig. 5 is a major set of findings and will remain in the main paper as figure 5. We changed Fig. 6 as stated above.

- Throughout the manuscript, *arqI-gloA2* should be italicized.

We only italicize this if we are describing a gene, or operon/DNA. If we are describing the *arqI-gloA2* as a signaling "axis" we do not. This is consistent with using italics when describing a gene or RNA versus a named axis. Protein would be capitalized.

- In Line 87 and other instances: the phrase "during the colonization of blood-rich organs" is used. Since the manuscript specifically mentions blood-rich organs and refers to sepsis, can the authors distinguish whether the bacteria colonized a specific organ? Additionally, did the authors monitor CFU levels in the blood? If not, using "survival" instead of "colonization" would be a more appropriate term.

This is an excellent point. Unfortunately, we did not measure blood titers and survival, but the WT-infected mice had outward signs of increased morbidity, including enhanced ruffling of fur and decreased activity (anecdotal observation). We have changed "colonization" to "survival".

- line 114: *NemR* responds to both MGO and GO. Please also cite the following relevant references:
J Bacteriol. 2010 Aug;192(16):4205-14. doi: 10.1128/JB.01127-09.
Mol Microbiol. 2013 Apr;88(2):395-412. doi: 10.1111/mmi.12192.

Done.

- line 130: AqI >> ArqI

Done.

- line 141: For clarity, the strain information should be added at the beginning:
"(MICs) in LB." >> "(MICs) for the *P. aeruginosa* MPAO1 strain in LB."

Done.

- line 221: In Figure 1C and its legend (Line 1547), 6 mM GO is stated, while the main text mentions 4 mM GO. This discrepancy should be checked and corrected for consistency.

It should read 6 mM GO – we corrected this. Thank you.

- Figure 1C: Although the authors stated that ArqI does not affect GO resistance, the spot titer assay suggests otherwise. GO resistance appears to be slightly enhanced, by at least 10-fold, in both Δ Flag-arqI and Δ arqI strains, which seems significant and should not be overlooked. This should be discussed in more detail.

By presenting quantified data in Figure 1D, the authors argue that ArqI has no effect on GO resistance. If this is the case, the GO panel in Figure 1C should be replaced with a clearer image to avoid misinterpretation. Otherwise, as currently shown, readers may perceive that ArqI confers GO resistance.

Indeed Fig. 3C was not the best representation of the 5 replicates we did to quantify in Fig. 3D. We thank the reviewer for pointing out this discrepancy. The differences result from starting cultures being a bit variable due to inherent inaccuracies of O.D. measurements/normalizing to optical density. Of note, these experiments seem easy but in fact were quite laborious and took several weeks to optimize and complete – one reason being that *Pseudomonas* does not form perfect colonies following GO treatment (like *E. coli* does). We have therefore taken a more representative figure and used it. Nevertheless, the reviewer is correct, there is a slight increase in resistance (mean % survival difference of 2-fold) that is not, however, statistically significant. We have thus added the line in the text and it now reads:

“Results shown in Fig. 1c demonstrate that deletion of the arqI-gloA2 operon substantially diminished GO resistance, but not MGO resistance. Both the wild-type and FLAG-tagged complemented versions of ArqI were unable to restore measurable GO resistance to statistical significance (however an approximate 2-fold increase was observed compared to the Δ arqI-gloA2 deletion strain), whereas complementation of gloA2 exhibited increased resistance to GO treatment compared to complementing with the full operon (Fig. 1c, d).”

Additionally, to align with the virulence test, it would be valuable to assess GO/MGO resistance using single in-frame deletion strains. While the authors expressed the respective genes in a double deletion background to dissect their specific roles, testing individual deletions could further clarify their contributions to GO resistance.

We discussed this approach while initially designing the mutants, however deletion of one gene in an operon can destabilize the remaining RNA and give false results. We have found this to be the

case many times over the years. Thus, we took the approach of deleting the full operon and complementing the individual components. We could go back and engineer early stop codons but at this point we think that, although this approach is a bit 'cleaner', it would not add any additional information to the paper or arguments we are trying to make. Again, we thank this reviewer for her/his most insightful comments and reading the manuscript to improve it for publication.

- Figure 1C: The arqI-gloA2 deletion should also be labeled with "EV" or, as in Figure 1D, marked as "(+EV)" for consistency.

Fixed.

- Figure 1CD: Is ::Flag-arqI correct, or should it be ::arqI-FLAG? Please verify and ensure consistency.

FLAG-arqI is correct; it's an N-terminal fusion.

- Fig 4J legends:

Line 1646: Should FLAG-ArqI-BirA* be corrected to ArqI-FLAG-BirA*? Please verify for consistency.

Should have read: ArqI-BirA*-FLAG; fixed.

Line 1647: FLAB >> FLAG

Fixed.

- Figure 5G: The protein structure displays gray-printed words, which should be removed for clarity.

Fig. 5G is the infection figure. We could not find the figure the reviewer is referring to.

- Line 445: HDOCK , >> HDOCK, please remove space

Fixed.

- Extended Figure 8: The label "C" is incorrect and should be corrected to "B".

Fixed.

- Line 663: mutant.t >> mutant. : remove t

Fixed.

- Fig 5: 'A' should be removed

Fixed.

- Line 739 C (bold) >> C (bold with underlined)

Fixed.

- Line 755~: The "Construction of Plasmid" section is overly detailed. Information such as primer pairs, restriction sites, and similar details can be summarized in the primer/plasmid table instead.

Since these explanations will go online we will leave them in so others can have a better picture of how we constructed all of these new plasmids, etc.

- Line 901: 'Clean' can be deleted

Done.

- Line 1119: PAO1 or MPAO1?

We used PAO1, as its standard genome annotation/proteome, is very similar to MPAO1.

- Line 1197: *Pseudomonas aeruginosa* >> *P. aeruginosa* (italic)

Done.

- Line 1250-1260: B-mercaptoethanol >> BME (as already abbreviated in line1225)

Done.

Reviewer #3 (Remarks to the Author):

This study reports the identification and characterization of a novel glyoxal (GO)-specific transcriptional response in *Pseudomonas aeruginosa*, centering on a previously uncharacterized two-gene operon, *arqI-gloA2*. Using RNA-seq and functional assays, the authors demonstrate that this operon is uniquely responsive to glyoxal, not methylglyoxal or other aldehydes, and is the most highly upregulated genomic region under GO exposure. The operon encodes *gloA2*, a glyoxalase involved in aldehyde detoxification, and *arqI*, an ABM domain-containing protein herein named Aldehyde-responsive quorum-sensing Inhibitor (ArqI).

A key novel finding is that ArqI modulates virulence by inhibiting *Pseudomonas* Quinolone Signal (PQS) production through a direct interaction with PqsA, the first enzyme in the PQS biosynthesis pathway. The atomic-resolution crystal structure of ArqI reveals a unique D3-symmetric hexameric assembly with a previously undescribed GO-derived post-translational modification on a conserved arginine residue (Arg49). Functional analysis shows that this modification, along with key residues involved in oligomerization (Arg16, Pro87), influences ArqI localization to the bacterial flagellar

pole—a process hypothesized to be involved in spatial regulation of signaling.

Furthermore, the study demonstrates that deletion of the *arqI-gloA2* operon significantly impairs bacterial colonization in a murine sepsis model, particularly in blood-rich organs. This finding implicates the operon in host adaptation and survival during systemic infection.

Collectively, this work defines the first GO-specific aldehyde signaling axis in bacteria that integrates environmental stress sensing with quorum-sensing regulation and virulence, revealing new molecular targets for antimicrobial development.

This is a dense, and comprehensive study. I feel that any concerns raised by previous reviewers were met and I do not have any new concerns resulting from the revised manuscript.